# A partially disordered crystallographic shear block structure as fast-charging negative electrode material for lithium-ion batteries

Yanchen Liu ®[1], Ana Guilherme Buzanich ®[2], Luciano A. Montoro[3], Hao Liu[4], Ye Liu[1], Franziska Emmerling ®[2], Patrícia A. Russo ®[1] ✉ & Nicola Pinna ®[1] ✉

A well-ordered crystalline structure is crucial in battery electrodes, as the dimensionality and connectivity of the interstitial sites inherently influence $Li^+$ ions diffusion kinetics. Niobium tungsten oxides block structures, composed of $ReO_3$-type blocks of specific sizes with well-defined metal sites, are promising fast-charging negative electrode materials. Structural disorder is generally detrimental to conductivity or ion transport. However, here, we report an anomalous partially disordered $Nb_{12}WO_{33}$ structure that significantly enhances Li-ion storage performance compared to the known monoclinic $Nb_{12}WO_{33}$ phase. The partially disordered phase consists of corner-shared $NbO_6$ octahedra blocks of varied sizes, including 5×4, 4×4, and 4×3, with a disordered arrangement of distorted $WO_4$ tetrahedra at the corners of the blocks. This structural arrangement is robust during lithiation/delithiation, exhibiting minor local structure changes during cycling. It enables accelerated Li-ion migration, resulting in promising fast-charging performance, namely, 62.5 % and 44.7 % capacity retention at 20 C and 80 C, respectively. This study highlights the benefits of introducing disorder into niobium tungsten oxide shear structures, through the establishment of clear structure-performance correlations, offering guidelines for designing materials with targeted properties.

Rechargeable Li-ion batteries have been widely applied as energy storage systems in portable electronics and electric vehicles[1]. The rapidly increasing demand for vehicles necessitates the development of fast-charging electrode materials that prioritize high safety standards. Negative electrode materials play a key role in governing the ion kinetics in the storage systems[2]. In practical applications, commercial graphite, which operates at low potential (-100 mV vs. $Li^+$/Li), is limited for high-rate usage due to the formation of a passivating solid-electrolyte interphase (SEI), and thus by sluggish ion insertion kinetics

and risk of Li dendrite formation caused by increased overpotential at high current density[3,4].

Recently, niobium tungsten oxides with crystallographic shear structure have shown promise as negative electrode materials with fast-charging properties owing to their moderate working potentials, open frameworks that are suited for rapid $Li^+$ diffusion, and multi-electron redox reactions[5–7]. Crystallographic shear block structures are made by the assembly of $ReO_3$-type blocks of corner-shared octahedra with specific sizes, which are joined together via edge-shared

[1]Department of Chemistry and The Center for the Science of Materials Berlin, Humboldt-Universität zu Berlin, Berlin, Germany. [2]Bundesanstalt für Materialforschung und -prüfung (BAM), Berlin, Germany. [3]Department of Chemistry, Universidade Federal de Minas Gerais, Belo Horizonte, MG, Brazil. [4]Institute for Applied Materials (IAM), Karlsruhe Institute of Technology (KIT), Eggenstein-Leopoldshafen, Germany. ✉e-mail: patricia.russo@hu-berlin.de; nicola.pinna@hu-berlin.de

octahedra. Some of those structures additionally contain metal tetrahedral sites joining the blocks[8–10]. These compounds generally present high crystallinity, as their formation takes place at relatively high temperatures (>700 °C). For instance, $Nb_{12}WO_{33}$ ($m$-$Nb_{12}WO_{33}$) has monoclinic structure that consists of 3 × 4 $ReO_3$-type blocks connected by edge-sharing octahedra, as well as tetrahedra at the corners of the blocks. Its structure allows for high Li-ion kinetics, but still remains insufficient to meet the high-rate requirements for practical applications[7,11,12]. Approaches to further improve the Li insertion-deinsertion kinetics include nanostructure engineering, such as the reduction of particle size or the introduction of porosity. These methods were found to be an effective way to facilitate Li-ion mobility in the niobium tungsten oxides $Nb_{14}W_3O_{44}$ and $Nb_{12}WO_{33}$[13–16]. However, they reduce the material's tap density, thereby compromising the volumetric energy density. In recent years, exploring the potential of modifications at the atomic level in oxides, such as introducing disorder in the occupation of the crystallographic sites by the cations, has emerged as an effective method to regulate the intrinsic properties, such as long-range ionic conductivity, and thus enhancing the overall performance of a material[17–20].

In this work, we propose the strategy of introducing disorder into a niobium tungsten oxide block structure for producing an improved negative electrode material with fast-charging properties for Li-ion batteries. By controlling the synthesis conditions, a partially disordered $Nb_{12}WO_{33}$ phase ($dt$-$Nb_{12}WO_{33}$) can be obtained, whose structure is ordered along the $c$ axis whereas the presence of little ordered shear blocks in a variety of sizes along $ab$ plane, including 5 × 4, 4 × 4, and 4 × 3, causes a little ordered arrangement of the tungsten tetrahedra at the corners of the blocks, as well as distortion of the $WO_4$ tetrahedra. Benefiting from the inherent flexibility of the partially disordered structural arrangement with local distortion, the local structure of $dt$-$Nb_{12}WO_{33}$ was found to experience only slight alterations during Li-ion insertion compared to that of $m$-$Nb_{12}WO_{33}$. Structural disruptions that occur in $m$-$Nb_{12}WO_{33}$ are mitigated in $dt$-$Nb_{12}WO_{33}$, which is able to maintain its overall structural integrity. In addition, the partially disordered structure ensures the presence of optimized channels for Li-ion transport during lithiation and delithiation, enhancing the rate performance.

## Results

### Structure of the partially disordered $Nb_{12}WO_{33}$

$m$-$Nb_{12}WO_{33}$ and $dt$-$Nb_{12}WO_{33}$ were synthesized via a solvothermal method, followed by thermal treatment in air at 1100 °C and 900 °C, respectively. The X-ray diffraction (XRD) pattern of $m$-$Nb_{12}WO_{33}$ (Fig. 1a and Supplementary Fig. 1) can be indexed to the known monoclinic ($C2$) phase of $Nb_{12}WO_{33}$, whose structure was first determined by Roth and Wadsley in 1965, and the only phase reported so far for the $Nb_2O_5$:$WO_3$ ratio of 6:1 in block structures of niobium tungsten

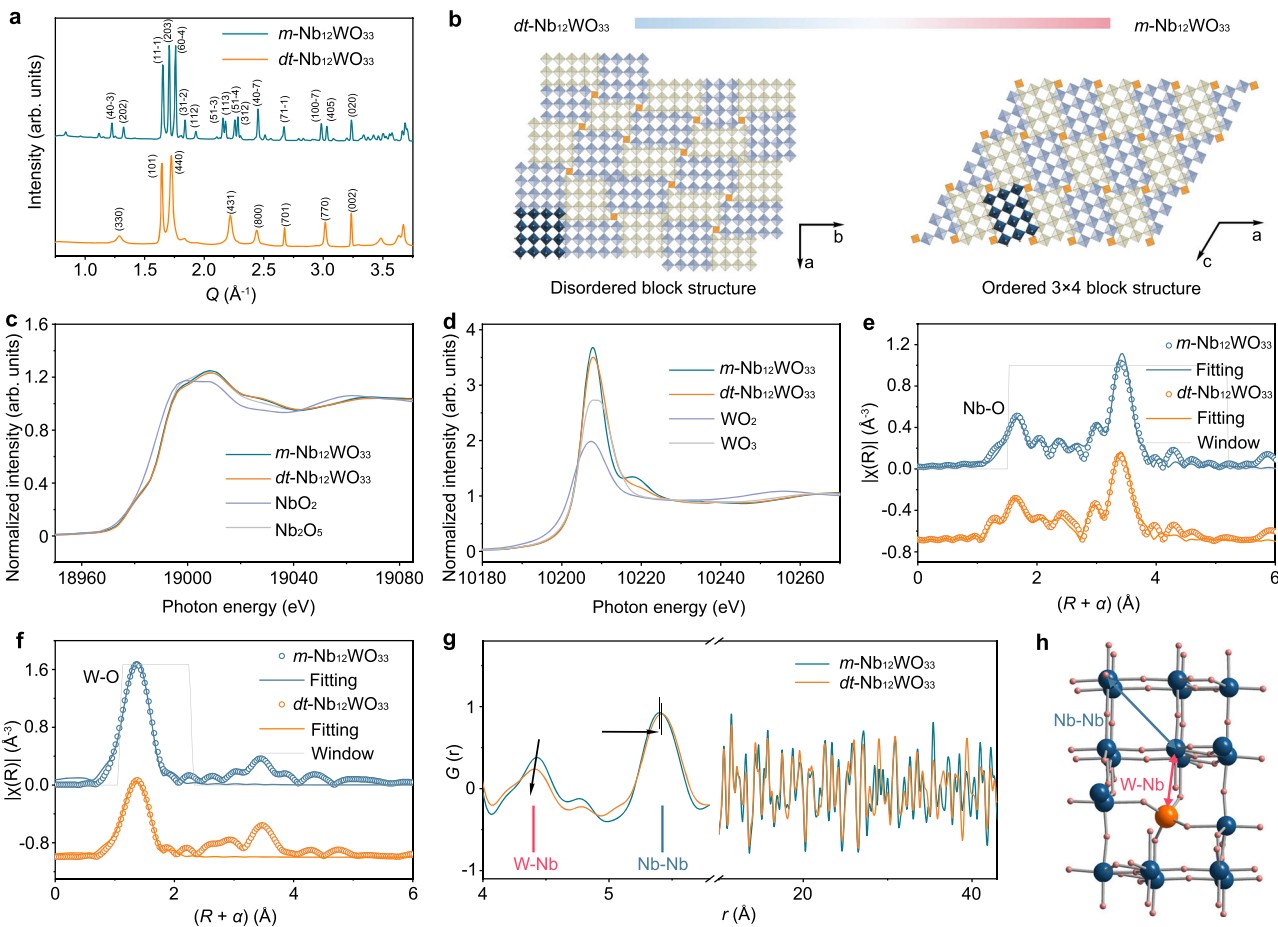

**Fig. 1 | Structural characterization of $m$-$Nb_{12}WO_{33}$ and $dt$-$Nb_{12}WO_{33}$. a** XRD patterns of $m$-$Nb_{12}WO_{33}$ and $dt$-$Nb_{12}WO_{33}$. **b** Schematic illustration of the structure for $m$-$Nb_{12}WO_{33}$ and $dt$-$Nb_{12}WO_{33}$; blue and green squares represent the octahedral sites, while orange squares denote tetrahedral sites (blue and green blocks are offset by 1/2 $b$ for $m$-$Nb_{12}WO_{33}$ and 1/2 $c$ for $dt$-$Nb_{12}WO_{33}$). Nb K-edge (**c**) and W $L_{III}$-edge (**d**) XANES spectra of $m$-$Nb_{12}WO_{33}$ and $dt$-$Nb_{12}WO_{33}$. **e** Fitting of the Nb K-edge first and second coordination shells in the FT-EXAFS spectra of $m$-$Nb_{12}WO_{33}$ and $dt$-$Nb_{12}WO_{33}$. **f** Fitting of the W $L_{III}$-edge first coordination shell in the FT-EXAFS spectra of $m$-$Nb_{12}WO_{33}$ and $dt$-$Nb_{12}WO_{33}$; $\alpha$ represents the phase shift between the apparent and actual atomic distances. **g** PDF pattern of $m$-$Nb_{12}WO_{33}$ and $dt$-$Nb_{12}WO_{33}$. **h** Schematic illustration of type I and VI cavities viewed along the $c$ axis; blue, orange and pink spheres represent Nb, W and O atoms, respectively; the atomic pairs Nb–Nb (blue) and W–Nb (red) in the PDF pattern in g are represented by blue and red lines, respectively, in the schematic structure in (**h**).

oxides[21]. The structure consists of $ReO_3$-type blocks of corner-sharing octahedra of size 3 × 4. These blocks are connected to each other by edge-sharing, with tetrahedral sites located between the individual blocks. X-ray diffraction studies by Roth and Wadsley[21] and Density Functional Theory (DFT) calculations reported by Koçer et al.[10] suggest that tungsten strongly prefers to occupy the tetrahedral sites at the corners of the blocks. On the other hand, Cheetham and Allen[22] also found a strong preference of the tungsten for the octahedral site at the center of the blocks in $Nb_{14}W_3O_{44}$ using neutron diffraction data. To determine the structure of $m$-$Nb_{12}WO_{33}$, we synthesized larger crystals (6–19 μm in length) (Supplementary Fig. 2) than those obtained via the solvothermal method (1–3 μm) and performed single-crystal XRD measurements. The crystallographic parameters obtained from the structure refinement are listed in Supplementary Table 1. The results, with a $R$ factor of 3.83%, gave the lattice parameters $a$ = 22.3002(15) Å, $b$ = 3.8279(2) Å, $c$ = 17.7490(12) Å and $\beta$ = 123.338°. According to the refinement data in Table S1, the tungsten atoms show a preference for the tetrahedral sites at the corners of the blocks. We have tried to replace the Nb1 site at the center of the blocks with a small percentage of tungsten atoms, but the refinement was not stable. However, very recently, Cardon et al.[23] have prepared high-quality centimeter-sized $m$-$Nb_{12}WO_{33}$ crystals for structure determination. The structural refinement, with an R factor of 3.62%, identified a higher symmetry space group ($I2/m$) than previously reported ($C2/m$). Roth and Wadsley[21] reported 100% Nb occupancy of the octahedral sites and 100% W occupancy of the tetrahedral sites. The results of Cardon et al.[23] show that tungsten has a strong preference for the tetrahedral sites, with a similar overall distribution of W and Nb as that obtained by Roth and Wadsley, except for the octahedral site located at the center of the blocks, for which the refinement indicates a 7% W occupancy. Therefore, tungsten mainly occupies the tetrahedral sites in $m$-$Nb_{12}WO_{33}$, with a small percentage additionally occupying the octahedral site at the center of the blocks. The same authors also found that, while the tetrahedral sites should be exclusively occupied by W in $Nb_{14}W_3O_{44}$, tungsten also has a relatively strong preference for the octahedral site at the center of the blocks, with a 26% W occupancy. Aberration-corrected scanning transmission electron microscopy (AC-STEM) imaging (Supplementary Fig. 3) shows the atomic arrangement of the niobium and tungsten in $m$-$Nb_{12}WO_{33}$ according to the above description. The bright spots correspond to the metals, with the brightest ones arising from the W atoms due to its higher $Z$-contrast. The metal atoms are seen arranged periodically according to the monoclinic lattice cell, with the tungsten atoms located mainly at the corners of the blocks, and shear planes seen along the structure.

The diffractogram of $dt$-$Nb_{12}WO_{33}$ (Fig. 1a) reveals a crystalline phase for the $Nb_{12}WO_{33}$ block structure, which has not been previously reported. The pattern is consistent with a tetragonal structure (space group $I4/mmm$) similar to that of M-$Nb_2O_5$[24,25]. Inductively coupled plasma-optical emission spectroscopy (ICP-OES) analysis confirms that the metal compositions of both compounds are around 12 Nb:1 W, consistent with the formula $Nb_{12}WO_{33}$ (Supplementary Table 2). Scanning electron microscopy (SEM) and transmission electron microscopy (TEM) images (Supplementary Figs. 4 and 5) show that the particle size of $dt$-$Nb_{12}WO_{33}$ ranges from 200 to 500 nm, while those of $m$-$Nb_{12}WO_{33}$ are in the range of 1–3 μm due to its higher synthesis temperature. The high-resolution TEM (HR-TEM) image of $dt$-$Nb_{12}WO_{33}$ in Supplementary Fig. 5e shows lattice fringes with a spacing of 0.28 nm, matching the d-spacing of the (501) planes of $dt$-$Nb_{12}WO_{33}$. The (260), (442), and (2–22) planes in the selected area electron diffraction (SAED) pattern of $dt$-$Nb_{12}WO_{33}$ are typically observed in the diffraction pattern of a tetragonal ($I4/mmm$) structure. The lattice fringes with a spacing of 0.36 nm observed in Supplementary Fig. 5b correspond to the (203) planes of the monoclinic structure of $m$-$Nb_{12}WO_{33}$, which is further confirmed through the SAED pattern. The uniform element distribution in the energy dispersive X-ray spectroscopy (EDS) maps reflects the homogeneity in the composition of the particles of both $m$-$Nb_{12}WO_{33}$ and $dt$-$Nb_{12}WO_{33}$ (Supplementary Figs. 5 and 6).

The oxidation state and local structure of the metals in the niobium tungsten oxides were probed by X-ray absorption spectroscopy. The Nb K-edge X-ray absorption near-edge structure (XANES) spectrum of $dt$-$Nb_{12}WO_{33}$ is identical to those of $m$-$Nb_{12}WO_{33}$ and $Nb_2O_5$, including the position of the edge, indicating that the oxidation state of niobium is 5+ (Fig. 1c). The niobium is octahedrally coordinated. The small pre-edge feature centered at ca. 18,980 eV indicates, however, a loss of symmetry from the perfect octahedral symmetry. Distortion of the octahedra, which arises partially from the second-order Jahn–Teller effect associated with the $d^0$ electron configuration of the $Nb^{5+}$ cations, leads to p–d orbital mixing and consequently to the dipole-forbidden s–d transition responsible for the appearance of the pre-edge peak[26,27]. The white line in the W $L_{III}$-edge on the XANES spectra (Fig. 1d) corresponds to electron transitions from the $2p_{3/2}$ core level to quasi-bound $5d$(W) + $2p$(O) mixed states[28]. The position of the edge is influenced by the oxidation state of the metal and suggests that tungsten is in the oxidation state 6+ in both $m$-$Nb_{12}WO_{33}$ and $dt$-$Nb_{12}WO_{33}$, as in $WO_3$. The edge is caused mainly by $2p{\rightarrow}5d$ electron transitions. In addition to the oxidation state, the coordination environment, including local disorder, also affects the intensity and shape of the white line, due to its effect on the density of states and distribution of unoccupied d states. The splitting of the $5d$ orbitals of the metal by the ligand field results in the splitting of the edge and can provide insights into the $5d$ electronic states and thus the coordination environment of the metal[29–31]. The magnitude of the ligand field split is highest for cations in an octahedral field, followed by cations in a distorted octahedral environment, and lowest for cations in tetrahedral coordination. Therefore, it has been reported that the white line for tungsten in octahedral coordination exhibits two peaks, while only one broad peak with an indistinct maximum is observed for W species in a distorted octahedral environment. Tungsten in tetrahedral symmetry led to one sharp asymmetrical peak[31]. The second derivative of the W $L_{III}$-edge XANES spectra allows a clearer visualization of the edge splitting. The second derivative of the $m$-$Nb_{12}WO_{33}$ and $dt$-$Nb_{12}WO_{33}$ spectra (Supplementary Fig. 7) shows only one peak, suggesting that tungsten mainly occupies the tetrahedral sites of these materials, contrasting with the clearer splitting observed for $WO_3$, in which tungsten is exclusively in octahedral coordination. The spectrum of $m$-$Nb_{12}WO_{33}$ shows a post-edge feature, which is typically found in the spectra of materials with regular $WO_4$ tetrahedra, and is almost absent from the $dt$-$Nb_{12}WO_{33}$ spectrum[31,32]. In addition, the intensity of the edge is lower for $dt$-$Nb_{12}WO_{33}$ compared to $m$-$Nb_{12}WO_{33}$, which is indicative of distortion in the former. This is because less distortion in the coordination environment around the tungsten tends to increase the overlap between W d-orbitals and O p-orbitals, which can raise the intensity of the white line due to increased electron density at the metal. Thus, the results suggest that the $WO_4$ and $WO_6$ polyhedra in $dt$-$Nb_{12}WO_{33}$ are more distorted.

Fitting of the extended X-ray absorption fine structure (EXAFS) spectra provided additional insights about the local structure of Nb and W in $m$-$Nb_{12}WO_{33}$ and $dt$-$Nb_{12}WO_{33}$. The Nb K-edge and W $L_{III}$-edge Fourier transform EXAFS spectra of the materials are shown in Fig. 1e, f, and the corresponding fitting results are listed in Supplementary Tables 3–6. The bond distances for the first Nb–O coordination shell are similar in $m$-$Nb_{12}WO_{33}$ and $dt$-$Nb_{12}WO_{33}$, and the same is found for the W–O first coordination shell. However, the coordination number of Nb and W in the first shell is smaller for $dt$-$Nb_{12}WO_{33}$, in agreement with the lower peak intensity of the first shell, which suggests a more disordered structure.

In the Raman spectra of $m$-$Nb_{12}WO_{33}$ and $dt$-$Nb_{12}WO_{33}$ (Supplementary Fig. 8), the three bands at 266, 629, and 992 $cm^{-1}$ are ascribed to the bending modes of Nb–O–Nb linkages, stretching modes of

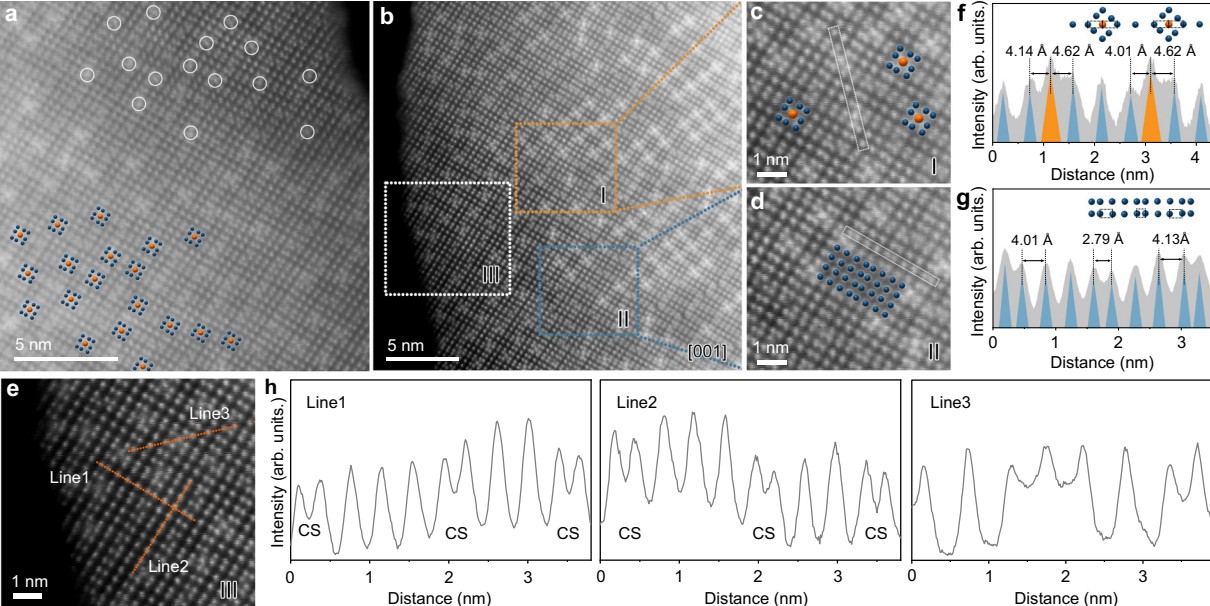

**Fig. 2 | AC-STEM study of *dt*-Nb$_{12}$WO$_{33}$.** (**a**) and (**b**) AC-STEM images of *dt*-Nb$_{12}$WO$_{33}$ (metals in octahedral sites and tetrahedral sites are indicated by blue and orange spheres, respectively). **c–e** Magnified AC-STEM images of the regions I, II, and III delimited in (**b**). **f–h** Line intensity profiles from the regions delimited in (**c**), (**d**), and (**e**), respectively. CS represents the crystallographic shear planes made by edge-sharing octahedra.

NbO$_6$ octahedra, and symmetrical Nb–O stretching vibrations of the NbO$_6$ edge-shared octahedra[33]. The band at 914 cm$^{-1}$ in *m*-Nb$_{12}$WO$_{33}$ could be assigned to the symmetric stretching vibrations of W–O bonds in WO$_4$, whereas a weak intensity band is observed for *dt*-Nb$_{12}$WO$_{33}$, suggesting distortion of the WO$_4$ tetrahedra[34,35].

AC-STEM images of *dt*-Nb$_{12}$WO$_{33}$ viewed along the [001] zone axis are shown in Fig. 2. Contrary to what is observed for *m*-Nb$_{12}$WO$_{33}$, the brighter spots (some delimited by circles in image a), corresponding to tungsten atoms in tetrahedral coordination, are heterogeneously distributed, reflecting the partially disordered nature of the structure. The WO$_4$ tetrahedra are found at the blocks' junctions. Figure 1b displays a schematic illustration of the structure of *dt*-Nb$_{12}$WO$_{33}$. While *m*-Nb$_{12}$WO$_{33}$ consists of 4 × 3 blocks of corner-shared octahedra, the structure of *dt*-Nb$_{12}$WO$_{33}$ contains blocks in a variety of sizes, such as 4 × 4, 4 × 3, 5 × 3, 6 × 3, 5 × 4, and 3 × 3, thus resulting in the non-uniform distribution of the WO$_4$ (Supplementary Fig. 9). A similar variation in block size has been reported several decades ago for M-Nb$_2$O$_5$ stabilized with WO$_3$[36]. The delimited regions I and II in Fig. 2b are shown in Fig. 2c and Fig. 2d, respectively. The intensity line profile corresponding to the region selected in Fig. 2c (Fig. 2e) shows the variation in the distances between W in tetrahedral sites and neighboring Nb atoms (higher intensity peaks correspond to W). The W–Nb distances around the tetrahedral site show a variation between 4.14 Å and 4.62 Å, which further supports the existence of tetrahedral distorted sites in the *dt*-Nb$_{12}$WO$_{33}$ structure. In addition, the intensity line profile from the region delimited in Fig. 2d shows the variation in the Nb–Nb distances within two different 4 × 4 blocks, reflecting the local distortion within the blocks. The intensity profiles in Fig. 2h correspond to Line1 and Line2 in (e) show that the intensity of the block centers is generally higher than that in the crystallographic shear (CS) planes. In addition, although the positions associated with the tetrahedral sites in Fig. 2f are brighter than those in the CS planes, the differences are not as high as expected if the tetrahedral sites were fully occupied with tungsten atoms[37]. Within the thin area at the edge of the particle delimited by Line3 in (e), the tetrahedral site shows slightly higher intensity compared with the octahedral sites at the center of the blocks (Fig. 2h). These results suggest that tungsten atoms predominantly occupy the tetrahedral sites, with some possible occupation of the octahedral sites at the center of the blocks, with similarities to the Nb and W distribution in *m*-Nb$_{12}$WO$_{33}$ discussed above.

Pair distribution function (PDF) measurements were carried out to investigate the atomic pair distribution in the structures (Fig. 1g). The W–Nb distances between W in the tetrahedral sites and Nb in octahedral sites at the corners of the blocks decrease from 4.43 in *m*-Nb$_{12}$WO$_{33}$ to 4.41 Å in *dt*-Nb$_{12}$WO$_{33}$, while the differences in amplitude are more striking, reflecting the structural heterogeneity introduced by the local distortion. In addition, the Nb–Nb distances within shear blocks expand from 5.40 to 5.42 Å, suggesting that more spacious pathways are available in *dt*-Nb$_{12}$WO$_{33}$ for the diffusion of the lithium ions.

The PDF peaks in the long-range (>10 Å) of *dt*-Nb$_{12}$WO$_{33}$ are broader and less intense than those of the monoclinic structure, which is consistent with the presence of partial structural disorder. The local distortion may arise from non-uniform strain distribution within the lattice of *dt*-Nb$_{12}$WO$_{33}$, which is formed at lower temperatures, leading to variations in the geometric configuration of the tetrahedral sites.

## Li-ion insertion/extraction behavior

The electrochemical performance of *m*-Nb$_{12}$WO$_{33}$ and *dt*-Nb$_{12}$WO$_{33}$ was evaluated in half-cells, with Li foil serving as the counter electrode. The galvanostatic charge/discharge profiles of the initial first cycle within the voltage range 1.0–3.0 V at 0.5 C (1 C = 190.7 mA g$^{-1}$, based on one electron transfer per transition metal) are presented in Fig. 3a. *m*-Nb$_{12}$WO$_{33}$ exhibits an initial discharge and charge capacities of 251.5 and 219.4 mA g$^{-1}$ (corresponding to the insertion and extraction of 17.1 and 15.0 Li$^+$ per formula, respectively), leading to a Coulombic efficiency of 87.3%, which results from the irreversible Li$^+$ trapping in the crystal host. The Li insertion behavior of *m*-Nb$_{12}$WO$_3$ was first studied by Cava et al., through the reaction with *n*-BuLi to form the insertion compound Li$_{10.7}$Nb$_{12}$WO$_{33}$[38]. A few studies have been reported more recently, with results that are consistent with those obtained here[12,15]. In contrast, a higher initial discharge and charge capacities of 269.7 and 257 mAh g$^{-1}$ (18.4 and 17.5 Li$^+$) are obtained for *dt*-Nb$_{12}$WO$_{33}$, corresponding to an average voltage of 1.48 and 1.60 V, respectively, which are lower than those for *m*-Nb$_{12}$WO$_{33}$ (1.53 and 1.66 V).

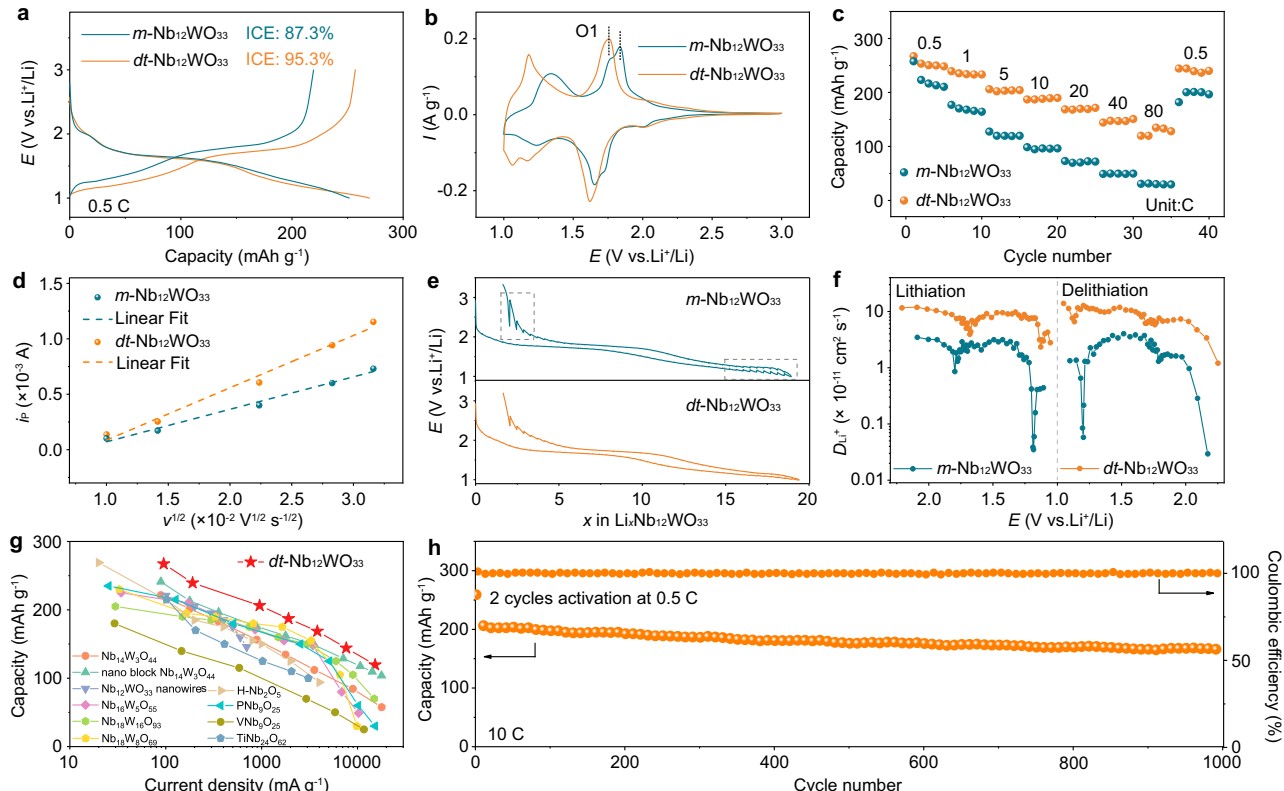

**Fig. 3 | Li-ion storage properties and diffusion kinetics in $m$-Nb$_{12}$WO$_{33}$ and $dt$-Nb$_{12}$WO$_{33}$. a** Charge and discharge profiles of $m$-Nb$_{12}$WO$_{33}$ and $dt$-Nb$_{12}$WO$_{33}$ at 0.5 C. **b** CV curves of $m$-Nb$_{12}$WO$_{33}$ and $dt$-Nb$_{12}$WO$_{33}$ at 0.1 mV s$^{-1}$. **c** Rate capability of $m$-Nb$_{12}$WO$_{33}$ and $dt$-Nb$_{12}$WO$_{33}$, as evaluated by discharge capacity. **d** Linear relationship between the peak current ($i_P$) and the square root of the scan rate ($v^{1/2}$) for peak O1. **e** GITT profiles of $m$-Nb$_{12}$WO$_{33}$ and $dt$-Nb$_{12}$WO$_{33}$ as a function of $x$ in Li$_x$Nb$_{12}$WO$_{33}$. **f** Variation of the $D_{Li+}$ determined by GITT. **g** Comparison of the rate capability between $dt$-Nb$_{12}$WO$_{33}$ and other reported niobium-based oxides with block structures as a function of discharge capacity (the references cited in the Supplementary Table S7). **h** Cycling performance of $dt$-Nb$_{12}$WO$_{33}$ at 10 C, as evaluated by discharge capacity.

Compared to $m$-Nb$_{12}$WO$_{33}$, which experiences a higher first-cycle capacity loss of 12.7%, $dt$-Nb$_{12}$WO$_{33}$ exhibits a promising initial Coulombic efficiency of 95.3%, indicating its potential for practical application. The main pairs of redox peaks (1.6–1.7 V) shown in the cyclic voltammograms (CV) of $m$-Nb$_{12}$WO$_{33}$ and $dt$-Nb$_{12}$WO$_{33}$ in Fig. 3b can be assigned to a two-phase transition reaction. An obvious voltage decrease in the ranges 1.7–1.6 and 1.4–1.1 V is observed for $dt$-Nb$_{12}$WO$_{33}$ when compared with $m$-Nb$_{12}$WO$_{33}$, indicating a lower average voltage for $dt$-Nb$_{12}$WO$_{33}$. The CV and d$Q$/d$V$ results of the initial three cycles for $dt$-Nb$_{12}$WO$_{33}$ display a highly overlapped profile (Supplementary Figs. 10 and 11), revealing its good structural stability.

As depicted in Fig. 3c and Supplementary Fig. 12, $dt$-Nb$_{12}$WO$_{33}$ shows better rate performance than $m$-Nb$_{12}$WO$_{33}$, with a reversible capacity of 168.8 and 144.1 mAh g$^{-1}$ at 20 C and 40 C, respectively. Even at the high rate of 80 C, the capacity remains at 119.5 mAh g$^{-1}$, corresponding to a high capacity retention of 44.7% from 0.5 to 80 C. In contrast, the capacity of $m$-Nb$_{12}$WO$_{33}$ declines rapidly, and a small capacity of 30.9 mAh g$^{-1}$ can be obtained at 80 C, corresponding to a capacity retention of 12.0%. Besides, the large voltage polarization of $m$-Nb$_{12}$WO$_{33}$ with increasing current density seen in Supplementary Fig. 13 due to the sluggish ion transport kinetics contrasts with the lower polarization increase for $dt$-Nb$_{12}$WO$_{33}$. The enhanced rate capability and reduced voltage polarization of $dt$-Nb$_{12}$WO$_{33}$ are a consequence of its fast reaction kinetics and smaller diffusion barriers for Li$^+$ ions across multiple sites within the partially disordered structure. CV curves were measured at various scan rates, and a positive correlation between the peak currents ($i_P$) and the square root of the scan rate ($v^{1/2}$) during Li$^+$ insertion and extraction is found (Fig. 3d and Supplementary Fig. 14). The relationship between the peak current and

the scan rate ($v$) is given by the Randles-Ševčík equation, $i_p = 2.69 \times 10^5 n^{3/2} C_0 A D^{1/2} v^{1/2}$, where $n$ is the number of electrons per reaction species, $C_0$ is the Li$^+$ concentration in the lattice, $A$ is the area of electrode, and $D$ represents the Li$^+$ diffusion coefficient. The parameters $n$, $C_0$, and $A$ have the same value for both materials[39]. Application of the Randles-Ševčík equation results in an average slope for $dt$-Nb$_{12}$WO$_{33}$ that is approximately twice as large as that of $m$-Nb$_{12}$WO$_{33}$. The Li$^+$ diffusion coefficients in $m$-Nb$_{12}$WO$_{33}$ and $dt$-Nb$_{12}$WO$_{33}$ were estimated via the galvanostatic intermittent titration technique (GITT), as shown in Fig. 3e and Supplementary Figs. 15–19. Figure 3e and Supplementary Fig. 15 display the recorded GITT profiles of the two materials. The corresponding Li$^+$ diffusion coefficients ($D_{Li+}$) as a function of voltage (Fig. 3f) were determined from the GITT profiles by using the simplified Fick's second law of diffusion[40]. The $D_{Li+}$ of $dt$-Nb$_{12}$WO$_{33}$ are in the range $1.4 \times 10^{-10}$–$1.2 \times 10^{-11}$, and are significantly larger than those for $m$-Nb$_{12}$WO$_{33}$ ($3.7 \times 10^{-11}$–$2.9 \times 10^{-13}$). In the low voltage region under 1.3 V and at the end of the charge stage of the GITT curves, the voltage relaxation for $m$-Nb$_{12}$WO$_{33}$ is clearly higher than for $dt$-Nb$_{12}$WO$_{33}$, as a result of greater charge transfer and mass transfer resistances. Thus, the minimum $D_{Li+}$ value for $dt$-Nb$_{12}$WO$_{33}$ is two orders of magnitude above that for $m$-Nb$_{12}$WO$_{33}$. The extremely small diffusion coefficient is a result of the two-phase transition occurring below 1.3 V, which is accompanied by a free-energy barrier, due to the coherency strain energy and interfacial energy between the two phases. This leads to sluggish ion diffusion kinetics and voltage hysteresis, which worsens with increasing current rates. In addition, the pseudocapacitive storage of Li$^+$ in $m$-Nb$_{12}$WO$_{33}$ and $dt$-Nb$_{12}$WO$_{33}$ was investigated through the relationship between peak current ($i_p$) and scan rate ($v$) taken from the CV curves measured at different scan rates[41], described by the

equation $i_p = av^b$. The resulting $b$ values indicate that the electrochemical process is partially controlled by capacitive behavior in $m$-$Nb_{12}WO_{33}$ and $dt$-$Nb_{12}WO_{33}$ (Supplementary Fig. 20). The $b$ value of $dt$-$Nb_{12}WO_{33}$ is slightly higher than that of $m$-$Nb_{12}WO_{33}$, suggesting a larger capacitive contribution in the former. $dt$-$Nb_{12}WO_{33}$ exhibits competitive electrochemical performance compared with the previously reported Nb-based negative electrode materials (Fig. 3g and Supplementary Table 7)[5,6,13,15,42–45], although it should be noted that direct comparisons are challenging due to experimental differences in electrode preparation, such as carbon content and mass loading.

The types of cavity sites for Li-ion insertion present in Wadsley-Roth phases of Nb-based oxides were first categorized by Cava et al.[38]. The structure of $dt$-$Nb_{12}WO_{33}$ possesses cavity sites of the types I, II, III, V, and VI, which provide potential pathways for Li-ion diffusion. The position of these cavities in the structure is schematized in Supplementary Fig. 21. Koçer et al.[11] have performed DFT calculations to determine the energetically favorable pathways for Li$^+$-ion motion within niobium tungsten oxide block structures. Li$^+$ diffusion was found to be energetically unfavorable across shear planes and through type VI cavities, which are formed by tetrahedrally coordinated tungsten at the junctions of the blocks and octahedral sites in the vertices of the blocks (Fig. 1b and Supplementary Fig. 21). In contrast, Li-ion diffusion through the sites within the blocks was found to involve lower energy barriers, making these the most likely pathways in the block structures. As suggested by the PDF data, the partial disorder in $dt$-$Nb_{12}WO_{33}$ shortens the distances between W in tetrahedral sites and Nb in octahedral sites at the corner of the blocks, making it very unlikely for Li-ions to diffuse via type VI cavities. However, disorder also leads to larger distances between the metal sites within the blocks, indicating wider pathways for lithium diffusion inside the blocks of $dt$-$Nb_{12}WO_{33}$, which contributes to the larger Li$^+$ diffusion coefficients and better rate capability of $dt$-$Nb_{12}WO_{33}$ compared to $m$-$Nb_{12}WO_{33}$.

The long-term cycling stability of $m$-$Nb_{12}WO_{33}$ and $dt$-$Nb_{12}WO_{33}$ is shown in Fig. 3h, Supplementary Figs. 22 and S23. $m$-$Nb_{12}WO_{33}$ exhibits a specific capacity of 148.8 mAh g$^{-1}$ at 5 C after 200 cycles, along with a capacity retention of 89.6%. $dt$-$Nb_{12}WO_{33}$ maintains a higher capacity of 196.0 mAh g$^{-1}$, corresponding to a comparable retention of 90.5%. Moreover, after 1000 cycles at 10 C, $dt$-$Nb_{12}WO_{33}$ is still able to deliver a high capacity of 165.2 mAh g$^{-1}$ (Fig. 3h), which corresponds to a good capacity retention of 80.5%.

To further evaluate the application potential of $dt$-$Nb_{12}WO_{33}$, full cells ($dt$-$Nb_{12}WO_{33}$||LFP) were constructed using commercial LiFePO$_4$ as positive electrode and $dt$-$Nb_{12}WO_{33}$ as negative electrode materials, and tested within the voltage range of 1.0–2.5 V (Supplementary Fig. 24). Considering practical conditions, the negative electrode material was assembled into full cells without undergoing pre-lithiation or activation. The average output voltage of $dt$-$Nb_{12}WO_{33}$|| LFP is $ca.$ 1.76 V, and it delivers a specific capacity of 204.8 mAh g$^{-1}$, based on the mass of active negative electrode material at 0.5 C. At different specific current rates from 1 C to 20 C, $dt$-$Nb_{12}WO_{33}$||LFP shows discharge capacities of 186.3, 158.6, 124.5, 97.8, and 71.8 mAh g$^{-1}$. $dt$-$Nb_{12}WO_{33}$||LFP maintains a reversible capacity of 108 mAh g$^{-1}$ at 5 C after 500 cycles, corresponding to a retention of 93.5%. Its good rate capability and long cycling life indicate the strong potential of $dt$-$Nb_{12}WO_{33}$ for practical applications.

Considering that $dt$-$Nb_{12}WO_{33}$ consists of smaller particles than $m$-$Nb_{12}WO_{33}$, we have additionally prepared and studied the electrochemical performance of $m$-$Nb_{12}WO_{33}$ with smaller particles (100–400 nm) and $dt$-$Nb_{12}WO_{33}$ with larger particles (1.0–1.3 μm) to evaluate the effect of the particle size on the performance. $m$-$Nb_{12}WO_{33}$ with a particle size similar to that of $dt$-$Nb_{12}WO_{33}$ was produced by ball milling (BM) of the initial $m$-$Nb_{12}WO_{33}$ material; the ball-milled sample is denoted $m$-$Nb_{12}WO_{33}$-BM. $dt$-$Nb_{12}WO_{33}$ dense microspheres (MS) with sizes 1.0–1.3 μm were synthesized via solvothermal method and subsequent calcination at 900 °C under air;

this sample is denoted $dt$-$Nb_{12}WO_{33}$-MS. The XRD patterns of $m$-$Nb_{12}WO_{33}$-BM and $dt$-$Nb_{12}WO_{33}$-MS are similar to those of $m$-$Nb_{12}WO_{33}$ and $dt$-$Nb_{12}WO_{33}$, respectively, except for a slight broadening of the peaks for $m$-$Nb_{12}WO_{33}$-BM, caused by the decrease of the particle size, and a slight narrowing of the peaks for $dt$-$Nb_{12}WO_{33}$-MS, caused by the increase of the particle size (Supplementary Fig. 25a, b). The Raman spectra of $m$-$Nb_{12}WO_{33}$-BM and $dt$-$Nb_{12}WO_{33}$-MS are also identical to those of their $m$-$Nb_{12}WO_{33}$ and $dt$-$Nb_{12}WO_{33}$ counterparts (Supplementary Fig. 25c, d). SEM and TEM images (Supplementary Fig. 26) show that the particle size of $m$-$Nb_{12}WO_{33}$-BM ranges from 100 to 400 nm, while those of $dt$-$Nb_{12}WO_{33}$ microspheres are in the range of 1–1.3 μm. The HR-TEM images of $m$-$Nb_{12}WO_{33}$-BM and $dt$-$Nb_{12}WO_{33}$-MS show lattice fringes with an interplanar spacing of 0.49 nm and 0.28 nm, consistent with the d-spacing of the (003) and (101) planes of $m$-$Nb_{12}WO_{33}$-BM and $dt$-$Nb_{12}WO_{33}$-MS, respectively. Additionally, the (71-1), (−407), and (316) planes of the monoclinic structure are observed in the SAED pattern of $m$-$Nb_{12}WO_{33}$-BM. The tetragonal structure of $dt$-$Nb_{12}WO_{33}$-MS is further confirmed through the diffraction associated with the (460), (−170), and (5−10) planes observed in the SAED pattern. The electrochemical properties of $m$-$Nb_{12}WO_{33}$-BM and $dt$-$Nb_{12}WO_{33}$-MS were investigated using half-cells. The initial Coulombic efficiency of $m$-$Nb_{12}WO_{33}$-BM is 86.1%, which is close to that of $m$-$Nb_{12}WO_{33}$ (87.3%), while $dt$-$Nb_{12}WO_{33}$-MS exhibits an initial Coulombic efficiency of 94.4%, similar to the 95.3% for $dt$-$Nb_{12}WO_{33}$ (Supplementary Fig. 27a). $m$-$Nb_{12}WO_{33}$-BM exhibits specific capacities of 56.8 and 29.2 mAh g$^{-1}$ at 40 and 80 C, respectively, which are comparable to the 49.2 and 30.9 mAh g$^{-1}$ delivered by $m$-$Nb_{12}WO_{33}$ at the same rates (Supplementary Fig. 27b, c). Therefore, decreasing the particle size of $m$-$Nb_{12}WO_{33}$ to sizes similar to those of the $dt$-$Nb_{12}WO_{33}$ does not improve the rate capability of the material. Moreover, $dt$-$Nb_{12}WO_{33}$-MS delivers a high capacity of 108.8 mAh g$^{-1}$ at 80 C, which is comparable to that of $dt$-$Nb_{12}WO_{33}$ and much higher than that of $m$-$Nb_{12}WO_{33}$ and $m$-$Nb_{12}WO_{33}$-BM (Supplementary Fig. 27d). The Li$^+$ diffusion coefficients in $m$-$Nb_{12}WO_{33}$-BM and $dt$-$Nb_{12}WO_{33}$-MS were investigated through GITT (Supplementary Fig. 27e, f). The $D_{Li+}$ of $dt$-$Nb_{12}WO_{33}$-MS and $m$-$Nb_{12}WO_{33}$-BM are in the ranges $1.4 \times 10^{-10}$–$1.1 \times 10^{-11}$ and $3.8 \times 10^{-11}$–$3.3 \times 10^{-13}$, respectively. These values are comparable to those of $dt$-$Nb_{12}WO_{33}$ and $m$-$Nb_{12}WO_{33}$, respectively. The $D_{Li+}$ of $dt$-$Nb_{12}WO_{33}$-MS continues to be larger than those of $m$-$Nb_{12}WO_{33}$-BM, despite the larger particles of the former sample. These results show that changing the size of the particles of the materials between 100 and 400 nm and ca. 1 μm does not significantly affect the lithium diffusion coefficients or rate capability of the materials, and therefore, the differences in the electrochemical performance of the $dt$-$Nb_{12}WO_{33}$ and $m$-$Nb_{12}WO_{33}$ are caused primarily by the differences in their structures.

## Structure evolution and reaction mechanism

To probe the long-range structure evolution of $m$-$Nb_{12}WO_{33}$ and $dt$-$Nb_{12}WO_{33}$ upon Li$^+$ insertion and extraction, operando XRD was performed within the voltage window of 1.0–3.0 V at 0.1 C. The diffractogram of the pristine $dt$-$Nb_{12}WO_{33}$ (Fig. 4b) exhibits reflections associated with the planes (101), (440), (431), (800), (701), (770), and (002). Three distinct regions (referred to as $T1$−$T3$) are observed for $dt$-$Nb_{12}WO_{33}$ during the initial discharging process. Two solid-solution reactions occur in the sloped regions, from the opening circuit voltage (OCV) to 1.70 V ($T1$) and from 1.65 to 1.0 V ($T3$), and a biphasic reaction occurs within the flat region ($T2$). In contrast, $m$-$Nb_{12}WO_{33}$ (Fig. 4a and Supplementary Fig. 28) exhibits two biphasic reactions during the initial lithiation process. The (020) diffraction peak of $m$-$Nb_{12}WO_{33}$ ($2\theta$-21.4°) disappears at $ca.$ 1.7 V during the initial discharge process. Simultaneously, a new peak emerges at a lower angle ($2\theta$-21.2°) within the $M2$ region, indicating a phase transition occurring in the plateau region. The peak shifts to a lower angle ($M3$) due to the expansion of the lattice along the $b$ direction, and the intensity of the peak begins to

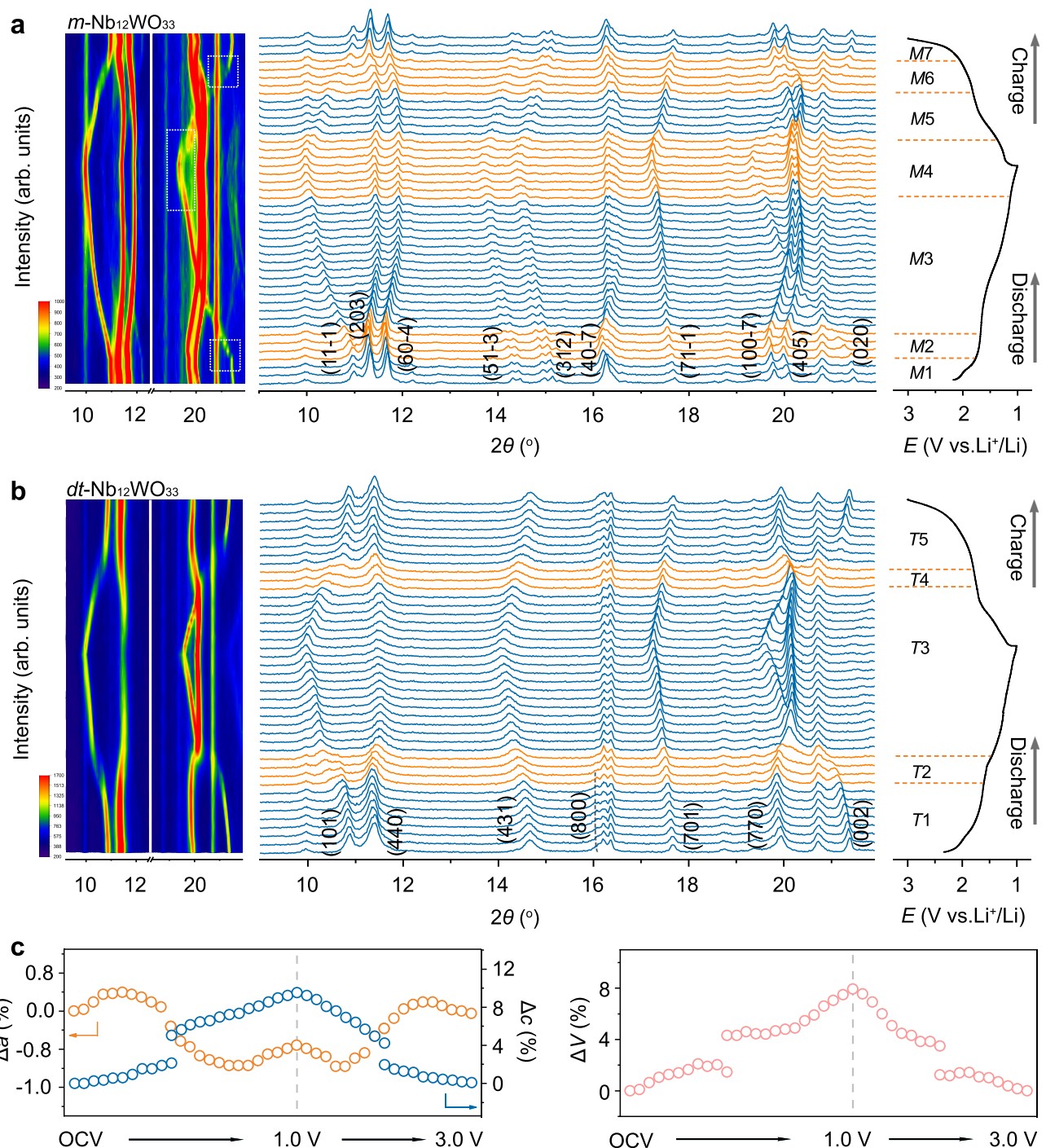

**Fig. 4 | Long-range structure evolution of *m*-Nb₁₂WO₃₃ and *dt*-Nb₁₂WO₃₃ during lithiation and delithiation.** *Operando* XRD patterns of (**a**) *m*-Nb₁₂WO₃₃ and (**b**) *dt*-Nb₁₂WO₃₃ electrodes, collected for the first discharge and charge processes. The corresponding XRD contour plots at selected angle ranges are shown on the left. **c** Unit cell parameters evolution during Li⁺ insertion and extraction for the *dt*-Nb₁₂WO₃₃ electrode.

decrease when the electrode is discharged to ca. 1.3 V. Within the *M*4 region, a new peak arises on the left side of the pattern (2*θ*~19.3°), indicative of the biphasic reactions during the final stage of the discharge process.

The evolution of the lattice parameters during the initial lithiation and delithiation processes of *dt*-Nb₁₂WO₃₃ is plotted in Fig. 4c. The reflection at 2*θ*≈16.4° is attributed to the lithium metal (Supplementary Fig. 29), which was used as a counter electrode in the *operando* cell. Upon discharging, the (101) reflection shifts continuously to lower angles, and the lattice parameter *c* increases monotonically until the

fully discharged state, indicating that Li⁺ ions are inserted into the crystal lattice along the *c* axis, perpendicular to the block plane, and columbic repulsion between Li⁺ increases. The positions of the (440) and (770) reflections shift to lower angles from the initial state to Li₃.₈Nb₁₂WO₃₃, as a result of the expansion of the *a*–*b* plane by 0.8%. The (800) reflection (2*θ*≈16.1°) shifts slightly to lower angles due to the expansion of the *a*-axis. Subsequently, the (440) and (770) reflections shift to higher angles from Li₃.₈Nb₁₂WO₃₃ to Li₁₃.₄Nb₁₂WO₃₃ due to a contraction of the *a*–*b* plane by 3.1%. Finally, the *a*–*b* plane experiences another expansion (0.9%) within the voltage region below 1.3 V, from

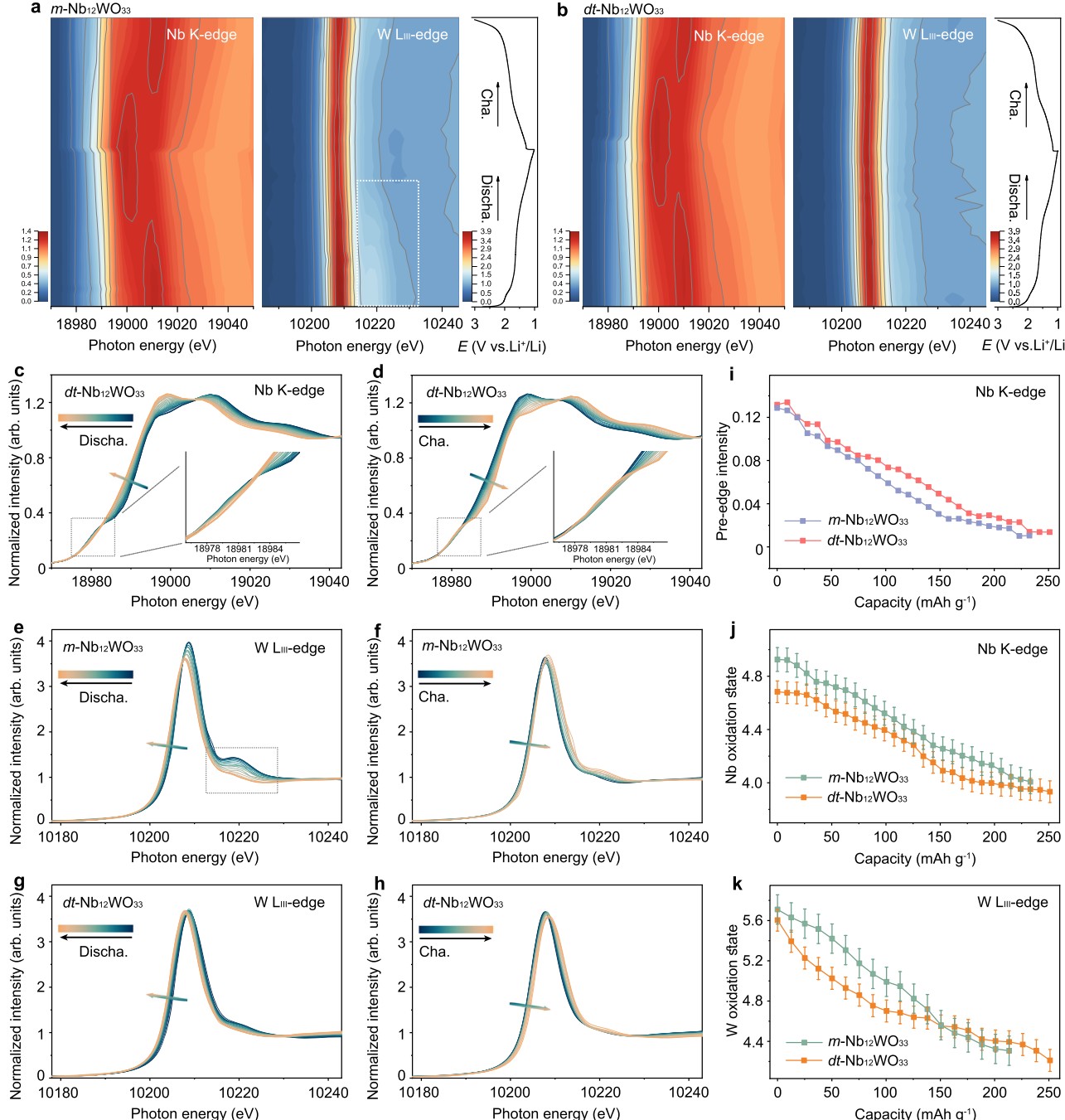

**Fig. 5 | *Operando* Nb K-edge and W L$_{III}$-edge XANES characterization during lithiation and delithiation.** *Operando* Nb K-edge and W L$_{III}$-edge XANES contour plots of (**a**) *m*-Nb$_{12}$WO$_{33}$ and (**b**) *dt*-Nb$_{12}$WO$_{33}$ (the corresponding voltage profiles for the first cycle are shown on the right side). *Operando* Nb K-edge XANES spectra of *dt*-Nb$_{12}$WO$_{33}$ for the (**c**) discharge and (**d**) charge processes. *Operando* W L$_{III}$-edge XANES spectra of *m*-Nb$_{12}$WO$_{33}$ for the (**e**) discharge and (**f**) charge processes.

*Operando* W L$_{III}$-edge XANES spectra of *dt*-Nb$_{12}$WO$_{33}$ for the (**g**) discharge and (**h**) charge processes. **i** Pre-edge integrated peak intensity from the Nb K-edge XANES spectra of *m*-Nb$_{12}$WO$_{33}$ and *dt*-Nb$_{12}$WO$_{33}$. Oxidation states of (**j**) Nb and (**k**) W as a function of *x* in Li$_x$Nb$_{12}$WO$_{33}$ during lithiation. Oxidation state error bars are estimated to be ±0.3 as a function of energy data resolution.

Li$_{13.4}$Nb$_{12}$WO$_{33}$ to Li$_{17.6}$Nb$_{12}$WO$_{33}$. The evolutions of the unit cell lattice parameters and volume are highly reversible during the following Li$^+$ extraction process. While the total lithiation process leads to a 9.5% expansion along the *c* axis, a contraction of 1.4% occurs along the *a* and *b* axes. The anisotropic evolution of the lattice parameters results in a volume change of 8.0% for *dt*-Nb$_{12}$WO$_{33}$, which is slight smaller than the 8.3% for *m*-Nb$_{12}$WO$_{33}$ (Supplementary Fig. 30). HAADF-STEM images of lithiated *dt*-Nb$_{12}$WO$_{33}$ and *m*-Nb$_{12}$WO$_{33}$ were taken to determine potential changes in the block structure caused by lithium insertion

(Supplementary Figs. 31 and 32, respectively). The images of lithiated *dt*-Nb$_{12}$WO$_{33}$ show similar disorder to that of the pristine material, with blocks of different sizes, including 5 × 4, 4 × 4, and 4 × 3. The atomic arrangement of niobium and tungsten observed in the images of lithiated *m*-Nb$_{12}$WO$_{33}$ matches the monoclinic structure along the *a*-axis, and indicates the retention of the block structure. Ex-situ Raman spectroscopy experiments were carried out to validate the Li-ion coordination behavior during lithiation/delithiation in *dt*-Nb$_{12}$WO$_{33}$ (Supplementary Fig. 33). A weak band at 330 cm$^{-1}$ appears during

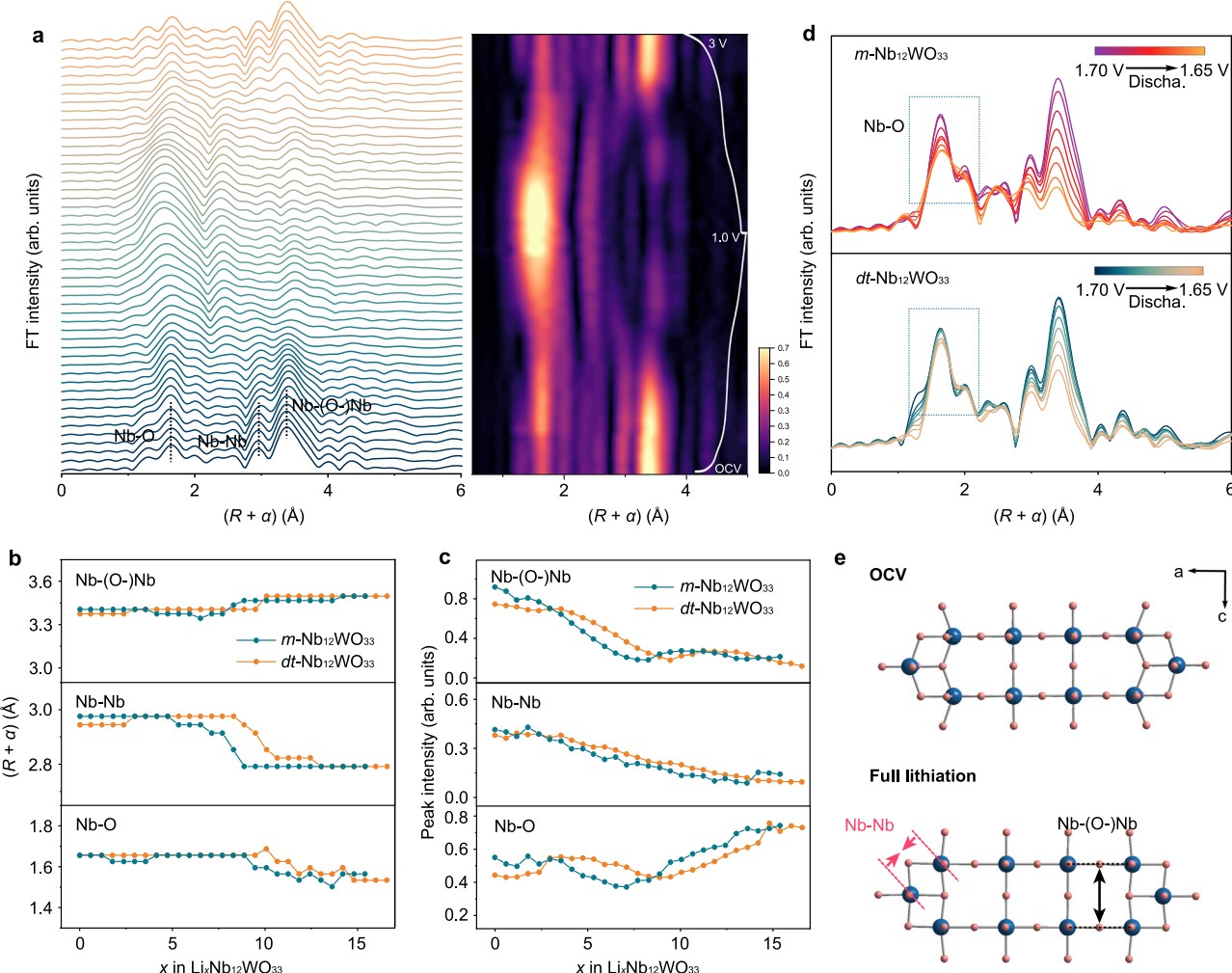

**Fig. 6 | *Operando* Nb K-edge EXAFS characterization during lithiation and delithiation. a** *Operando* Nb K-edge EXAFS spectra and contour plot of *dt*-$Nb_{12}WO_{33}$ (the corresponding voltage profile for the first cycle is shown on the right side). **b** and (**c**) Variation of the radial distance and peak intensity of the *operando* Nb K-edge EXAFS spectra as a function of $x$ in $Li_xNb_{12}WO_{33}$ during lithiation. **d** Nb K-edge *operando* EXAFS spectra during the two-phase reaction within the voltage range 1.65–1.70 V. **e** Illustration of the structure of *dt*-$Nb_{12}WO_{33}$ in the OCV and fully lithiated states; blue and pink spheres represent Nb and O atoms, respectively.

lithiation, corresponding to the Li–O–Nb(W) bending mode[6]. The Raman spectrum recovered after the full charging process further indicates that the structural evolution during lithiation/delithiation is highly reversible.

Operando Nb K-edge and W $L_{III}$-edge XANES and EXAFS data of *m*-$Nb_{12}WO_{33}$ and *dt*-$Nb_{12}WO_{33}$ were used to get insights into the charge transfer processes involving niobium and tungsten, as well as the evolution of the local structure around the metal atoms during discharge and charge (Figs. 5–7). In the XANES contour plots for the Nb K-edge and W $L_{III}$-edge of *m*-$Nb_{12}WO_{33}$ and *dt*-$Nb_{12}WO_{33}$ cycled at 0.15 C (Fig. 5a, b), the absorption edge exhibits a reversible energy shift during the initial cycle, meaning that both metals participate in the charge compensation process. The position of the edges shifts to lower energies as the metals are reduced during discharge (Fig. 5 and Supplementary Fig. 34). The oxidation states of Nb and W were determined by the half-height method, which has been used as a relatively accurate method to calculate the valence states[33,46]. Note that the initial oxidation states of Nb and W are lower than 5+ and 6+, respectively, due to self-discharge occurring during the waiting period before the measurements. To demonstrate this self-discharge effect, we show the voltage profiles of *m*-$Nb_{12}WO_{33}$ and *dt*-$Nb_{12}WO_{33}$ and images of the respective electrode materials as a function of time (Supplementary Fig. 35). The voltage decreases slowly with time from

OCV until ca. 15 h, remaining essentially unchanged after that. To study the color change during the waiting time, the *m*-$Nb_{12}WO_{33}$ and *dt*-$Nb_{12}WO_{33}$ powders were used as positive electrodes in half-cells. The color of the powders changes from white to light blue during this period, a color that is typical of partially reduced tungsten in tungsten-containing oxides. The oxidation states of niobium and tungsten decrease down to 3.8+ and 4.2+, respectively, for *dt*-$Nb_{12}WO_{33}$ in the fully discharged state. These values indicate a charge transfer of 16.2 electrons per formula unit. The corresponding values are 3.9+ and 4.3+, respectively, for *m*-$Nb_{12}WO_{33}$, resulting in 14.9 electrons per formula unit and thus a smaller contribution from charge compensation and smaller capacity. In addition, the oxidation states of Nb and W in *m*-$Nb_{12}WO_{33}$ show a nearly linear variation with the amount of $Li^+$ inserted into the structure (Fig. 5j, k). A more accentuated decrease in the oxidation state of tungsten is observed for *dt*-$Nb_{12}WO_{33}$ up to ca. $Li_8Nb_{12}WO_{33}$, suggesting that in this material, the tungsten is preferentially reduced up to a content of 8 $Li^+$ per formula unit, thus having a larger contribution to the charge compensation within this range. A decrease of the pre-edge intensity in the Nb K-edge XANES spectra of both materials occurs during lithiation (Fig. 5c, i and Supplementary Fig. 34). This reflects an increase of the symmetry in the octahedral environment around the Nb atoms, i.e., a gradual decrease of the octahedra distortion, in agreement with experimental results

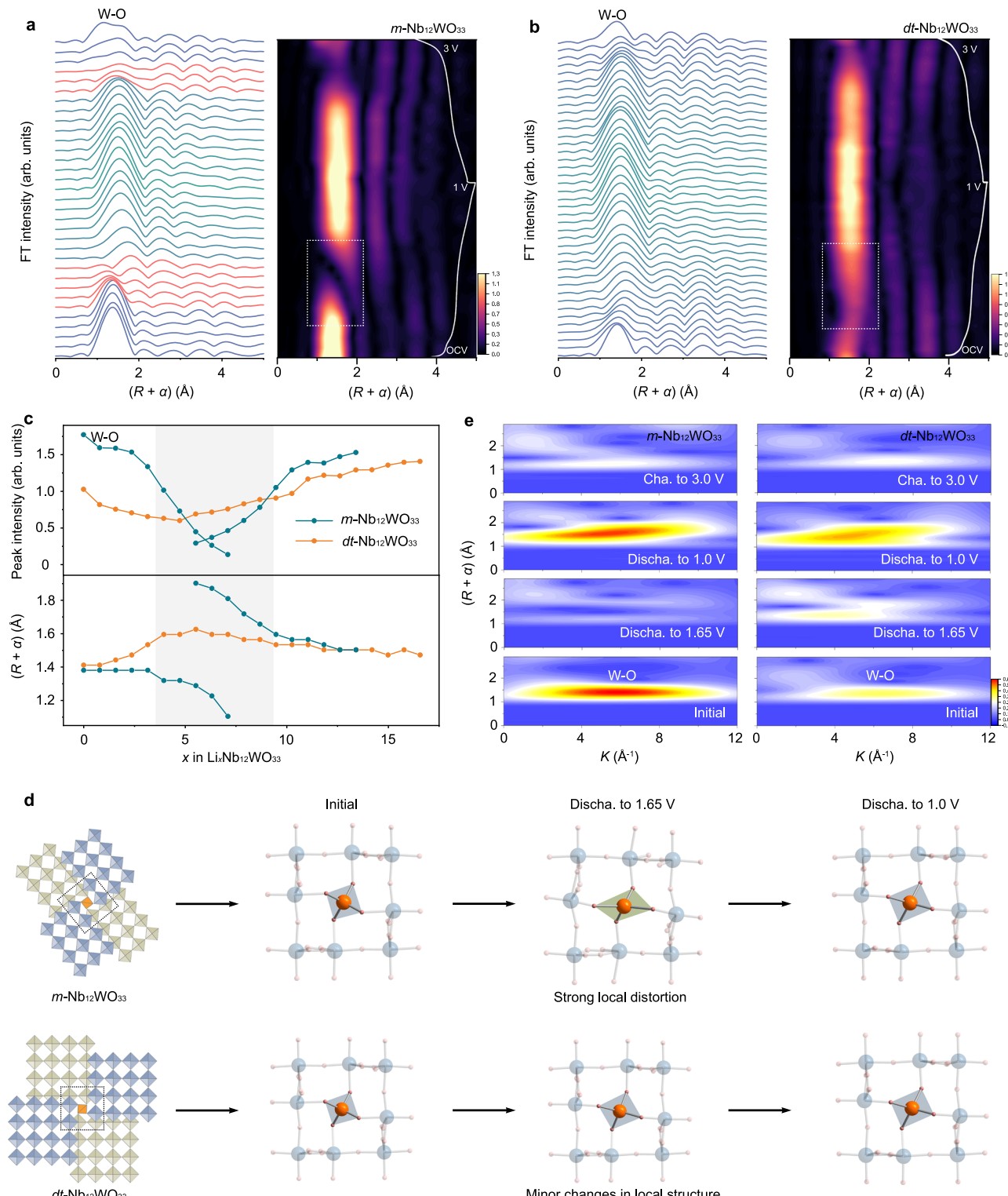

**Fig. 7 | *Operando* W L$_{III}$-edge EXAFS characterization during lithiation and delithiation.** *Operando* W L$_{III}$-edge EXAFS spectra and contour plots for (**a**) *m*-Nb$_{12}$WO$_{33}$ and (**b**) *dt*-Nb$_{12}$WO$_{33}$ (the corresponding voltage profile for the first cycle is shown on the right side). **c** Variation of the radial distance and peak intensity of the *operando* W L$_{III}$-edge EXAFS spectra as a function of $x$ in Li$_x$Nb$_{12}$WO$_{33}$ during lithiation. **d** Schematic representation of the W tetrahedral sites structural evolution for *m*-Nb$_{12}$WO$_{33}$ and *dt*-Nb$_{12}$WO$_{33}$ at different discharged states; blue and green squares represent the octahedral sites, while orange squares denote tetrahedral sites; blue, orange, and pink spheres represent Nb, W, and O atoms, respectively. **e** WT-EXAFS spectra of *m*-Nb$_{12}$WO$_{33}$ (left side) and *dt*-Nb$_{12}$WO$_{33}$ (right side) at different discharged states (the blue-yellow-red color gradient represents the increase of the EXAFS intensity).

and DFT calculations reported for other niobium tungsten oxide block structures[5,10,33]. It is attributed partially to the increase of the energy of the d states during lithiation, as the metal is reduced and its electron configuration deviates from $d^0$, mitigating the second-order

Jahn−Teller effect. The second derivative of the W L$_{III}$-edge XANES spectra remains similar to those of the corresponding pristine materials, showing only one peak, suggesting that tungsten remains mainly in tetrahedral coordination in *m*-Nb$_{12}$WO$_{33}$ and *dt*-Nb$_{12}$WO$_{33}$ during

discharge (Supplementary Figs. 36 and 37). However, there are clear differences in the evolution of the spectra of the materials (Fig. 5e–h). For $m$-Nb$_{12}$WO$_{33}$, the intensity of the edge diminishes during lithiation, and the intensity of the post-edge feature also decreases. This can be attributed to the WO$_4$ and WO$_6$ polyhedra getting progressively more distorted as the discharge process advances. As a result of the distortion around the W atoms, a decrease in the overlapping between W $d$-orbitals and O $p$-orbitals can occur, decreasing the electron density at the metal center. The changes in the spectra are only partially reversible during the charge process. On the contrary, the intensity of the white line in the spectra of $dt$-Nb$_{12}$WO$_{33}$ does not change significantly compared to $m$-Nb$_{12}$WO$_{33}$, suggesting less important changes. Less electron density on the tungsten in $dt$-Nb$_{12}$WO$_{33}$ due to distortion could contribute to the more accentuated decrease of the oxidation state of tungsten in $dt$-Nb$_{12}$WO$_{33}$ compared to $m$-Nb$_{12}$WO$_{33}$ at the initial stages of discharge (Fig. 5k).

The evolution of the local structure and coordination environment around the Nb and W atoms during lithiation and delithiation was evaluated through *operando* $k^2$-weight EXAFS spectra (Supplementary Figs. 38 and 39). The Fourier transform (FT) Nb K-edge EXAFS spectra of $m$-Nb$_{12}$WO$_{33}$ and $dt$-Nb$_{12}$WO$_{33}$ (Fig. 6a and Supplementary Fig. 40) show peaks at 1.6 Å, 2.9 Å, and 3.4 Å corresponding to the interatomic distances between Nb–O, Nb–Nb, and Nb–(O)–Nb, respectively. It should be noted that there is a difference of ca. 0.4 Å between the measured and actual atomic distances due to phase shifts during the scattering process[47,48]. The evolution of the radial distance and intensity of the peaks associated with the first and second coordination shells of Nb as a function of the lithium content (Fig. 6b, c, respectively) reveals that the amplitude for Nb–O in $m$-Nb$_{12}$WO$_{33}$ remains constant from the OCV state to Li$_{3.0}$Nb$_{12}$WO$_{33}$. However, it subsequently undergoes a dramatic decrease from Li$_{3.0}$Nb$_{12}$WO$_{33}$ to Li$_{7.1}$Nb$_{12}$WO$_{33}$, which is associated with the two-phase transition reaction. In contrast, $dt$-Nb$_{12}$WO$_{33}$ experiences a more stable evolution within the same range. Figure 6d compares selected *operando* EXAFS spectra of the materials during the two-phase reaction, within the voltage range 1.65–1.70 V. Less significant changes are observed in the first peak for $dt$-Nb$_{12}$WO$_{33}$ compared to that of $m$-Nb$_{12}$WO$_{33}$, meaning that much smaller structural changes occur in the first oxygen coordination shell around the niobium in the former, which reflects the robustness of the partially disordered phase. This robustness ensures less disruption of the structure during lithiation, providing stable pathways for Li$^+$ ions diffusion. The magnitude of the Nb–Nb and Nb–(O)–Nb peaks decreases continuously, which can be attributed to the shielding effect of Nb–Nb corner-sharing interactions during lithium insertion into the structure[33]. The Nb–O radial distance in $dt$-Nb$_{12}$WO$_{33}$ shows a decreasing trend from Li$_{10}$Nb$_{12}$WO$_{33}$, possibly as a result of the increasing symmetry of the NbO$_6$ octahedra. The Nb–Nb radial distance in $dt$-Nb$_{12}$WO$_{33}$ contracts by 0.18 Å along the shear planes, from Li$_{8.3}$Nb$_{12}$WO$_{33}$ to the discharged state, causing the neighboring blocks to slide closer and resulting in the lattice contraction. The Nb–(O)–Nb radial distance begins to increase from Li$_{10.7}$Nb$_{12}$WO$_{33}$, caused by an expansion perpendicular to the shear plane. The weak peak observed in the *operando* Nb K-edge EXAFS spectra at a radial distance of 3.1 Å, from Li$_{10.7}$Nb$_{12}$WO$_{33}$ during the discharge process, may be a contribution from the Nb–W1 radial distance. This peak becomes noticeable due to a shift in the Nb–Nb radial distance to smaller values, coupled with a decrease in intensity and a shift to higher values of the Nb–(O)–Nb peak. The local structure evolution within the shear blocks is consistent with the anisotropic evolution of the lattice parameters found via *operando* XRD, and is illustrated for $dt$-Nb$_{12}$WO$_{33}$ in Fig. 6e.

A similar qualitative analysis was performed for the *operando* W L$_{III}$-edge FT-EXAFS spectra of $m$-Nb$_{12}$WO$_{33}$ and $dt$-Nb$_{12}$WO$_{33}$ during lithiation/delithiation (Fig. 7a–c). The W–O radial distance in the polyhedra of $m$-Nb$_{12}$WO$_{33}$ is 1.4 Å and remains constant during discharge from the OCV state to Li$_{3.2}$Nb$_{12}$WO$_{33}$, accompanied by a decrease of the intensity (Fig. 7a). The radial distance subsequently decreases gradually from Li$_{3.2}$Nb$_{12}$WO$_{33}$ to Li$_{7.1}$Nb$_{12}$WO$_{33}$, reaching a value of 1.1 Å, while the intensity sharply declines until the peak completely disappears from Li$_{2.4}$Nb$_{12}$WO$_{33}$ to Li$_{7.1}$Nb$_{12}$WO$_{33}$ during the two-phase transition reaction. Another peak at a higher radial distance of 2.0 Å starts to appear from Li$_{5.5}$Nb$_{12}$WO$_{33}$. Therefore, two types of W–O bond lengths, differing approximately 0.6 Å, with corresponding low amplitude peaks, seem to exist during the two-phase transition reaction within the voltage range 1.65–1.70 V, suggesting that the local tetrahedral environment around the tungsten experiences significant anisotropic distortion, with a markedly asymmetric distribution of the bond lengths. In contrast, the radial distance of the W–O peak in the $dt$-Nb$_{12}$WO$_{33}$ spectra gradually increases from the OCV state to Li$_{3.0}$Nb$_{12}$WO$_{33}$, and then remains constant, while a gradual increase of the peak intensity takes place from Li$_{3.0}$Nb$_{12}$WO$_{33}$ to Li$_{3.0}$Nb$_{12}$WO$_{33}$ during the two-phase transition reaction (Fig. 7b and Supplementary Fig. 41). This result reveals less significant changes in the local coordination environment around the tungsten atoms, which are mainly in tetrahedral sites, arising from the partially disordered structural arrangement present in $dt$-Nb$_{12}$WO$_{33}$, which minimizes local structural disruption during lithiation, and thus contributes to a stable intercalation structure (Fig. 7d). The amplitude of the W–O peak in the $dt$-Nb$_{12}$WO$_{33}$ spectra slowly increases upon further discharge to 1.0 V, in contrast with the fast increase observed for $m$-Nb$_{12}$WO$_{33}$. The W–O radial distance is larger in the discharged compared to the OCV state for both materials, as a result of the changes in the electron configuration and size of the reduced tungsten ions. A broad and low-intensity W–O peak is observed in the $m$-Nb$_{12}$WO$_{33}$ spectra during the two-phase reaction region of the delithiation process. This strong local disorder results in an irreversible change of the W–O radial distance for the fully charged state, which may account for the low initial columbic efficiency observed for $m$-Nb$_{12}$WO$_{33}$. However, the W–O radial distance of the fully charged $dt$-Nb$_{12}$WO$_{33}$ is the same as in the OCV state, indicating a good reversibility of the local structure during the Li$^+$ ions insertion and extraction.

To further investigate the evolution of the W coordination environment, wavelet transform (WT) analysis of the W L$_{III}$-edge EXAFS spectra was carried out (Fig. 7e). The scattering peaks at (5.8 Å$^{-1}$, 1.38 Å) and (6.5 Å$^{-1}$, 1.38 Å) are attributed to the contributions of W–O of $m$-Nb$_{12}$WO$_{33}$ and $dt$-Nb$_{12}$WO$_{33}$, respectively, and the peak intensity is related to the evolution of the local coordination environment. During lithiation, the intensity of the W–O peak of $m$-Nb$_{12}$WO$_{33}$ sharply decreases after discharging to 1.6 V, while it only slightly decreases for $dt$-Nb$_{12}$WO$_{33}$. In the following charging process, the intensity is only partially recovered for $m$-Nb$_{12}$WO$_{33}$, while it almost completely recovers to that of the OCV state in the case of $dt$-Nb$_{12}$WO$_{33}$. These results further confirm the relatively minor alterations of the coordination environment around the tungsten atoms in $dt$-Nb$_{12}$WO$_{33}$ during the electrochemical process and the reversibility of those alterations.

To evaluate the degradation mechanisms during long-term cycling, SEM, XRD, and XAS analyses of the materials were performed after 500 cycles. SEM images of $m$-Nb$_{12}$WO$_{33}$ and $dt$-Nb$_{12}$WO$_{33}$ (Supplementary Fig. 42) show the absence of cracks or holes on the surface of the materials. Comparison between the X-ray diffractograms of the materials before and after cycling does not indicate the irreversible formation of other crystalline phases (Supplementary Fig. 43a, b) and suggests that the long-range structure is maintained after cycling. The Nb K-edge XANES spectra of $m$-Nb$_{12}$WO$_{33}$ and $dt$-Nb$_{12}$WO$_{33}$ before and after cycling almost overlap (Supplementary Fig. 43c). A similar result is obtained with the W L$_{III}$-edge XANES spectra (Supplementary Fig. S43d), which suggests reversibility of the redox reactions for both materials. Supplementary Fig. 43e compares the Nb

K-edge EXAFS spectra of the initial and cycled $m$-$Nb_{12}WO_{33}$ and $dt$-$Nb_{12}WO_{33}$. In the case of $m$-$Nb_{12}WO_{33}$, there is a shift of the Nb–O peak after cycling from 1.50 to 1.54 Å, accompanied by a decrease of the peak intensity, whereas only a slight decrease in the intensity of the Nb–O peak is observed for $dt$-$Nb_{12}WO_{33}$, without significant peak shift. As for the W $L_{III}$-edge EXAFS spectra (Supplementary Fig. 43f), a more accentuated decrease in the W–O peak intensity is observed for $m$-$Nb_{12}WO_{33}$ after cycling than for $dt$-$Nb_{12}WO_{33}$. The results reveal that irreversible local distortions occur within the $NbO_6$ and $WO_4$ polyhedra in both materials after repeated lithiation and delithiation processes, although the effect is smaller for $dt$-$Nb_{12}WO_{33}$. Therefore, irreversible local distortions appear to be the primary degradation mechanism during long-term cycling in $m$-$Nb_{12}WO_{33}$ and $dt$-$Nb_{12}WO_{33}$, leading to a decrease in capacity.

The data suggests that the partial disorder of the $dt$-$Nb_{12}WO_{33}$ structure, introduced by the variability of the $ReO_3$-type blocks' dimensions and consequent disorder of the tetrahedral tungsten sites position, as well as distortion of those tetrahedra, leads to a robust structure that provides suitable and stable pathways for $Li^+$ ions diffusion. As a result, the partial disorder of the structure allows for $Li^+$ fast diffusion and therefore better rate capability than $m$-$Nb_{12}WO_{33}$, together with higher stability.

To assess the economic viability of $dt$-$Nb_{12}WO_{33}$, particularly the potential cost increase of introducing tungsten into a niobium oxide, we have considered the price of the metals and their estimated availability on the planet. The prices of niobium and tungsten are affected by supply constraints, geopolitical factors, and processing technology. According to a sustainability evaluation carried out by Tkaczyk et al., the price of tungsten fluctuated after 2005, with peaks in 2006 (37 USD/kg) and 2012 (57 USD/kg), and then decreased to 35 USD/kg in 2017[49]. As for niobium, the price remained stable at around 40–50 USD/kg since 2010, and was 50 USD/kg in 2017. A similar trend is found for the prices of niobium and tungsten precursors with identical characteristics (e.g., ligand and purity) sold by the main chemical reagent suppliers in Germany. Theoretically, the inclusion of W in Nb oxide compounds does not lead to higher material costs while simultaneously enhancing the electrochemical performance. On the other hand, the tungsten reserves on the planet are smaller than those of niobium, with the tungsten content in the earth crust estimated as 1.25 mg $kg^{-1}$ compared to 20 mg $kg^{-1}$ for niobium. However, the atomic percentage of tungsten in $dt$-$Nb_{12}WO_{33}$ is only 8 %, while the performance gains from the introduction of tungsten are significant. For example, $dt$-$Nb_{12}WO_{33}$ delivers a high capacity of 119.5 mAh $g^{-1}$ at 15.3 A $g^{-1}$, compared to 94 mAh $g^{-1}$ at 4 A $g^{-1}$ for H-$Nb_2O_5$ and 60 mAh $g^{-1}$ at 0.132 A $g^{-1}$ for $Nb_{12}O_{29}$[43,50]. Therefore, considering the current prices of tungsten and niobium, the incorporation of tungsten seems economically advantageous, especially taking into account the potential use of the materials in high-end battery applications such as electric vehicles, where high power density makes it more efficient and user-friendly by reducing charging time.

A partially disordered $Nb_{12}WO_{33}$ phase ($dt$-$Nb_{12}WO_{33}$) has been obtained, which consists of $ReO_3$-type blocks of varied sizes with more spacious channels for ion diffusion, irregular arrangement of the tetrahedral sites, and strong distortion of those sites, for application as a negative electrode material in Li-ion batteries. The unique structure results in wider pathways for $Li^+$ ions diffusion and minor changes in the local structure during discharge and charge compared to the monoclinic structure ($m$-$Nb_{12}WO_{33}$), providing suitable and stable pathways for faster $Li^+$ diffusion rates. The performance of $dt$-$Nb_{12}WO_{33}$ is significantly enhanced with respect to that of $m$-$Nb_{12}WO_{33}$, showing a 44.7% capacity retention at 80 C and a high initial columbic efficiency of 95.3%. The strategy reported here of introducing disorder in the cation arrangements of block oxides is promising for the future design of electrode materials for lithium-ion batteries for fast-charging applications.

## Methods
### Material synthesis
To synthesize the $Nb_{12}WO_{33}$ precursors, ammonium niobate oxalate (99.99%, Sigma−Aldrich) and ammonium tungstate hydrate (99.9%-W, Sigma−Aldrich) were dissolved in 16 mL of tert-butanol (ACS reagent, ≥99.7%, Sigma−Aldrich), and the solution was transferred to a Teflon-lined stainless-steel autoclave, and then heated at 170 °C for 5 days. The obtained product was collected by centrifugation, washed with absolute ethanol, and then it was dried at 60 °C overnight. Subsequently, the $Nb_{12}WO_{33}$ precursors were heated in air atmosphere at 900 °C and 1100 °C for 10 h, respectively. $m$-$Nb_{12}WO_{33}$-BM was prepared by ball milling of $m$-$Nb_{12}WO_{33}$ using 0.3 cm zirconia balls in a swing ball mill with a frequency of 20 Hz in a 10 mL stainless-steel jar (SBM, Retsch MM400). The material was ball milled with a ball-to-powder ratio of 25:1 for 90 min. To synthesize the $dt$-$Nb_{12}WO_{33}$ microspheres ($dt$-$Nb_{12}WO_{33}$-MS), stoichiometric amounts of niobium chloride (99.99%-Nb, ABCR) and tungsten chloride (≥99.9%, ABCR) were dissolved in 16 mL of ethanol (99.8%, Sigma−Aldrich), and the solution was transferred to a Teflon-lined stainless-steel autoclave, and then heated at 180 °C for 2 days. The product was collected by centrifugation and washed with absolute ethanol, and subsequently heated in air at 900 °C for 10 h. Larger crystals of $m$-$Nb_{12}WO_{33}$ were synthesized via a molten flux method[51,52]. Stoichiometric amounts of $Nb_2O_5$ (0.5 mmol, 99.5%, ThermoFisher GmbH) and $WO_3$ (0.083 mmol, Sigma−Aldrich) were mixed by manually grinding, and subsequently $H_3BO_3$ (5 mmol, 99.99%-B, ABCR) was added and mixed with the oxide precursors. The powder mixture was placed in an alumina crucible, heated at 1050 °C for 10 h in a muffle oven, with a heating rate of 150 °C $h^{-1}$ to reach the final temperature. After cooling to room temperature with a cooling rate of 60 °C $h^{-1}$, the powder was washed with water to remove the remaining $H_3BO_3$.

### Electrochemical measurements
Electrochemical tests were carried out in a coin cell (stainless steel, CR2032, Shandong Gelon Lib Co., Ltd), which was assembled in an Ar-filled glove box. The electrode was prepared by mixing the active material, conductive carbon black (Super P, Timcal), and poly(vinyl difluride) (PVDF, Alfa Aesar GmbH & Co. KG) in N-methyl-2-pyrrolidone (NMP, anhydrous 99.5%, Sigma−Aldrich) with a weight ratio of 7:2:1. The resulting slurry was uniformly cast onto Cu/C foil (10 μm) with a doctor blade apparatus, followed by drying at 80 °C in vacuum for 12 h. The electrodes of 12 mm in diameter were punched out with a disc cutting machine (MSK-T10, MTI Corp.). The loading mass of active material in each electrode of 12 mm in diameter is 1.5–2.0 mg $cm^{-2}$. 1.0 M $LiPF_6$ in ethylene carbonate/diethyl carbonate/dimethyl carbonate (EC/DEC/DMC) in a volume ratio of 1:1:1 was used as the electrolyte (battery grade, $H_2O \leq 15$ ppm, HF ≤ 50 ppm, Sigma−Aldrich). Lithium foil (0.45 mm, Cambridge Energy Solutions) was used as the negative electrode, and a glass fiber (260 μm, Whatman) was used as the separator. 85 μL of electrolyte was added to each coin cell. The full cells were assembled in coin cells (stainless steel, CR2032, Shandong Gelon Lib Co., Ltd) with commercial $LiFePO_4$ as the positive electrode and $dt$-$Nb_{12}WO_{33}$ as the negative electrode. The capacity ratio of the negative to positive electrode (N/P ratio) was about 0.93. The charge/discharge performance was carried out on a Land CT2001A battery test system in a voltage range of 1.0−3.0 V. CV was performed at various scan rates on a Bio-Logic VMP3 multichannel potentiostat/galvanostat within 1.0−3.0 V. The Coulombic efficiency of half-cells was defined as the ratio of charge capacity to discharge capacity in each cycle. All electrochemical measurements were performed at 25 ± 2 °C. Two or three independent cells were tested for a single

electrochemical experiment. The data presented was selected from a representative cell that reflects the typical performance.

## Materials characterization

The XRD measurements were carried out at PETRA-III at DESY (Deutsches Elektronensynchrotron) in Hamburg, Germany, using synchrotron radiation ($\lambda = 0.1036$ Å). Rietveld refinement was conducted with the GSAS-II software[53]. The Raman spectra were carried out on an XPLORA plus Raman microscope with a 532 nm laser. Transmission electron microscopy (TEM), high-resolution TEM (HR-TEM), high-angle annular dark-field scanning transmission electron microscopy (HAADF-STEM), and energy dispersive X-ray (EDX) elemental mapping analyses were conducted on a FEI Talos F200S scanning/transmission electron microscope (S/TEM) at an acceleration voltage of 200 kV. Aberration-corrected STEM (AC-STEM) analysis was performed using a FEI Titan Themis Cubed STEM microscope (FEI Company), equipped with a Cs probe corrector and operated at 300 kV. HAADF images were acquired with collection angles between 66 and 200 mrad to improve Z-contrast imaging. The electrodes for ex-situ AC-STEM, ex-situ XAS, and ex-situ Raman spectra were tested in coin cells cycled at 0.1 C, then disassembled, collected, and washed with dimethyl carbonate (DMC) three times to remove residual electrolyte in an Ar-filled glove box at $25 \pm 2$ °C, followed by vacuum drying. Subsequently, the electrodes were sealed in a glass vial under an Ar atmosphere until the measurement. The morphology of the samples was evaluated using a Phenom Pharos Desktop SEM from Phenom World using an accelerating voltage of 10 kV and a secondary electron detector. The elemental compositions of materials were analyzed by inductively coupled plasma-optical emission spectrometry (ICP-OES, Agilent 5110). The single-crystal X-ray diffraction data was collected with a BRUKER D8 VENTURE area detector with Mo-K radiation ($\lambda = 0.71073$). The structure refinement was conducted with Shelx[54]. *Operando* XRD patterns were measured on a STOE STADI MP diffractometer using Mo-Kα radiation ($\lambda = 0.70930$ Å) at a scan rate of 0.62°/min with a step size of 0.495 °. The electrochemical tests were performed within the voltage window of 1.0–3.0 V at 0.1 C and $25 \pm 2$ °C on a Bio-Logic VMP3 multichannel potentiostat/galvanostat. High-resolution synchrotron X-ray diffraction and total scattering measurements were performed at beamline ID31 at the European Synchrotron Radiation Facility (ESRF). The sample powders were loaded into cylindrical slots (approx. 1 mm thickness) held between Kapton windows in a high-throughput sample holder. Each sample was measured in transmission with an incident X-ray energy of 75.00 keV ($\lambda = 0.1653$ Å). Measured intensities were collected using a Pilatus CdTe 2 M detector (1679 × 1475 pixels, 172 × 172 μm² each) positioned with the incident beam in the corner of the detector. X-ray absorption spectra at the Nb K-edge (18986 eV) and W $L_{III}$-edge (10207 eV) were collected at the BAM*line* of the BESSY-II (Berlin, Germany)[55], operated by the Helmholtz-Zentrum Berlin for Materials and Energy. The incident X-ray beam was provided by a super bend magnet (7 T) source and subsequently energetically narrowed by a Double Crystal Monochromator (DCM), with Si (111) crystals, which is used to fine scan the energy range and an intrinsic resolution of $\Delta E / E = 2 \times 10^{-4}$. The beam size on the sample was 4 mm (horizontal) × 2 mm (vertical). The XAS measurements were performed in transmission mode, containing both X-ray absorption near-edge structure (XANES) and extended X-ray absorption fine structure (EXAFS). Pouched coin cells with Kapton film windows were assembled for *operando* XAS measurements. The pouched coin cell was placed between two Argon-filled ionization chambers ($I_0$, $I_1$, 5 cm and 15 cm long, respectively). The electrochemical tests were performed within the voltage window of 1.0–3.0 V at 0.15 C on a Bio-Logic VMP3 multichannel potentiostat/galvanostat at $25 \pm 2$ °C. Nb and W metal foils were used to calibrate

the energy at the respective energies. The measurement protocol was the following: 10 eV steps until 20 eV before the edge, followed by 0.5 eV steps until 20 eV above the edge, and 2 eV steps until 200 eV above the edge. From then on, equidistant k-steps were taken every 0.04 Å until 16 Å. The acquired spectra were extracted, calibrated, and normalized using the Athena and Larch software[56]. XANES spectra were normalized using the Athena software by subtracting a fitted pre-edge baseline and dividing by the edge step obtained from a post-edge fit, resulting in the spectra with a pre-edge near zero and a post-edge near one. The Fourier Transformations for the EXAFS spectra fitting is made in k-space between 2 and 14 Å$^{-1}$ for Nb K-edge, and 2 and 12 Å$^{-1}$ for W $L_{III}$-edge. The resulting Nb K-edge R-space is used for fitting with the model of tetragonal $Nb_2O_5$ and monoclinic $Nb_{12}WO_{33}$. The resulting W $L_{III}$-edge R-space is used for fitting with the model of monoclinic $Nb_{12}WO_{33}$. The Fourier Transformations for the operando EXAFS spectra are made in k-space between 2 and 13 Å$^{-1}$ for Nb K-edge, and 2 and 9 Å$^{-1}$ for W $L_{III}$-edge. The FT-EXAFS plots are the result of this.

## Data availability

The authors declare that all data supporting the findings of this study are available in the article and its Supplementary Information files. The data generated in this study can be obtained from the corresponding authors upon request. Source data are provided with this paper.

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

## Acknowledgements

Y.L. acknowledges the fellowship from the China Scholarship Council (CSC, No.202106200022). Christoph Erdmann is acknowledged for

transmission electron microscopy measurements. XAS experiments were performed at the BAM*line* at the BESSY-II storage ring (Helmholtz Centre Berlin). We thank the Helmholtz-Zentrum Berlin für Materialien und Energie for the allocation of synchrotron radiation beamtime. We acknowledge the European Synchrotron Radiation Facility (ESRF) for the provision of synchrotron radiation facilities. We would like to thank the Momentum Transfer team for facilitating the measurements and Jakub Drnec for assistance and support in using beamline ID31. The measurement setup was developed with funding from the European Union's Horizon 2020 research and innovation program under the STREAMLINE project (grant agreement ID 870313). L.A.M. acknowledges LNNano – Brazilian Nanotechnology National Laboratory (CNPEM/MCTI) for the use of the Electron Microscopy Laboratory (LME) open access facility and FAPEMIG/Brazil for financial support. Wei Zhang is acknowledged for fruitful discussions. Xingyu Yao is acknowledged for assistance in *operando* XAS measurements. We acknowledge support by the Open Access Publication Fund of Humboldt-Universität zu Berlin.

## Author contributions

P.A.R. and N.P. conceived and supervised the project; Y.L. performed the materials syntheses, structural characterizations, electrochemical testing, operando XRD, and data processing; A.G.B. carried out ex-situ and operando XAS measurements; L.A.M. performed AC-STEM; H.L. performed PDF experiments; Y.L. and Y.E.L. performed Raman spectra; F.E. analyzed the powder XRD and single-crystal XRD data; Y.L. and P.A.R. wrote the manuscript. All authors contributed to the discussion and revision of the manuscript.

## Funding

## Competing interests

The authors declare no competing interests.
