## [Peer Review file · Nature Communications]

A Partially Disordered Crystallographic Shear Block Structure as Fast-Charging Negative Electrode Material for Lithium-Ion Batteries

Corresponding Author: Professor Nicola Pinna

Version 0:

Reviewer comments:

Reviewer #1

(Remarks to the Author)

Review Summary: It appears entirely possible that all conclusions from the manuscript relating material differences to electrochemical performance could come down to the particle images in Figure S3. The diameters across continuous regions of the dt phase are 100 to 300 nm, while those of the m phase are about 1 μm . Thus, diffusion times about 100 times longer should be expected for the ordered m phase based on diffusion length alone. It appears that the authors made small particles of Nb₁₂WO₃₃ by annealing from solution at a lower temperature than typically used for niobium tungsten oxides. No conclusion can be drawn on whether the disordered blocks have any effect on the rate. As a result of this alone, despite the myriad other issues described below, this work cannot achieve its stated goals: "This study highlights the benefits of introducing disorder into niobium tungsten oxide shear structures, through the establishment of clear structure- performance correlations, offering valuable guidelines for designing materials with targeted properties." It could seriously mislead the field for this reason and for the hyperbolic language combined with underexplained and/or underreferenced phenomena described below.

Why is so much broadening applied to the calculated pattern of the dt I₄/mmm phase? The calculated peaks are much broader than the experimental peaks.

How are the biphasic phase transitions being determined? The operando data (left side) of Figure 4 show much more clearly a biphasic transition in the dt phase only, while the m phase looks to be monophasic, possibly with reflections moving past one another.

Can you explain the 'self-discharging' and why are the niobium and tungsten oxidation states not 5+ and 6+ at x = 0 in the XANES spectra? Are the compounds reduced? Are they black in colour? Why would self-discharging occur in x = 0? They should not even be inside batteries. But if they are, do the voltages show self-discharge?

Reported XRD state that there is a phase transition for the dt phase but not the m phase while the operando EXAFS says that m phase undergoes the transition: "m-Nb₁₂WO₃₃ remains constant from the OCV state to Li_{3.0}Nb₁₂WO₃₃. However, it subsequently undergoes a dramatic decrease from Li_{3.0}Nb₁₂WO₃₃ to Li_{7.1}Nb₁₂WO₃₃, which is associated with the two-phase transition reaction. In contrast, dt-Nb₁₂WO₃₃ experiences a more stable evolution within the same range."

This sentences about EXAFS changes, especially considering the particle size magnitudinal differences, seems unjustified: "This robustness ensures minimal disruption of the lattice during lithiation, providing stable pathways for Li⁺ ions diffusion and thus better rate capability." EXAFS also does not particularly measure the lattice, but gives local interatomic information. Also, the almost identical sentence is used later in the EXAFS section: "This result reveals a topotactic evolution of the local structure of the tetrahedral sites, arising from the inherent robustness of the disordered structural arrangement present in dt-Nb₁₂WO₃₃, which inhibits local structural disruption during lithiation and preserves its overall integrity, and thus contributes to a stable intercalation lattice that facilitates fast Li⁺ ion diffusion." What is topotactic local structure? This is all repeated once more at the end of the EXAFS section: "The data suggests that the disorder of the dt-Nb₁₂WO₃₃ structure, introduced

by the variability of the ReO₃-type blocks' dimensions and consequent disorder of the tetrahedral tungsten sites position, as well as distortion of those tetrahedra, leads to a robust structure that provides suitable and stable pathways for Li⁺ ions diffusion. As a result, the disorder of the structure allows for Li⁺ fast diffusion and therefore better rate capability than m-Nb₁₂WO₃₃, together with higher stability." A few sentences later in the conclusion, it is repeated again: "resulting instead in a topotactic structural evolution, thus allowing higher Li⁺ diffusion rates."

Back to a similar sentence in the abstract, "leading to a topotactic local structure evolution during cycling, as determined by operando X-ray diffraction and X-ray absorption spectroscopy.", how can XRD inform local structure evolution?

How do S27 and S28 do what is stated in the following: "The second derivative of the W LIII-edge XANES spectra suggest that tungsten remains in tetrahedral coordination in m-Nb₁₂WO₃₃ and dt-Nb₁₂WO₃₃ during the discharge (Figures S27, S28)"

What is the meaning of this statement? Reference 19 does not even appear to contain XAS data. "The magnitude of the Nb-Nb and Nb-(O-)Nb peaks decrease continuously, which can be attributed to the shielding effect of Nb-Nb corner sharing interactions during lithium insertion into the lattice¹⁹."

Why does block plane contraction give rise to a new peak rather than a shift in peak position? "It is noted that a new peak corresponding to a Nb-(O-)Nb interatomic distance appears in the spectra at a lower radial distance at 3.1 Å and remains unchanged until the fully discharged state. This new peak arises from the contraction within the block plane."

Should not changes in the W EXAFS be expected even for the dt phase since lattice contraction and expansion are taking place?

Reversibility and rate are conflated throughout the manuscript. It is repeatedly said that higher structural reversibility gives higher rate. This does not make sense. Anyway, smaller particles are giving higher rate.

Reviewer #2

(Remarks to the Author)

Liu et al. report on the synthesis, structural characterization and electrochemical behavior of two forms of Nb₁₂WO₃₃, the known stable monoclinic phase obtained at high synthesis temperature and a little ordered block-type structure obtained at lower temperature. While m-Nb₁₂WO₃₃ has already been proven to be an excellent fast-charging anode material in previous studies, its capability is exceeded by little ordered dt-Nb₁₂WO₃₃ that shows a strikingly superior Li-uptake and release behavior as well as an increased cycling stability that is convincingly evidenced by charging and discharging profiles, CV curves and rate capability measurements at different capacities. These findings are unexpected and indeed remarkable.

The morphology and structure of both starting samples were comprehensively characterized by powder XRD, SEM + EDXS, HAADF-STEM, XANES and Raman spectroscopy. Structural changes during the first cycle were studied in detail by operando XRD and changes of the oxidation state and coordination of the metals by operando XANES and EXAFS, respectively, and a lot of useful information was collected. The measurements and interpretation of the results were obviously carried out with great care and expertise. Based on the data obtained, the authors conclude that the disorder, in particular that of the W-occupied tetrahedral positions, is responsible for fast Li diffusion and increased rate capability in dt-Nb₁₂WO₃₃. The main weak point of this study is that the interpretation of the data relies on a structure model for m-Nb₁₂WO₃₃ which is likely not fully accurate. The structure determination based on single crystal X-ray data was done by Roth and Wadsley almost 60 years ago (ref. 21). Though this work had certainly been of the utmost quality at that time, the structure determination does not meet today's standards, e. g. because only a part of the data obtained from a tiny needle-shaped crystal was used (h0l and h1l reflections). However, the structure is basically correct apart from the claim that W exclusively occupies the tetrahedral position. This configuration provided a better fit of F_{obs} and F_{calc} in the original study. Moreover, Roth and Wadsley found such a preference of W for the tetrahedral site also for three other block-type niobium tungsten oxides.

These models have been undisputed in literature - and mostly still are - despite contradicting experimental evidence. In 1983, a neutron diffraction study of Nb₁₄W₃O₄₄ by Cheetham and Allen unraveled a varying cation distribution: the highest W content was found for the central four octahedra (39 % W) and while the tetrahedral site contains 27 % W only.⁽¹⁾ Based on that observation, it seems likely that W also shows in m-Nb₁₂WO₃₃ a preference for the centers of the blocks as well as for tetrahedral site while the sites in the CS planes are disfavored but does not occupy 100% of the tetrahedral site.

If the original structural model was correct, HAADF-STEM images of m-Nb₁₂WO₃₃ would clearly show the tetrahedral position with extreme high brightness. Due to Z contrast with $I \sim Z^2$, a pure W position should be about 3 times brighter than Nb. This is not the case for dt-Nb₁₂WO₃₃. Note that in HAADF-STEM images of the W-richer phase Nb₈W₉O₄₇, the position of a 100% W site appears very bright besides grey ones with mixed occupancy.⁽²⁾ On the first view on the HAADF-STEM images of dt-Nb₁₂WO₃₃ (Figures 2 and S8), the intensities of the metal positions appear to be differing in a small intensity range and rather randomly. However, there are some recognizable trends. That the tetrahedral sites in combination with their close neighbors appear as bright patches in thicker regions of Figure 2 is owing to a higher density of scattering centers there (i.e. smaller M-M distances) than inside the blocks. The analysis of the intensity distribution in thin areas by line intensity profiles (Figure R1) shows that (i) the positions inside the blocks tend to be generally brighter than those in the shear planes and (ii) the positions in the tetrahedral site are equally bright or not much brighter than the surrounding ones and the ones in the block centers (Figure 2e, Line 5 in Figure R1). This is not only the case for the defective block structure

investigated here but also for the perfect structure of $\text{Nb}_{14}\text{W}_3\text{O}_{44}$ (cf. Figures 1f in (3) or 2h in ref. 7). A HAADF-STEM image of $\text{m-Nb}_{12}\text{WO}_{33}$ along [010] (unfortunately only the direction [1-10] is investigated here (Figure S2)) would reveal the discrepancy from the idealized model (ref 21) most pronounced as the tetrahedral position is supposed to be 100% W while the others are 100 % Nb. The interpretation of the operando results and the conclusions concerning the role of the tetrahedral site for improved electrochemical behavior of $\text{dt-Nb}_{12}\text{WO}_{33}$ are based on the classic structure model for $\text{m-Nb}_{12}\text{WO}_{33}$ that appears to be wrong in respect of the occupancy of the tetrahedral and octahedral positions by W and Nb. As a correct determination of the W distribution in $\text{m-Nb}_{12}\text{WO}_{33}$ is indispensable for an accurate evaluation of the experimental results, I encourage the authors to try the measurement of single crystal X-ray data for an up-to-date structure determination. They have grown well-developed crystals of $\text{m-Nb}_{12}\text{WO}_{33}$ (Figure S3) that are small but possibly suitable nonetheless. Such a study would complement the insights gained by HAADF-STEM images of this structure along [010]. It should be noted that the structure of $\text{dt-Nb}_{12}\text{WO}_{33}$ is closely related if not identical to that of $\text{M-Nb}_2\text{O}_5$, a metastable Nb_2O_5 polymorph obtained at 900 °C as well.(4) It is tetragonal with the lattice constants being $a = 20.44\text{Å}$, $c = 3.832\text{Å}$ which are very similar to those of $\text{dt-Nb}_{12}\text{WO}_{33}$ (cf. Figure S1b). In the first article by Mertin et al., a simple structure with [4x4] blocks was suggested but a subsequent HRTEM study by Heurung und Gruehn revealed a little ordered structure with differing block sizes like that now observed for $\text{dt-Nb}_{12}\text{WO}_{33}$.(5) Moreover, this sample was prepared with an addition of a small amount of WO_3 which apparently stabilizes the M modification. It is interesting to note that in Figure S8 at the bottom of the central part, there is a domain with [4x4] blocks corresponding to the original suggestion for the structure of $\text{M-Nb}_2\text{O}_5$. Thus, the present investigation unknowingly confirms this structural model after more than half a century – an amazing finding. The term “disordered” as used in the title and at different places in the manuscript is somewhat misleading, because the order is perfect along the c axis whereas the arrangement of blocks with varying sizes in the ab plane indeed is little ordered. Additional HAADF-STEM images of $\text{m-Nb}_{12}\text{WO}_{33}$ after several cycles would provide valuable information about the preservation of the block structure during electrochemical reaction.

Further comments:

Page 4: “The pattern profile is best fitted with a tetragonal cell (space group $I4/mmm$), although the atomic positions cannot be properly refined due to the presence of disorder in the structure (Figure S1).” How was the Rietveld refinement of the XRD pattern possible without a plausible structural model for $\text{dt-Nb}_{12}\text{WO}_{33}$?

Page 4-5: The descriptions of Figures S4b and S4e are mixed up.

Page 9: “... Li^+ diffusion through tetrahedral site cavities...” Is this possible? These sites should be fully occupied by W and Nb.

Figure 3g: The figure caption is inaccurate as “ $\text{Nb}_{18}\text{W}_{16}\text{O}_{93}$ ” is not a block structure.

Figure 4a,b: The reflection at $2\theta \approx 16^\circ$ is not explained.

Supporting Information:

Materials characterization: the types and characteristics of the two different microscopes used for S/TEM should be presented together.

References:

1. Cheetham, Allen. Cation Distribution in the Complex Oxide, $\text{W}_3\text{Nb}_{14}\text{O}_{44}$; a Time-of-Flight Neutron Diffraction Study. *J. Chem. Soc., Chem. Commun.* 1983, 1370-1372
2. Kirkland, Saxton. Cation Segregation in $\text{Nb}_{16}\text{W}_{18}\text{O}_{94}$ Using High Angle Annular Dark Field Scanning Transmission Electron Microscopy and Image Processing. *J. Microsc.* 2002, 206, 1–6.
3. Yang et al. Achieving Ultrahigh-Rate and High-Safety Li^+ Storage Based on Interconnected Tunnel Structure in Micro-Size Niobium Tungsten Oxides. *Adv. Mater.* 2020, 32, 1905295.
4. Mertin, Anderson, Gruehn. Über die Kristallstruktur von $\text{M-Nb}_2\text{O}_5$. *J. Solid State Chem.* 1970, 1, 419-424.
5. Heurung, Gruehn. Neues zur Struktur und Stabilität von $\text{M-Nb}_2\text{O}_5$. *Z. anorg. allg. Chem.* 1982, 491,101-112.

Reviewer #3

(Remarks to the Author)

The manuscript introduces a novel disordered phase of $\text{Nb}_{12}\text{WO}_{33}$ ($\text{dt-Nb}_{12}\text{WO}_{33}$) designed to enhance fast-charging capabilities in lithium-ion batteries. This study effectively highlights the role of structural disorder within crystallographic shear block structures, demonstrating improved Li^+ -ion diffusion kinetics and rate performance. The use of advanced characterization techniques, including operando XRD and XAS, provides a solid foundation for correlating structural features with electrochemical performance. However, there are key concerns and areas for improvement that must be addressed to enhance the scientific rigor and practical relevance of the study.

1. Discrepancy Between TEM Observations and DFT Model

The irregular tetrahedral features observed in TEM images underscore the structural disorder inherent in $\text{dt-Nb}_{12}\text{WO}_{33}$. However, the computational model employed for DFT calculations appears to assume an idealized structure that does not align with the experimental data. This disconnect raises questions about the accuracy of the computational predictions. Recommendation: Revise the computational model to reflect the experimentally observed disorder, incorporating variability in tetrahedral sites and distortions. Provide a comparative analysis between TEM-derived parameters and DFT inputs to validate the computational approach.

2. Economic Viability of Nb-WO Compared to Nb-Oxides

The inclusion of tungsten in Nb₁₂WO₃₃ introduces higher material costs compared to simpler Nb-based oxides. A detailed discussion of the economic implications and trade-offs between performance enhancements and material cost is essential for evaluating scalability.

3. White-Line Edge Analysis in XANES

The XANES spectra, particularly the white-line edge, provide critical information about electronic structure changes resulting from disorder. However, the current discussion is limited and does not fully explain how disorder enhances performance.

4. Assumptions in GITT-Based Diffusion Coefficient Calculations (Figure 3f)

The reported diffusion coefficients derived from GITT analysis rely on certain assumptions regarding material properties and boundary conditions based on Fick's laws of diffusion. A detailed explanation of these assumptions is crucial for ensuring the reliability of the results.

5. Structural Stability and Long-Term Performance

Although the manuscript demonstrates excellent capacity retention over 1000 cycles, it lacks a discussion of potential degradation mechanisms. Long-term cycling may induce fractures or irreversible distortions within the disordered structure.

6. Comparison with Other Nb-Based Anode Materials

The manuscript compares dt-Nb₁₂WO₃₃ with its ordered counterpart (m-Nb₁₂WO₃₃). However, a broader comparison with other Nb-based anode materials from the literature would contextualize the improvements reported in this study.

Version 1:

Reviewer comments:

Reviewer #1

(Remarks to the Author)

The manuscript has been greatly improved with the revisions. It is now nearly ready for publication. One issue, not on the technical part, is that the data in the technoeconomics section is dated, in part, and the current prices are inaccurate. Specifically, the sentence "Currently, the prices of niobium and tungsten are approximately 80 USD/kg and 43 USD/kg, respectively." contains values that are inaccurate, and this line does not even have a citation. It is challenging to get good mineral prices, especially for certain elements, but one should compare on an equivalent basis (such as EXW China, Europe, etc) or, if unknown, it would be better not to include faulty data that will mislead others. A more challenging issue that must be addressed is that the comparison in Figure 3g is misleading since a high carbon content (20%) was used in the electrodes in this work. The loading was on the low side of standard, so that could be acceptable, but the carbon content also has a big role and high current data in particular cannot be compared directly with this discrepancy.

Reviewer #2

(Remarks to the Author)

The revision of the manuscript by Liu et al. has been done thoroughly and comprehensively. All reviewer comments have been considered in an adequate manner, and the overall quality of this excellent article has been further improved thereby. In particular, the structural description of the two modifications of Nb₁₂WO₃₃ is now complete: the single crystal structure determination of m-Nb₁₂WO₃₃ and the careful examination of HAADF-STEM images of dt-Nb₁₂WO₃₃ sheds new light onto important structural questions such as the localization of W in the block structures. Moreover, these findings allow a more reliable interpretation of the spectroscopic results obtained for the initial and the lithiated samples. The additional investigation of the influence of crystal size on electrochemical properties is certainly of widespread interest. I recommend the publication of the article after the following minor points have been addressed:

1. Title and text: "partially ordered" is nowhere explained (suggestion: introduce it on page 3 where it is first mentioned in the text).

2. page 1: "a disordered arrangement of the tungsten tetrahedra at the corners of the blocks,"

page 3: "an irregular distribution of the tetrahedral sites within the structure". Can this distribution really be called "disordered" or "irregular"? The tetrahedral sites still occur at their "regular" positions, namely at the block corners. Maybe it is better to combine the two parts of the description, e. g.: "The partially disordered Nb₁₂WO₃₃ phase consists of corner-shared NbO₆ octahedra blocks of varied sizes, including 5x4, 4x4, and 4x3, causing a little ordered arrangement of the tungsten tetrahedra at the corners of the blocks, as well as distortion of the WO₄ tetrahedra."

Also in Fig. 1b, the designation "disordered" and "ordered tetrahedral position" is somewhat misleading. Better, e. g.: "disordered block structure" and "3x4 blocks with tetrahedral sites"

3. page 4: "we have synthesized larger crystals" Larger than what? Better give crystal size (1-3 microns) instead.

4. page 4: "According to the refinement data in Table S1, the tungsten atoms show a preference for the tetrahedral sites at the corners of the blocks, with a 100 % occupancy."

Misleading. In Table S1, the occupancy is given as 0.4438. This means that the site is only partially filled (not 100%) but that there is exclusively W.

From the crystallographic point of view, it is interesting to note that the y value is given as 0.2500(2). Is this measuring error really significant? It turns a high symmetry 2b position (0, 0.25, 0) into a 4g (0,y,0).

The x,y,z values in the header of Table S1 are incorrectly given in Angstrom but these are fractional coordinates without dimensions.

5. page 7 center: "suggest that tungsten is in the oxidation state 6+ in both m-Nb₁₂WO₃₃ and dt-Nb₁₂WO₃₃." Sentence used twice here.

6. page 7: "The magnitude of the ligand field split is higher -> highest for cations in octahedral field,..."

7. page 8: "whereas a weaken -> weak intensity band is observed for dt-Nb₁₂WO₃₃,"

8. page 15: The abbreviations "BM" and "MS" should be introduced properly.

9. Give the direction of observation for Figures S9, S31 and S32.

10. Table 2 ff: curvfit -> curve fit or curve fitting

Reviewer #3

(Remarks to the Author)

During the revision process, the authors significantly improved the overall quality of the manuscript. However, I still have concerns regarding the commercial viability of Nb-oxide anode materials for electric vehicle (EV) applications. The authors state the following: "As for niobium, the price remained stable at around 40–50 USD/kg since 2010, and was 50 USD/kg in 2017. Currently, the prices of niobium and tungsten are approximately 80 USD/kg and 43 USD/kg, respectively."

In comparison, the current price of graphite is around 5–10 USD/kg, while Si/C composites are priced at approximately 60–100 USD/kg. However, Si-based anode materials offer much higher energy densities (200–300 Wh/kg), which provide a compelling advantage for EV markets. In contrast, Nb-oxide still suffers from relatively low energy densities and requires a larger volume of material in battery packs, which may hinder its competitiveness in practical EV applications. It seems usage of Nb-oxides have no beneficial as an anode material.

Version 2:

Reviewer comments:

Reviewer #2

(Remarks to the Author)

All comments and objections have been answered adequately. Thus, I recommend the publication of this manuscript in the current form.

REVIEWERS' COMMENTS

We thank all the reviewers for their constructive comments and suggestions to improve our work. The manuscript has been revised according to their recommendations. We have included additional data and expanded the discussion, as detailed in our point-by-point responses below. All corresponding modifications in the manuscript are highlighted in yellow.

Reviewer #1 (Remarks to the Author):

(1) Review Summary: It appears entirely possible that all conclusions from the manuscript relating material differences to electrochemical performance could come down to the particle images in Figure S3. The diameters across continuous regions of the dt phase are 100 to 300 nm, while those of the m phase are about 1 μm . Thus, diffusion times about 100 times longer should be expected for the ordered m phase based on diffusion length alone. It appears that the authors made small particles of $\text{Nb}_{12}\text{WO}_{33}$ by annealing from solution at a lower temperature than typically used for niobium tungsten oxides. No conclusion can be drawn on whether the disordered blocks have any effect on the rate. As a result of this alone, despite the myriad other issues described below, this work cannot achieve its stated goals: "This study highlights the benefits of introducing disorder into niobium tungsten oxide shear structures, through the establishment of clear structure-performance correlations, offering valuable guidelines for designing materials with targeted properties." It could seriously mislead the field for this reason and for the hyperbolic language combined with underexplained and/or underreferenced phenomena described below.

Reply: The size of the particles can have a big effect on the electrochemical performance of a material, as mentioned by the reviewer, particularly when the size of the particles is decreased down to the nanoscale, as it shortens the ion diffusion paths. However, the size of the particles of the materials studied here are in the sub-micron range, where those effects are less important. There are also differences in the structure evolution of the materials during discharge and charge, which should not be expected if the differences in performance were due to the differences in the size of the particles.

To clarify the effect of particle size on the performance of the materials, we have added additional results for *m*- $\text{Nb}_{12}\text{WO}_{33}$ and *dt*- $\text{Nb}_{12}\text{WO}_{33}$ with identical particle sizes. The manuscript has been revised to include these new results. The following discussion and figures have been added to the

manuscript and supplementary information. The data shows that, for the particle size range studied here, particle size effects do not contribute significantly to the differences observed in the performance.

“Considering that $dt\text{-Nb}_{12}\text{WO}_{33}$ consists of smaller particles than $m\text{-Nb}_{12}\text{WO}_{33}$, we have additionally prepared and studied the electrochemical performance of $m\text{-Nb}_{12}\text{WO}_{33}$ with smaller particles (100-400 nm) and $dt\text{-Nb}_{12}\text{WO}_{33}$ with larger particles (1.0-1.3 μm) to evaluate the effect of the particle size on the performance. $m\text{-Nb}_{12}\text{WO}_{33}$ ($m\text{-Nb}_{12}\text{WO}_{33}\text{-BM}$) with a particle size similar to that of $dt\text{-Nb}_{12}\text{WO}_{33}$ was produced by ball milling of the initial $m\text{-Nb}_{12}\text{WO}_{33}$ material. $dt\text{-Nb}_{12}\text{WO}_{33}$ dense microspheres ($dt\text{-Nb}_{12}\text{WO}_{33}\text{-MS}$) with sizes 1.0-1.3 μm were synthesized via solvothermal method and subsequent calcination at 900 $^{\circ}\text{C}$ under air. The XRD patterns of $m\text{-Nb}_{12}\text{WO}_{33}\text{-BM}$ and $dt\text{-Nb}_{12}\text{WO}_{33}\text{-MS}$ are similar to those of $m\text{-Nb}_{12}\text{WO}_{33}$ and $dt\text{-Nb}_{12}\text{WO}_{33}$, respectively, except for a slight broadening of the peaks for $m\text{-Nb}_{12}\text{WO}_{33}\text{-BM}$, caused by the decrease of the particle size, and a slight narrowing of the peaks for $dt\text{-Nb}_{12}\text{WO}_{33}\text{-MS}$, caused by the increase of the particle size (**Figures S25a, S25b**). The Raman spectra of $m\text{-Nb}_{12}\text{WO}_{33}\text{-BM}$ and $dt\text{-Nb}_{12}\text{WO}_{33}\text{-MS}$ are also identical to those of their $m\text{-Nb}_{12}\text{WO}_{33}$ and $dt\text{-Nb}_{12}\text{WO}_{33}$ counterparts (**Figures S25c, S25d**). SEM and TEM images (**Figure S26**) show that the particle size of $m\text{-Nb}_{12}\text{WO}_{33}\text{-BM}$ ranges from 100 to 400 nm, while those of $dt\text{-Nb}_{12}\text{WO}_{33}$ microspheres are in the range of 1-1.3 μm . The HR-TEM images of $m\text{-Nb}_{12}\text{WO}_{33}\text{-BM}$ and $dt\text{-Nb}_{12}\text{WO}_{33}\text{-MS}$ show lattice fringes with an interplanar spacing of 0.49 nm and 0.28 nm, consistent with the d-spacing of the (003) and (101) planes of $m\text{-Nb}_{12}\text{WO}_{33}\text{-BM}$ and $dt\text{-Nb}_{12}\text{WO}_{33}\text{-MS}$, respectively. Additionally, the (71-1), (-407) and (316) planes of the monoclinic structure are observed in the SAED pattern of $m\text{-Nb}_{12}\text{WO}_{33}\text{-BM}$. The tetragonal structure of $dt\text{-Nb}_{12}\text{WO}_{33}\text{-MS}$ is further confirmed through the diffraction associated with the (460), (-170) and (5-10) planes observed in the SAED pattern. The electrochemical properties of $m\text{-Nb}_{12}\text{WO}_{33}\text{-BM}$ and $dt\text{-Nb}_{12}\text{WO}_{33}\text{-MS}$ were investigated using half-cells. The initial Coulombic efficiency of $m\text{-Nb}_{12}\text{WO}_{33}\text{-BM}$ is 86.1%, which is close to that of $m\text{-Nb}_{12}\text{WO}_{33}$ (87.3%), while $dt\text{-Nb}_{12}\text{WO}_{33}\text{-MS}$ exhibits an initial Coulombic efficiency of 94.4%, similar to the 95.3% for $dt\text{-Nb}_{12}\text{WO}_{33}$ (**Figure S27a**). $m\text{-Nb}_{12}\text{WO}_{33}\text{-BM}$ exhibits specific capacities of 56.8 and 29.2 mAh g^{-1} at 40 and 80 C, respectively, which are comparable to the 49.2 and 30.9 mAh g^{-1} delivered by $m\text{-Nb}_{12}\text{WO}_{33}$ at the same rates (**Figures S27b, S27c**). Therefore, decreasing the particle size of $m\text{-Nb}_{12}\text{WO}_{33}$ to sizes similar to those of the $dt\text{-Nb}_{12}\text{WO}_{33}$ does not improve the rate capability of the material. Moreover, $dt\text{-}$

$\text{Nb}_{12}\text{WO}_{33}\text{-MS}$ delivers a high capacity of 108.8 mAh g^{-1} at 80 C , which is comparable to that of $dt\text{-Nb}_{12}\text{WO}_{33}$ and much higher than that of $m\text{-Nb}_{12}\text{WO}_{33}$ and $m\text{-Nb}_{12}\text{WO}_{33}\text{-BM}$ (Figure S27d). The Li^+ diffusion coefficients in $m\text{-Nb}_{12}\text{WO}_{33}\text{-BM}$ and $dt\text{-Nb}_{12}\text{WO}_{33}\text{-MS}$ were investigated through GITT (Figure S27e and Figure S27f). The D_{Li^+} of $dt\text{-Nb}_{12}\text{WO}_{33}\text{-MS}$ and $m\text{-Nb}_{12}\text{WO}_{33}\text{-BM}$ are in the ranges 1.4×10^{-10} – 1.1×10^{-11} and 3.8×10^{-11} – 3.3×10^{-13} , respectively. These values are comparable to those of $dt\text{-Nb}_{12}\text{WO}_{33}$ and $m\text{-Nb}_{12}\text{WO}_{33}$, respectively. The D_{Li^+} of $dt\text{-Nb}_{12}\text{WO}_{33}\text{-MS}$ continues to be larger than those of $m\text{-Nb}_{12}\text{WO}_{33}\text{-BM}$, despite the larger particles of the former sample. These results show that changing the size of the particles of the materials between $100\text{-}400 \text{ nm}$ and ca. $1 \mu\text{m}$ does not significantly affect the lithium diffusion coefficients or rate capability of the materials, and therefore, the differences in the electrochemical performance of the $dt\text{-Nb}_{12}\text{WO}_{33}$ and $m\text{-Nb}_{12}\text{WO}_{33}$ are caused primarily by the differences in their structures.”

Figure S25. XRD patterns of (a) $m\text{-Nb}_{12}\text{WO}_{33}\text{-BM}$ and $m\text{-Nb}_{12}\text{WO}_{33}$, and (b) $dt\text{-Nb}_{12}\text{WO}_{33}\text{-MS}$ and $dt\text{-Nb}_{12}\text{WO}_{33}$. Raman spectra of (c) $m\text{-Nb}_{12}\text{WO}_{33}\text{-BM}$ and $m\text{-Nb}_{12}\text{WO}_{33}$, and (d) $dt\text{-Nb}_{12}\text{WO}_{33}\text{-MS}$ and $dt\text{-Nb}_{12}\text{WO}_{33}$.

Figure S26. (a) SEM, (b) TEM, (c) HR-TEM and SAED images of *m*-Nb₁₂WO₃₃-BM. (d) SEM, (e) TEM, (f) HR-TEM and SAED images of *dt*-Nb₁₂WO₃₃-MS.

Figure S27. (a) Initial discharge-charge profiles of $m\text{-Nb}_{12}\text{WO}_{33}\text{-BM}$ and $dt\text{-Nb}_{12}\text{WO}_{33}\text{-MS}$. (b) Rate capability of $m\text{-Nb}_{12}\text{WO}_{33}\text{-BM}$ and $dt\text{-Nb}_{12}\text{WO}_{33}\text{-MS}$. Discharge-charge profiles of (c) $m\text{-Nb}_{12}\text{WO}_{33}\text{-BM}$ and (d) $dt\text{-Nb}_{12}\text{WO}_{33}\text{-MS}$ at various rates. (e) GITT profiles of $m\text{-Nb}_{12}\text{WO}_{33}\text{-BM}$ and $dt\text{-Nb}_{12}\text{WO}_{33}\text{-MS}$ as a function of x in $\text{Li}_x\text{Nb}_{12}\text{WO}_{33}$. (f) Variation of the D_{Li^+} determined by GITT with potential.

The following experimental procedures were added to the experimental part in the supplementary information.

“ $m\text{-Nb}_{12}\text{WO}_{33}\text{-BM}$ was prepared by ball milling of $m\text{-Nb}_{12}\text{WO}_{33}$ using a zirconium ball in a swing ball mill with a frequency of 20 Hz in a jar volume of 10 mL (SBM, Retsch MM400). The material was ball milled with a ball-to-powder ratio of 25:1 for 90 min. To synthesize the $dt\text{-Nb}_{12}\text{WO}_{33}$ microspheres ($dt\text{-Nb}_{12}\text{WO}_{33}\text{-MS}$), stoichiometric amounts of niobium chloride and tungsten chloride were dissolved in 16 mL of ethanol, and the solution was transferred to a Teflon-lined

stainless-steel autoclave, and then heated at 180 °C for 2 days. The product was collected by centrifugation and washed with absolute ethanol, and subsequently heated in air at 900 °C for 10 h.”

(2) Why is so much broadening applied to the calculated pattern of the dt $I4/mmm$ phase? The calculated peaks are much broader than the experimental peaks.

Reply: The XRD pattern of $dt\text{-Nb}_{12}\text{WO}_{33}$ could not be accurately refined due to the presence of disorder in the structure and therefore absence of a suitable structural model. We used the structure of $M\text{-Nb}_2\text{O}_5$ as a model, which is closely related to that of $dt\text{-Nb}_{12}\text{WO}_{33}$, to obtain the cell parameters. However, the atomic positions could not be refined. This was not clearly explained in the manuscript. Therefore, in order to avoid misinterpretations, we have now removed the refinement results for $dt\text{-Nb}_{12}\text{WO}_{33}$ and only include the Rietveld refinement results for $m\text{-Nb}_{12}\text{WO}_{33}$. The text was modified accordingly, as shown below. In addition, Figure S1 of the SI has been modified to include only the Rietveld refinement results for $m\text{-Nb}_{12}\text{WO}_{33}$.

“The pattern is consistent with a tetragonal structure (space group $I4/mmm$) similar to that of $M\text{-Nb}_2\text{O}_5$ ^{24,25}.”

Figure S1. Rietveld refinement of the XRD patterns of $m\text{-Nb}_{12}\text{WO}_{33}$ ($R_{wp} = 9.6\%$).

(3) How are the biphasic phase transitions being determined? The operando data (left side) of Figure 4 show much more clearly a biphasic transition in the dt phase only, while the m phase looks to be monophasic, possibly with reflections moving past one another.

Reply: The colors used in the contour plots of the *operando* XRD data were not well-chosen, making it difficult to clearly identify the biphasic phase transition for $m\text{-Nb}_{12}\text{WO}_{33}$. Following the reviewer comment, we have adjusted the color scheme of the contour plots to enhance contrast and improve clarity. Figure 4 of the manuscript has been revised and a new figure has been added to the SI to make the results clearer. The discussion in the manuscript has been revised accordingly. The changes to the manuscript and SI are described below.

“In contrast, $m\text{-Nb}_{12}\text{WO}_{33}$ (Figure 4a and Figure S28) exhibits two biphasic reactions during the initial lithiation process. The (020) diffraction peak of $m\text{-Nb}_{12}\text{WO}_{33}$ ($2\theta\sim 21.4^\circ$) disappears at ca. 1.7 V during the initial discharge process. Simultaneously, a new peak emerges at a lower angle ($2\theta\sim 21.2^\circ$) within the $M2$ region, indicating a phase transition occurring in the plateau region. The peak shifts to lower angle ($M3$) due to the expansion of the lattice along the b direction, and the intensity of the peak begins to decrease when the electrode is discharged to ca. 1.3 V. Within the $M4$ region, a new peak arises on the left side of the pattern ($2\theta\sim 19.3^\circ$), indicative of the biphasic reactions during the final stage of the discharge process.”

Figure S28. *Operando* XRD contour plots of $m\text{-Nb}_{12}\text{WO}_{33}$ at selected angle ranges.

Figure 4. Long-range structure evolution of $m\text{-Nb}_{12}\text{WO}_{33}$ and $dt\text{-Nb}_{12}\text{WO}_{33}$ during lithiation and delithiation. Operando XRD patterns of (a) $m\text{-Nb}_{12}\text{WO}_{33}$ and (b) $dt\text{-Nb}_{12}\text{WO}_{33}$ electrodes, collected for the first discharge and charge processes. The corresponding XRD contour plots at selected angle ranges are shown on the left. (c) Unit cell parameters evolution during Li^+ insertion and extraction for the $dt\text{-Nb}_{12}\text{WO}_{33}$ electrode.

(4) Can you explain the 'self-discharging' and why are the niobium and tungsten oxidation states not 5+ and 6+ at $x = 0$ in the XANES spectra? Are the compounds reduced? Are they black in colour? Why would self-discharging occur in $x = 0$? They should not even be inside batteries. But if they are, do the voltages show self-discharge?

Reply: Self-discharge refers to the loss of charge in lithium-ion batteries that occurs when the battery remains in the open-circuit voltage (OCV) state for a period of time. It happened in our

study because the operando XAS measurements could not be performed immediately after the assembly of the pounced coin cells. Thus, the coin cells were in the OCV state for some hours before the measurement. During this time, reactions between the electrode and electrolyte result in the oxidation of the electrolyte and reduction of the oxidation state of the metals from 5+ and 6+ for Nb and W, respectively. Self-discharge in Li-ion batteries is described in the literature (*Electrochem. Acta* 2002, 47, 1217-1223; *J. Solid State Electrochem.* 2022, 26, 163-170) and has been observed in operando measurements performed by other authors, such as operando XANES measurements of H-Nb₂O₅ (*Energy Environ. Sci.* 2022, 15, 254-264). To demonstrate this self-discharge effect, we show the voltage profiles of *m*-Nb₁₂WO₃₃ and *dt*-Nb₁₂WO₃₃ and images of the respective electrode materials as a function of time (Figure S35). The voltage decreases slowly with time from OCV until ca. 15 h, remaining essentially unchanged after that. The color of the powders changes from white to light blue during this period, a color that is typical of partially reduced tungsten in tungsten-containing oxides.

Figure S35 has been included in the revised supplementary information and additional discussion and references have been added to the revised manuscript (see below).

Figure 5j and 5k, which depicted the variation of the oxidation states of niobium and tungsten as a function of x in Li _{x} Nb₁₂WO₃₃ have been revised to avoid misunderstandings. These figures now depict the variation of the oxidation states of niobium and tungsten as a function of the specific capacity during the lithiation process of the materials (see below).

“Note that the initial oxidation states of Nb and W are lower than 5+ and 6+, respectively, due to self-discharge occurring during the waiting period before the measurements. To demonstrate this self-discharge effect, we show the voltage profiles of *m*-Nb₁₂WO₃₃ and *dt*-Nb₁₂WO₃₃ and images of the respective electrode materials as a function of time (**Figure S35**). The voltage decreases slowly with time from OCV until ca. 15 h, remaining essentially unchanged after that. To study the color change during the waiting time, the *m*-Nb₁₂WO₃₃ and *dt*-Nb₁₂WO₃₃ powders were used as positive electrodes in half cells. The color of the powders changes from white to light blue during this period, a color that is typical of partially reduced tungsten in tungsten-containing oxides.”

Figure S35. (a) Voltage profiles of $m\text{-Nb}_{12}\text{WO}_{33}$ and $dt\text{-Nb}_{12}\text{WO}_{33}$ as a function of time. (b) Images of $m\text{-Nb}_{12}\text{WO}_{33}$ and $dt\text{-Nb}_{12}\text{WO}_{33}$ showing the change in the color of the materials from the initial white to light blue with time.

Figure 5. Operando Nb K-edge and W L_{III}-edge XANES characterization during lithiation and delithiation. Operando Nb K-edge and W L_{III}-edge XANES contour plots of (a) *m*-Nb₁₂WO₃₃ and (b) *dt*-Nb₁₂WO₃₃ (the corresponding voltage profiles for the first cycle are shown on the right side). Operando Nb K-edge XANES spectra of *dt*-Nb₁₂WO₃₃ for the (c) discharge and (d) charge processes. Operando W L_{III}-edge XANES spectra of *m*-Nb₁₂WO₃₃ for the (e) discharge and (f) charge processes. Operando W L_{III}-edge XANES spectra of *dt*-Nb₁₂WO₃₃ for the (g) discharge and (h) charge processes. (i) Pre-edge integrated peak intensity from the Nb K-edge XANES spectra of *m*-Nb₁₂WO₃₃ and *dt*-Nb₁₂WO₃₃. Oxidation states of (j) Nb and (k) W as a function of *x* in Li_{*x*}Nb₁₂WO₃₃ during lithiation.

(5) Reported XRD state that there is a phase transition for the *dt* phase but not the *m* phase while the operando EXAFS says that *m* phase undergoes the transition: "*m*-Nb₁₂WO₃₃ remains constant from the OCV state to Li_{3.0}Nb₁₂WO₃₃. However, it subsequently undergoes a dramatic decrease from Li_{3.0}Nb₁₂WO₃₃ to Li_{7.1}Nb₁₂WO₃₃, which is associated with the two-phase transition reaction. In contrast, *dt*-Nb₁₂WO₃₃ experiences a more stable evolution within the same range."

Reply: The colors used in the contour plots of the operando XRD data made it difficult to clearly identify the biphasic phase transition for *m*-Nb₁₂WO₃₃. We have adjusted the color scheme of the contour plots to enhance contrast and improve clarity. Figure 4 of the manuscript has been revised and a new figure has been added to the SI to make the results clearer. The discussion in the manuscript has been revised accordingly. The changes to the manuscript and SI are described in the reply to the reviewer's comment (3).

It is our interpretation of the operando XRD data that it indicates that *m*-Nb₁₂WO₃₃ undergoes a phase transition. The (020) diffraction peak of *m*-Nb₁₂WO₃₃ ($2\theta \sim 21.4^\circ$) disappears at *ca.* 1.7 V during the initial discharge process. Simultaneously, a new peak emerges at a lower angle ($2\theta \sim 21.2^\circ$) within the *M2* region, indicating a phase transition occurring in the plateau region. The peak shifts to lower angle (*M3*) due to the expansion of the lattice along the *b* direction, and the intensity of the peak begins to decrease when the electrode is discharged to *ca.* 1.3 V. Within the *M4* region, a new peak arises on the left side of the pattern ($2\theta \sim 19.3^\circ$), indicative of the biphasic reactions during the final stage of the discharge process. As observed in the operando Nb K-edge EXAFS spectra (Figure 6b and 6c), the intensity of the peak associated with Nb–O in *m*-Nb₁₂WO₃₃ significantly decreases during discharge from Li_{3.0}Nb₁₂WO₃₃ to Li_{7.1}Nb₁₂WO₃₃, which is the region

within the phase transition occurs. Therefore, in our opinion, there is no contradiction between the operando XRD and EXAFS discussions.

(6) This sentences about EXAFS changes, especially considering the particle size magnitudinal differences, seems unjustified: "This robustness ensures minimal disruption of the lattice during lithiation, providing stable pathways for Li⁺ ions diffusion and thus better rate capability." EXAFS also does not particularly measure the lattice, but gives local interatomic information. Also, the almost identical sentence is used later in the EXAFS section: "This result reveals a topotactic evolution of the local structure of the tetrahedral sites, arising from the inherent robustness of the disordered structural arrangement present in *dt*-Nb₁₂WO₃₃, which inhibits local structural disruption during lithiation and preserves its overall integrity, and thus contributes to a stable intercalation lattice that facilitates fast Li⁺ ion diffusion." What is topotactic local structure? This is all repeated once more at the end of the EXAFS section: "The data suggests that the disorder of the *dt*-Nb₁₂WO₃₃ structure, introduced by the variability of the ReO₃-type blocks' dimensions and consequent disorder of the tetrahedral tungsten sites position, as well as distortion of those tetrahedra, leads to a robust structure that provides suitable and stable pathways for Li⁺ ions diffusion. As a result, the disorder of the structure allows for Li⁺ fast diffusion and therefore better rate capability than *m*-Nb₁₂WO₃₃, together with higher stability." A few sentences later in the conclusion, it is repeated again: "resulting instead in a topotactic structural evolution, thus allowing higher Li⁺ diffusion rates." Back to a similar sentence in the abstract, "leading to a topotactic local structure evolution during cycling, as determined by operando X-ray diffraction and X-ray absorption spectroscopy.", how can XRD inform local structure evolution?

Reply: We have removed the terms “topotactic” and “lattice” from the manuscript. We have corrected the sentences mentioned by the reviewer, as well as others, and have also revised Figure 7d. The modifications made to the manuscript are shown below.

“This structural arrangement is found to be extremely robust during lithiation/delithiation, leading to relatively minor changes in the local structure during cycling, as determined by operando X-ray absorption spectroscopy.”

“Benefiting from the inherent flexibility of the partially disordered structural arrangement with local distortion, the local structure of $dt\text{-Nb}_{12}\text{WO}_{33}$ was found to experience only slight alterations during Li-ion insertion compared to that of $m\text{-Nb}_{12}\text{WO}_{33}$.”

“This result reveals less significant changes in the local coordination environment around the tungsten atoms, which are mainly in tetrahedral sites, arising from the partially disordered structural arrangement present in $dt\text{-Nb}_{12}\text{WO}_{33}$, which minimizes local structural disruption during lithiation, and thus contributes to a stable intercalation structure (**Figure 7d**).”

“These results further confirm the relatively minor alterations of the coordination environment around the tungsten atoms in $dt\text{-Nb}_{12}\text{WO}_{33}$ during the electrochemical process and the reversibility of those alterations.”

“The unique structure results in wider pathways for Li^+ ions diffusion and minor changes in the local structure during discharge and charge compared to the monoclinic structure ($m\text{-Nb}_{12}\text{WO}_{33}$), providing suitable and stable pathways for faster Li^+ diffusion. The performance of $dt\text{-Nb}_{12}\text{WO}_{33}$ is significantly enhanced with respect to that of $m\text{-Nb}_{12}\text{WO}_{33}$, showing a 44.7 % capacity retention at 80 C and a high initial columbic efficiency of 95.3 %.”

Figure 7. Operando W L_{III} -edge EXAFS characterization during lithiation and delithiation. Operando W L_{III} -edge EXAFS spectra and counter plots for (a) $m\text{-Nb}_{12}\text{WO}_{33}$ and (b) $dt\text{-Nb}_{12}\text{WO}_{33}$ (the corresponding voltage profile for the first cycle is shown on the right side). (c) Variation of the radial distance and peak intensity of the operando W L_{III} -edge EXAFS spectra as a function of

x in $\text{Li}_x\text{Nb}_{12}\text{WO}_{33}$ during lithiation. (d) Schematic representation of the W tetrahedral sites structural evolution for $m\text{-Nb}_{12}\text{WO}_{33}$ and $dt\text{-Nb}_{12}\text{WO}_{33}$ at different discharged states. (e) WT-EXAFS spectra of $m\text{-Nb}_{12}\text{WO}_{33}$ (left side) and $dt\text{-Nb}_{12}\text{WO}_{33}$ (right side) at different discharged states (the blue-yellow-red color gradient represents the increase of the EXAFS intensity).

(7) How do S27 and S28 do what is stated in the following: "The second derivative of the W L_{III} -edge XANES spectra suggest that tungsten remains in tetrahedral coordination in $m\text{-Nb}_{12}\text{WO}_{33}$ and $dt\text{-Nb}_{12}\text{WO}_{33}$ during the discharge (Figures S27, S28)" What is the meaning of this statement?

Reply: The context for this sentence was provided earlier in the manuscript, and we tried to avoid repetition. However, we should have given a more detailed explanation here as well. Therefore, we have revised the discussion to make it clearer, with the changes shown below.

"The second derivative of the W L_{III} -edge XANES spectra remains similar to those of the corresponding pristine materials, showing only one peak, suggesting that tungsten remains mainly in tetrahedral coordination in $m\text{-Nb}_{12}\text{WO}_{33}$ and $dt\text{-Nb}_{12}\text{WO}_{33}$ during discharge (Figures S36, S37). However, there are clear differences in the evolution of the spectra of the materials (Figure 5e-h). For $m\text{-Nb}_{12}\text{WO}_{33}$, the intensity of the edge diminishes during lithiation and the intensity of the post-edge feature also decreases. This can be attributed to the WO_4 and WO_6 polyhedra getting progressively more distorted as the discharge process advances. As a result of the distortion around the W atoms, a decrease in the overlapping between W d-orbitals and O p-orbitals can occur, decreasing the electron density at the metal center. The changes in the spectra are only partially reversible during the charge process. On the contrary, the intensity of the white line in the spectra of $dt\text{-Nb}_{12}\text{WO}_{33}$ does not change significantly compared to $m\text{-Nb}_{12}\text{WO}_{33}$, suggesting less important changes. Less electron density on the tungsten in $dt\text{-Nb}_{12}\text{WO}_{33}$ due to distortion could contribute to the more accentuated decrease of the oxidation state of tungsten in $dt\text{-Nb}_{12}\text{WO}_{33}$ compared to $m\text{-Nb}_{12}\text{WO}_{33}$ at the initial stages of discharge (Figure 5k)."

(8) Reference 19 does not even appear to contain XAS data. "The magnitude of the Nb-Nb and Nb-(O-)Nb peaks decrease continuously, which can be attributed to the shielding effect of Nb-Nb corner sharing interactions during lithium insertion into the lattice¹⁹."

Reply: We apologize for the mistake. The reference is incorrect. The correct one is reference 33 (*Energy Environ. Sci.* **15**, 2022, 254-264). It is discussed in reference 33 that the intensity of the peaks associated with Nb–Nb distances in the Nb K-edge *operando* EXAFS spectra of H-Nb₂O₅ dramatically decreases when the discharge reaches a plateau. Lithium insertion into the interlayer space between corner-shared NbO₆ along the *b* axis increases the shielding effect on the Nb–Nb corner-sharing interactions, resulting in the decrease of the FT intensity. Similarly, the continuous decrease of the intensity of the peaks associated with distances between the metals in the Nb K-edge *operando* EXAFS spectra of *m*-Nb₁₂WO₃₃ and *dt*-Nb₁₂WO₃₃ during lithiation is attributed to a shielding effect on Nb–Nb corner sharing interactions.

The reference has been corrected in the revised manuscript:

“The magnitude of the Nb–Nb and Nb–(O)–Nb peaks decrease continuously, which can be attributed to the shielding effect of Nb-Nb corner sharing interactions during lithium insertion into the structure³³.”

(9) Why does block plane contraction give rise to a new peak rather than a shift in peak position? "It is noted that a new peak corresponding to a Nb-(O-)Nb interatomic distance appears in the spectra at a lower radial distance at 3.1 Å and remains unchanged until the fully discharged state. This new peak arises from the contraction within the block plane."

Reply: We had difficulties assigning this peak and determining its most likely origin. We have revised the analysis of the Nb K-edge EXAFS spectra for *m*-Nb₁₂WO₃₃ and *dt*-Nb₁₂WO₃₃, following the comment of the reviewer. Based on the position of the peak at 3.1 Å, we believe it may represent a contribution from the Nb-W1 scattering path in the second coordination shell of Nb, which is characterized by a relatively low coordination number. This peak becomes noticeable due to a shift in the Nb-Nb radial distance to smaller values, coupled with a decrease in intensity and a shift to higher values of the Nb-(O)-Nb peak.

We have revised the manuscript as described below.

“The weak peak observed in the *operando* Nb K-edge EXAFS spectra at a radial distance at 3.1 Å, from Li_{10.7}Nb₁₂WO₃₃ during the discharge process, may be a contribution from the Nb-W1 radial

distance. This peak becomes noticeable due to a shift in the Nb-Nb radial distance to smaller values, coupled with a decrease in intensity and a shift to higher values of the Nb-(O)-Nb peak.”

Figure 1. Structural characterization of $m\text{-Nb}_{12}\text{WO}_{33}$ and $dt\text{-Nb}_{12}\text{WO}_{33}$. (a) XRD patterns of $m\text{-Nb}_{12}\text{WO}_{33}$ and $dt\text{-Nb}_{12}\text{WO}_{33}$. (b) Schematic illustration of the structure for $m\text{-Nb}_{12}\text{WO}_{33}$ and $dt\text{-Nb}_{12}\text{WO}_{33}$ (blue and green blocks are offset by $1/2 b$ for $m\text{-Nb}_{12}\text{WO}_{33}$ and $1/2 c$ for $dt\text{-Nb}_{12}\text{WO}_{33}$). Nb K-edge (c) and W L_{III}-edge (d) XANES spectra of $m\text{-Nb}_{12}\text{WO}_{33}$ and $dt\text{-Nb}_{12}\text{WO}_{33}$. (e) Fitting of the Nb K-edge first and second coordination shells in the FT-EXAFS spectra of $m\text{-Nb}_{12}\text{WO}_{33}$ and $dt\text{-Nb}_{12}\text{WO}_{33}$. (f) Fitting of the W L_{III}-edge first coordination shell in the FT-EXAFS spectra of $m\text{-Nb}_{12}\text{WO}_{33}$ and $dt\text{-Nb}_{12}\text{WO}_{33}$. (g) PDF pattern of $m\text{-Nb}_{12}\text{WO}_{33}$ and $dt\text{-Nb}_{12}\text{WO}_{33}$. (h) Schematic illustration of type I and VI cavities viewed along the c axis; the atomic pairs Nb-Nb (blue) and W-Nb (red) in the PDF pattern in g are represented by blue and red lines, respectively, in the schematic structure in h.

Figure 6. Operando Nb K-edge EXAFS characterization during lithiation and delithiation. (a) Operando Nb K-edge EXAFS spectra and contour plot of $dt\text{-Nb}_{12}\text{WO}_{33}$ (the corresponding voltage profile for the first cycle is shown on the right side). (b) and (c) Variation of the radial distance and peak intensity of the operando Nb K-edge EXAFS spectra as a function of x in $\text{Li}_x\text{Nb}_{12}\text{WO}_{33}$ during lithiation. (d) Nb K-edge operando EXAFS spectra during the two-phase reaction within the voltage range 1.65-1.70 V. (e) Illustration of the structure of $dt\text{-Nb}_{12}\text{WO}_{33}$ in the OCV and fully lithiated states.

Table S5. Curvefit parameters of Nb K-edge EXAFS of $m\text{-Nb}_{12}\text{WO}_{33}$.

Path	$d / \text{Å}$	N	$R / \text{Å}$	$\sigma^2 / \text{Å}^2$
Nb-O1	1.80	1.5	1.82(1)	0.004(2)
Nb-O2	1.95	2	1.97(1)	0.004(2)

Nb-O3	1.98	2	2.00(1)	0.004(2)
Nb-Nb1	3.36	2	3.37(1)	0.004
Nb-W1	3.60	1	3.61(1)	0.004
Nb-Nb2	3.83	3	3.84(1)	0.004
Nb-O-Nb1	3.85	4	3.84(1)	0.008(1)
Nb-O-Nb2	3.88	4	3.86(1)	0.008(1)
Nb-O-Nb-O	3.85	2	3.84(1)	0.008(1)
Nb-O3	3.92	4	3.87(4)	0.013(8)
O-Nb-O	4.31	4	4.26(4)	0.013(8)
Nb-W2	4.82	1	4.81(16)	0.012(16)

S_0^2 of the first shell for this fit is 0.5. ΔE_0 was refined as a global fit parameter. Data ranges: $2 \leq k \leq 14 \text{ \AA}^{-1}$, $1.52 \leq R \leq 5.2 \text{ \AA}$. The number of variable parameters is 12, out of a total of 28.1 independent data points. R factor for the fitting is 2.0 %. The Debye-Waller factors were constrained as follows to reduce the number variables: $\sigma^2(\text{Nb-O1}) = \sigma^2(\text{Nb-O2}) = \sigma^2(\text{Nb-O3})$, $\sigma^2(\text{Nb-Nb1}) = \sigma^2(\text{Nb-W1}) = \sigma^2(\text{Nb-Nb2})$, $\sigma^2(\text{Nb-O3}) = \sigma^2(\text{O-Nb-O})$, $\sigma^2(\text{Nb-O-Nb1}) = \sigma^2(\text{Nb-O-Nb2}) = \sigma^2(\text{Nb-O-Nb-O})$.

Table S6. Curvefit parameters of Nb K-edge EXAFS of *dt-N₁₂WO₃₃*.

Path	$d / \text{\AA}$	N	$R / \text{\AA}$	$\sigma^2 / \text{\AA}$
Nb-O1	1.80	1.5	1.83(1)	0.004(2)
Nb-O2	1.97	3	2.00(1)	0.004(2)
Nb-Nb1	3.36	4	3.37(1)	0.006(1)
Nb-W1	3.60	1	3.61(1)	0.006(1)
Nb-Nb2	3.83	4	3.85(1)	0.006(1)
Nb-O-Nb1	3.84	4	3.81(1)	0.004
Nb-O-Nb2	3.89	4	3.85(1)	0.004
Nb-O-Nb-O	3.85	2	3.82(1)	0.004
Nb-O3	3.92	4	3.91(6)	0.013(14)

Nb-O4	4.09	4	4.07(6)	0.013(14)
O-Nb-O	4.31	4	4.29(6)	0.013(14)
Nb-W2	4.50	1	4.48(7)	0.009(7)
Nb-W3	4.82	1	4.80(7)	0.009(7)

S_0^2 of the first shell for this fit is 0.5. ΔE_0 was refined as a global fit parameter. Data ranges: $2 \leq k \leq 14 \text{ \AA}$, $1.52 \leq R \leq 5.2 \text{ \AA}$. The number of variable parameters is 12, out of a total of 28.1 independent data points. R factor for the fitting is 2.0 %. The debye-Waller factors were constrained as follows to reduce the number variables: $\sigma^2(\text{Nb-O1}) = \sigma^2(\text{Nb-O2})$, $\sigma^2(\text{Nb-Nb1}) = \sigma^2(\text{Nb-W1}) = \sigma^2(\text{Nb-Nb2})$, $\sigma^2(\text{Nb-O3}) = \sigma^2(\text{Nb-O4}) = \sigma^2(\text{O-Nb-O})$, $\sigma^2(\text{Nb-O-Nb1}) = \sigma^2(\text{Nb-O-Nb2}) = \sigma^2(\text{Nb-O-Nb-O})$ and $\sigma^2(\text{Nb-W1}) = \sigma^2(\text{Nb-W2})$.

(10) Should not changes in the W EXAFS be expected even for the *dt* phase since lattice contraction and expansion are taking place?

Reply: There are changes in the W EXAFS spectra of *dt*- $\text{Nb}_{12}\text{WO}_{33}$, which were discussed in the manuscript. However, these changes are less significant compared to those observed in the spectra of *m*- $\text{Nb}_{12}\text{WO}_{33}$. The discussion of the W EXAFS spectra of *dt*- $\text{Nb}_{12}\text{WO}_{33}$ and *m*- $\text{Nb}_{12}\text{WO}_{33}$ during discharge and charge that were in the manuscript are shown below (highlighted in green).

A similar qualitative analysis was performed for the *operando* W L_{III}-edge FT-EXAFS spectra of *m*- $\text{Nb}_{12}\text{WO}_{33}$ and *dt*- $\text{Nb}_{12}\text{WO}_{33}$ during lithiation/delithiation (**Figure 7a, b, c**). The W–O radial distance in the polyhedra of *m*- $\text{Nb}_{12}\text{WO}_{33}$ is 1.4 Å and remains constant during discharge from the OCV state to $\text{Li}_{3.2}\text{Nb}_{12}\text{WO}_{33}$, accompanied by a decrease of the intensity (**Figure 7a**). The radial distance subsequently decreases gradually from $\text{Li}_{3.2}\text{Nb}_{12}\text{WO}_{33}$ to $\text{Li}_{7.1}\text{Nb}_{12}\text{WO}_{33}$, reaching a value of 1.1 Å, while the intensity sharply declines until the peak completely disappears from $\text{Li}_{2.4}\text{Nb}_{12}\text{WO}_{33}$ to $\text{Li}_{7.1}\text{Nb}_{12}\text{WO}_{33}$ during the two-phase transition reaction. Another peak at a higher radial distance of 2.0 Å starts to appear from $\text{Li}_{5.5}\text{Nb}_{12}\text{WO}_{33}$. Therefore, two types of W–O bond lengths, differing approximately 0.6 Å, with corresponding low amplitude peaks, seem to exist during the two-phase transition reaction within the voltage range 1.65-1.70 V, suggesting that the local tetrahedral environment around the tungsten experiences significant anisotropic distortion,

with a markedly asymmetric distribution of the bond lengths. In contrast, the radial distance of the W–O peak in the *dt*-Nb₁₂WO₃₃ spectra gradually increases from the OCV state to Li_{1.0}Nb₁₂WO₃₃, and then remains constant, while a gradual increase of the peak intensity takes place from Li_{1.0}Nb₁₂WO₃₃ to Li_{3.0}Nb₁₂WO₃₃ during the two-phase transition reaction (Figure 7b, S41). This result reveals less significant changes in the local coordination environment around the tungsten atoms, which are mainly in tetrahedral sites, arising from the partially disordered structural arrangement present in *dt*-Nb₁₂WO₃₃, which minimizes local structural disruption during lithiation, and thus contributes to a stable intercalation structure (Figure 7d). The amplitude of the W–O peak in the *dt*-Nb₁₂WO₃₃ spectra slowly increases upon further discharge to 1.0 V, in contrast with the fast increase observed for *m*-Nb₁₂WO₃₃. The W–O radial distance is larger in the discharged compared to the OCV state for both materials, as a result of the changes in the electron configuration and size of the reduced tungsten ions. A broad and low intensity W–O peak is observed in the *m*-Nb₁₂WO₃₃ spectra during the two-phase reaction region of the delithiation process. This strong local disorder results in an irreversible change of the W–O radial distance for the fully charged state, which may account for the low initial columbic efficiency observed for *m*-Nb₁₂WO₃₃. However, the W–O radial distance of the fully charged *dt*-Nb₁₂WO₃₃ is the same as in the OCV state, indicating a good reversibility of the local structure during the Li⁺ ions insertion and de-insertion.

(11) Reversibility and rate are conflated throughout the manuscript. It is repeatedly said that higher structural reversibility gives higher rate. This does not make sense. Anyway, smaller particles are giving higher rate.

Reply: We agree with the reviewer that structural reversibility and rate are two unrelated concepts. As discussed in the text, the disorder present in the structure of *dt*-Nb₁₂WO₃₃ appears to create wider pathways for Li-ion motion (as suggested by the PDF data), which contributes to the higher rates observed. Additionally, this disorder leads to fewer structural changes during lithium insertion, providing stability to the pathways for fast lithium diffusion. We have identified three sentences in the manuscript that could imply that reversibility leads to higher rates and have removed any mention of rate from those sentences.

Regarding the comment “Anyway, smaller particles are giving higher rate”, we have shown in our reply to comment (1) that, within the particle size range studied here, particle size effects do not

significantly contribute to the differences in diffusion coefficients and rate capabilities observed. Instead, structural differences appear to be the primary factor affecting the performance differences between the two materials, which is consistent with the differences observed in the structural evolution during lithium insertion and extraction.

“This robustness ensures less disruption of the structure during lithiation, providing stable pathways for Li^+ ions diffusion.”

“This result reveals less significant changes in the local coordination environment around the tungsten atoms, which are mainly in tetrahedral sites, arising from the partially disordered structural arrangement present in *dt*- $\text{Nb}_{12}\text{WO}_{33}$, which minimizes local structural disruption during lithiation, and thus contributes to a stable intercalation structure (**Figure 7d**).”

“The unique structure results in wider pathways for Li^+ ions diffusion and minor changes in the local structure during discharge and charge compared to the monoclinic structure (*m*- $\text{Nb}_{12}\text{WO}_{33}$), providing suitable and stable pathways for faster Li^+ diffusion rates. The performance of *dt*- $\text{Nb}_{12}\text{WO}_{33}$ is significantly enhanced with respect to that of *m*- $\text{Nb}_{12}\text{WO}_{33}$, showing a 44.7 % capacity retention at 80 C and a high initial columbic efficiency of 95.3%.”

Reviewer #2 (Remarks to the Author):

Liu et al. report on the synthesis, structural characterization and electrochemical behavior of two forms of $\text{Nb}_{12}\text{WO}_{33}$, the known stable monoclinic phase obtained at high synthesis temperature and a little ordered block-type structure obtained at lower temperature. While *m*- $\text{Nb}_{12}\text{WO}_{33}$ has already been proven to be an excellent fast-charging anode material in previous studies, its capability is exceeded by little ordered *dt*- $\text{Nb}_{12}\text{WO}_{33}$ that shows a strikingly superior Li-uptake and release behavior as well as an increased cycling stability that is convincingly evidenced by charging and discharging profiles, CV curves and rate capability measurements at different capacities. These findings are unexpected and indeed remarkable.

(1) The morphology and structure of both starting samples were comprehensively characterized by powder XRD, SEM + EDXS, HAADF-STEM, XANES and Raman spectroscopy. Structural changes during the first cycle were studied in detail by operando XRD and changes of the oxidation state and coordination of the metals by operando XANES and EXAFS, respectively, and a lot of useful information was collected. The measurements and interpretation of the results were obviously carried out with great care and expertise. Based on the data obtained, the authors conclude that the disorder, in particular that of the W-occupied tetrahedral positions, is responsible for fast Li diffusion and increased rate capability in *dt*- $\text{Nb}_{12}\text{WO}_{33}$. The main weak point of this study is that the interpretation of the data relies on a structure model for *m*- $\text{Nb}_{12}\text{WO}_{33}$ which is likely not fully accurate. The structure determination based on single crystal X-ray data was done by Roth and Wadsley almost 60 years ago (ref. 21). Though this work had certainly been of the utmost quality at that time, the structure determination does not meet today's standards, e. g. because only a part of the data obtained from a tiny needle-shaped crystal was used (h0l and h1l reflections). However, the structure is basically correct apart from the claim that W exclusively occupies the tetrahedral position. This configuration provided a better fit of F_{obs} and F_{calc} in the original study. Moreover, Roth and Wadsley found such a preference of W for the tetrahedral site also for three other block-type niobium tungsten oxides.

These models have been undisputed in literature - and mostly still are - despite contradicting experimental evidence. In 1983, a neutron diffraction study of $\text{Nb}_{14}\text{W}_3\text{O}_{44}$ by Cheetham and Allen unraveled a varying cation distribution: the highest W content was found for the central four octahedra (39 % W) and while the tetrahedral site contains 27 % W only.(1) Based on that observation, it seems likely that W also shows in *m*- $\text{Nb}_{12}\text{WO}_{33}$ a preference for the centers of the

blocks as well as for tetrahedral site while the sites in the CS planes are disfavored but does not occupy 100% of the tetrahedral site.

If the original structural model was correct, HAADF-STEM images of $m\text{-Nb}_{12}\text{WO}_{33}$ would clearly show the tetrahedral position with extreme high brightness. Due to Z contrast with $I \sim Z^2$, a pure W position should be about 3 times brighter than Nb. This is not the case for $dt\text{-Nb}_{12}\text{WO}_{33}$. Note that in HAADF-STEM images of the W-richer phase $\text{Nb}_8\text{W}_9\text{O}_{47}$, the position of a 100% W site appears very bright besides grey ones with mixed occupancy.(2) On the first view on the HAADF-STEM images of $dt\text{-Nb}_{12}\text{WO}_{33}$ (Figures 2 and S8), the intensities of the metal positions appear to be differing in a small intensity range and rather randomly. However, there are some recognizable trends. That the tetrahedral sites in combination with their close neighbors appear as bright patches in thicker regions of Figure 2 is owing to a higher density of scattering centers there (i.e. smaller M-M distances) than inside the blocks. The analysis of the intensity distribution in thin areas by line intensity profiles (Figure R1) shows that (i) the positions inside the blocks tend to be generally brighter than those in the shear planes and (ii) the positions in the tetrahedral site are equally bright or not much brighter than the surrounding ones and the ones in the block centers (Figure 2e, Line 5 in Figure R1). This is not only the case for the defective block structure investigated here but also for the perfect structure of $\text{Nb}_{14}\text{W}_3\text{O}_{44}$ (cf. Figures 1f in (3) or 2h in ref. 7). A HAADF-STEM image of $m\text{-Nb}_{12}\text{WO}_{33}$ along [010] (unfortunately only the direction [1-10] is investigated here (Figure S2)) would reveal the discrepancy from the idealized model (ref 21) most pronounced as the tetrahedral position is supposed to be 100% W while the others are 100 % Nb. The interpretation of the operando results and the conclusions concerning the role of the tetrahedral site for improved electrochemical behavior of $dt\text{-Nb}_{12}\text{WO}_{33}$ are based on the classic structure model for $m\text{-Nb}_{12}\text{WO}_{33}$ that appears to be wrong in respect of the occupancy of the tetrahedral and octahedral positions by W and Nb. As a correct determination of the W distribution in $m\text{-Nb}_{12}\text{WO}_{33}$ is indispensable for an accurate evaluation of the experimental results, I encourage the authors to try the measurement of single crystal X-ray data for an up-to-date structure determination. They have grown well-developed crystals of $m\text{-Nb}_{12}\text{WO}_{33}$ (Figure S3) that are small but possibly suitable nonetheless. Such a study would complement the insights gained by HAADF-STEM images of this structure along [010]. It should be noted that the structure of $dt\text{-Nb}_{12}\text{WO}_{33}$ is closely related if not identical to that of $M\text{-Nb}_2\text{O}_5$, a metastable Nb_2O_5 polymorph obtained at 900 °C as well.(4) It is tetragonal with the lattice constants being $a = 20.44\text{\AA}$, $c = 3.832\text{\AA}$ which are very similar to those of $dt\text{-Nb}_{12}\text{WO}_{33}$ (cf. Figure S1b). In the first article by Mertin et al., a simple structure with [4x4]

blocks was suggested but a subsequent HRTEM study by Heurung und Gruehn revealed a little ordered structure with differing block sizes like that now observed for *dt*-Nb₁₂WO₃₃.⁽⁵⁾ Moreover, this sample was prepared with an addition of a small amount of WO₃ which apparently stabilizes the M modification. It is interesting to note that in Figure S8 at the bottom of the central part, there is a domain with [4x4] blocks corresponding to the original suggestion for the structure of M-Nb₂O₅. Thus, the present investigation unknowingly confirms this structural model after more than half a century – an amazing finding.

Reply: Following the reviewer suggestions, we have prepared larger crystals of *m*-Nb₁₂WO₃₃ and performed single crystal measurements. The new data and corresponding discussion are shown below. We have also made changes to the discussion according to the comments of the reviewer on the similarities between the structures of *m*-Nb₁₂WO₃₃ and M-Nb₂O₅. These modifications are shown below. Moreover, as suggested by the reviewer, we have added additional line intensity analysis of thin areas of the *dt*-Nb₁₂WO₃₃ HAADF-STEM images. The modified Figure 2 is shown below, together with the additional discussion included in the revised manuscript.

Changes made to the discussion and SI:

“X-ray diffraction studies by Roth and Wadsley²¹ and Density Functional Theory (DFT) calculations reported by Koçer et al.¹⁰ suggest that tungsten strongly prefers to occupy the tetrahedral sites at the corners of the blocks. On the other hand, Cheetham and Allen²² found also a strong preference of the tungsten for the octahedral site at the center of the blocks in Nb₁₄W₃O₄₄ using neutron diffraction data. To determine the structure of *m*-Nb₁₂WO₃₃, we have synthesized larger crystals (**Figure S2**) and performed single crystal XRD measurements. The crystallographic parameters obtained from the structure refinement are listed in **Table S1**. The results with a R factor of 3.83 % gave the lattice parameters $a = 22.3002(15)$ Å, $b = 3.8279(2)$ Å, $c = 17.7490(12)$ Å and $\beta = 123.338$ °. According to the refinement data in **Table S1**, the tungsten atoms show a preference for the tetrahedral sites at the corners of the blocks, with a 100 % occupancy. We have tried to replace the Nb1 site at the center of the blocks with a small percentage of tungsten atoms but the refinement was not stable. However, very recently, Cardon et al.²³ have prepared high-quality centimeter sized *m*-Nb₁₂WO₃₃ crystals for structure determination. The structural refinement, with a R factor of 3.62 %, identified a higher symmetry space group (I2/m) than

previously reported (C2/m). Roth and Wadsley²¹ reported 100 % Nb occupancy of the octahedral sites and 100 % W occupancy of the tetrahedral sites. The results of Cardon et al.²³ show that tungsten has a strong preference for the tetrahedral sites, with a similar overall distribution of W and Nb as that obtained by Roth and Wadsley, except for the octahedral site located at the center of the blocks, for which the refinement indicates a 7 % W occupancy. Therefore, tungsten mainly occupies the tetrahedral sites in *m*-Nb₁₂WO₃₃, with a small percentage additionally occupying the octahedral site at the center of the blocks. The same authors also found that, while the tetrahedral sites should be exclusively occupied by W in Nb₁₄W₃O₄₄, tungsten also has a relatively strong preference for the octahedral site at the center of the blocks, with a 26 % W occupancy.”

“The diffractogram of *dt*-Nb₁₂WO₃₃ (**Figure 1a**) reveals a new crystalline phase for the Nb₁₂WO₃₃ block structure, which has not been previously reported. The pattern is consistent with a tetragonal structure (space group *I4/mmm*), similar to that of M-Nb₂O₅^{24,25}.”

“While *m*-Nb₁₂WO₃₃ consists of 4x3 blocks of corner-shared octahedra, the structure of *dt*-Nb₁₂WO₃₃ contains blocks in a variety of sizes, such as 4x4, 4x3, 5x3, 6x3, 5x4, and 3x3, thus resulting in the non-uniform distribution of the WO₄ (**Figure S9**). A similar variation in block size has been reported several decades ago for M-Nb₂O₅ stabilized with WO₃³⁶.”

“The intensity profiles in **Figure 2h** corresponding to Line1 and Line2 in (e) show that the intensity of the block centers is generally higher than that in the crystallographic shear (CS) planes. In addition, although the positions associated with the tetrahedral sites in **Figure 2f** are brighter than those in the CS planes, the differences are not as high as expected if the tetrahedral sites were fully occupied with tungsten atoms³⁷. Within the thin area at the edge of the particle delimited by Line3 in (e), the tetrahedral site shows slightly higher intensity compared with the octahedral sites at the center of the blocks (**Figure 2h**). These results suggest that tungsten atoms predominantly occupy the tetrahedral sites, with some possible occupation of the octahedral sites at the center of the blocks, with similarities to the Nb and W distribution in *m*-Nb₁₂WO₃₃ discussed above.”

Changes made to the experimental section:

“Larger crystals of *m*-Nb₁₂WO₃₃ was synthesized via a molten flux method^{1,2}. Stoichiometric amounts of Nb₂O₅ (0.5 mmol) and WO₃ (0.083 mmol) were mixed by manually grinding, and subsequently H₃BO₃ (5 mmol) was added and mixed with the oxide precursors. The powder mixture was placed in an alumina crucible, heated at 1050 °C for 10 h in a muffle oven, with a heating rate of 150 °C h⁻¹ to reach the final temperature. After cooling to room temperature with a cooling rate of 60 °C h⁻¹, the powder was washed with water to remove the remaining H₃BO₃.”

“The single crystal X-ray diffraction data was collected with a BRUKER D8 VENTURE area detector with Mo-K radiation ($\lambda = 0.71073$). The structure refinement was conducted with Shelx⁵.”

Changes made to the reference section:

References:

- 22 Cheetham, A. K., & Allen, N. C. Cation distribution in the complex oxide, W₃Nb₁₄O₄₄; a time-of-flight neutron diffraction study. *J. Chem. Soc., Chem. Commun.* 1370-1372 (1983).
- 23 Cardon, P. et al. Intrinsic structures and electrochemical properties of floating zone grown Nb₁₂WO₃₃ and Nb₁₄W₃O₄₄ single crystals. *Cryst. Growth Des.* **25**, 1265-1275 (2025).
- 24 Mertin, W., Anderson, S., & Gruehn, R. Über die Kristallstruktur von M-Nb₂O₅. *J. Solid State Chem.* **1**, 419-424 (1970).
- 25 Ding, H. et al. Controlled synthesis of pure-phase metastable tetragonal Nb₂O₅ anode material for high-performance lithium batteries. *J. Solid State Chem.* **299**, 122136 (2021).
- 36 Heurung, G., & Gruehn, R. Beiträge zur Untersuchung anorganischer nichtstöchiometrischer Verbindungen. XVIII. Neues zur Struktur und Stabilität von M-Nb₂O₅. *Z. anorg. allg. Chem.* **491**, 101-112 (1982).
- 37 Kirkland, A. I., & Saxton, W. O. Cation segregation in Nb₁₆W₁₈O₉₄ using high angle annular dark field scanning transmission electron microscopy and image processing. *J. Microsc.* **206**, 1-6 (2002).

References in SI:

- 1 Ftini, M. M., Krifa, M. & Amor, H. KM_{0.5}O₁₃ and KNb_{1.76}Sb_{3.24}O₁₃. *Acta Cryst.* **58**, i106-i108 (2002).
- 2 Kawashima, K. et al. Chloride flux growth of idiomorphic AWO₄ (A= Sr, Ba) single microcrystals. *Cryst. Growth Des.* **18**, 5301–5310 (2018).

Figure S2. (a) XRD pattern and (b) TEM images of the larger $m\text{-Nb}_{12}\text{WO}_{33}$ crystals.

Figure 2. AC-STEM study of $dt\text{-Nb}_{12}\text{WO}_{33}$. (a) and (b) AC-STEM images of $dt\text{-Nb}_{12}\text{WO}_{33}$ (metals in octahedral sites and tetrahedral sites are indicated by blue and orange spheres, respectively). (c - e) Magnified AC-STEM images of the regions I, II and III delimited in (b). (f - h) Line intensity profiles from the regions delimited in (c), (d) and (e), respectively.

Table S1. Crystallographic parameters for $m\text{-Nb}_{12}\text{WO}_{33}$.

chemical formula	Nb _{1.50} W _{0.12} O _{4.12}
crystal system	monoclinic
space group	C 2/m
a (Å)	22.3002(15)
b (Å)	3.8279(2)
c (Å)	17.7490(12)
α (deg)	90
β (deg)	123.338(2)
V (Å ³)	1265.78(14)
Z	16
density (calculated) (g/cm ³)	4.793
Temp. (K)	101(2)
abs. coeff. (mm ⁻¹)	9.865
F (0 0 0)	1660
Theta range for data collection (deg)	2.186 to 27.521
reflections collected	14015
Independent reflections	1664 [R(int) = 0.0825]
Completeness to theta = 25.242°	99.6%
refinement method	full-matrix least-squares on F ²
data / restraints / parameters	1664 / 0 / 92
goodness-of-fit on F ²	1.038
final R indices [I>2sigma(I)]	R1 = 0.0383, wR2 = 0.0851
R indices (all data)	R1 = 0.0484, wR2 = 0.0932
largest diff. peak and hole (Å ⁻³)	1.820/-1.590

	x (Å)	y (Å)	z (Å)	occ.	U _{iso} (Å ²)
Nb1	0.41697(4)	0	0.38056(5)	1	0.0145(2)
Nb2	0.46442(4)	0	0.71313(5)	1	0.00531(16)
Nb3	0.29725(4)	0	0.47235(5)	1	0.00553(16)
Nb4	0.24819(4)	0	0.13672(5)	1	0.00485(16)

Nb5	0.12922(4)	0	0.23061(5)	1	0.00460(16)
Nb6	0.63319(4)	0	0.95611(5)	1	0.00455(16)
W1	0	-0.2500(2)	0	0.4438(16)	0.0049(2)
O1	0.6975(3)	0	0.9200(4)	1	0.0047(12)
O2	0.5250(3)	0	0.6749(4)	1	0.0071(12)
O3	0.3508(3)	0	0.4238(4)	1	0.0098(13)
O4	0.1785(3)	0	0.1766(4)	1	0.0068(12)
O5	0.5496(3)	0	0.8452(4)	1	0.0036(11)
O6	0.3777(3)	0	0.5959(4)	1	0.0071(12)
O7	0.2052(3)	0	0.3488(4)	1	0.0099(13)
O8	0.3250(3)	0	0.2493(4)	1	0.0098(13)
O9	0.5	0	0.5	1	0.0126(19)
O10	0.2779(3)	-0.5	0.4805(4)	1	0.0053(12)
O11	0.4515(3)	-0.5	0.7301(4)	1	0.0035(11)
O12	0.0320(3)	0	0.0991(4)	1	0.0067(12)
O13	0.0733(3)	-0.5	0.0168(4)	1	0.0053(12)
O14	0.1039(3)	-0.5	0.2317(4)	1	0.0073(12)
O15	0.2249(3)	-0.5	0.1039(4)	1	0.0065(12)
O16	0.6496(3)	-0.5	0.9922(4)	1	0.0077(12)
O17	0.4105(3)	-0.5	0.3705(4)	1	0.0123(14)

(2) The term “disordered” as used in the title and at different places in the manuscript is somewhat misleading, because the order is perfect along the c axis whereas the arrangement of blocks with varying sizes in the ab plane indeed is little ordered.

Reply: We agree with the reviewer that the term *disordered* is misleading, as the material is not completely disordered. We revised the title and text throughout the manuscript by replacing the term *disordered* with *partially disordered*. The related contents are revised as below.

“A Partially Disordered Crystallographic Shear Block Structure as Fast-Charging Anode Material for Lithium-Ion Batteries”

“Here, we report a new anomalous partially disordered $\text{Nb}_{12}\text{WO}_{33}$ structure that significantly enhances the Li-ion storage performance compared to the known monoclinic $\text{Nb}_{12}\text{WO}_{33}$ phase. The partially disordered $\text{Nb}_{12}\text{WO}_{33}$ phase consists of corner-shared NbO_6 octahedra blocks of varied sizes, including 5×4 , 4×4 , and 4×3 , with a disordered arrangement of the tungsten tetrahedra at the corners of the blocks, as well as distortion of the WO_4 tetrahedra.”

“It leads also to accelerated Li-ion migration within the partially disordered phase that results in excellent fast-charging performance, namely, 62.5 % and 44.7 % capacity retention at 20 C and 80 C, respectively.”

“Keywords: niobium tungsten oxides, partially disordered distribution, high rate, lithium-ion batteries; operando studies.”

“By controlling the synthesis conditions, a partially disordered $\text{Nb}_{12}\text{WO}_{33}$ phase (*dt*- $\text{Nb}_{12}\text{WO}_{33}$) can be obtained, whose structure is characterized by the presence of shear blocks in a variety of sizes, an irregular distribution of the tetrahedral sites within the structure, and strong local distortions. Benefiting from the inherent flexibility of the partially disordered structural arrangement with local distortion, the local structure of *dt*- $\text{Nb}_{12}\text{WO}_{33}$ was found to experience only slight alterations during Li-ion insertion compared to that of *m*- $\text{Nb}_{12}\text{WO}_{33}$.”

“In addition, the partially disordered structure adaptability ensures the presence of optimized channels for Li-ion transport during lithiation and delithiation, enhancing the rate performance.”

“Structure of the partially disordered $\text{Nb}_{12}\text{WO}_{33}$ ”

“Contrary to what is observed for $m\text{-Nb}_{12}\text{WO}_{33}$, the brighter spots (some delimited by circles in image a), corresponding to tungsten atoms in tetrahedral coordination, are heterogeneously distributed, reflecting the partially disordered nature of the structure.”

“The PDF peaks in the long-range ($> 10 \text{ \AA}$) of $dt\text{-Nb}_{12}\text{WO}_{33}$ are broader and less intense than those of the monoclinic structure, which is consistent with the presence of partial structural disorder.”

“The excellent rate capability and reduced voltage polarization of $dt\text{-Nb}_{12}\text{WO}_{33}$ are a consequence of its superior reaction kinetics and smaller diffusion barriers for Li^+ ions across multiple sites within the partially disordered structure.”

“Less significant changes are observed in the first peak for $dt\text{-Nb}_{12}\text{WO}_{33}$ compared to that of $m\text{-Nb}_{12}\text{WO}_{33}$, meaning that much smaller structural changes occur in the first oxygen coordination shell around the niobium in the former, which reflects the robustness of the partially disordered phase.”

“The data suggests that the partial disorder of the $dt\text{-Nb}_{12}\text{WO}_{33}$ structure, introduced by the variability of the ReO_3 -type blocks' dimensions and consequent disorder of the tetrahedral tungsten sites position, as well as distortion of those tetrahedra, leads to a robust structure that provides suitable and stable pathways for Li^+ ions diffusion. As a result, the partial disorder of the structure allows for Li^+ fast diffusion and therefore better rate capability than $m\text{-Nb}_{12}\text{WO}_{33}$, together with higher stability.”

“A partially disordered $\text{Nb}_{12}\text{WO}_{33}$ phase ($dt\text{-Nb}_{12}\text{WO}_{33}$) has been obtained for the first time, which consists of ReO_3 -type blocks of varied sizes with more spacious channels for ion diffusion, irregular arrangement of the tetrahedral sites, and strong distortion of those sites, for application as anode material in Li-ion batteries.”

(3) Additional HAADF-STEM images of $m\text{-Nb}_{12}\text{WO}_{33}$ after several cycles would provide valuable information about the preservation of the block structure during electrochemical reaction.

Reply: As suggested by the reviewer, we have obtained HAADF-STEM images of lithiated $m\text{-Nb}_{12}\text{WO}_{33}$ and $dt\text{-Nb}_{12}\text{WO}_{33}$ after cycling, and modified the manuscript and supplementary information accordingly. The changes made are described below.

“HAADF-STEM images of lithiated $dt\text{-Nb}_{12}\text{WO}_{33}$ and $m\text{-Nb}_{12}\text{WO}_{33}$ were taken to determine potential changes in the block structure caused by lithium insertion (Figures S31 and S32, respectively). The images of lithiated $dt\text{-Nb}_{12}\text{WO}_{33}$ show similar disorder to that of the pristine material, with blocks of different sizes, including 5x4, 4x4, and 4x3. The atomic arrangement of niobium and tungsten observed in the images of lithiated $m\text{-Nb}_{12}\text{WO}_{33}$ matches the monoclinic structure along the a -axis, and indicate the retention of the block structure.”

Changes made to the experimental section:

“The electrodes for ex-situ AC-STEM were washed with dimethyl carbonate (DMC) for three times, and then dried in vacuum.”

Figure S31. AC-STEM images of lithiated $dt\text{-Nb}_{12}\text{WO}_{33}$ after the second cycle (metals in octahedral sites and tetrahedral sites are indicated by blue and orange spheres, respectively).

Figure S32. AC-STEM images of lithiated $m\text{-Nb}_{12}\text{WO}_{33}$ after the second cycle (metals in octahedral sites and tetrahedral sites are indicated by blue and orange spheres, respectively).

Further comments:

(4) Page 4: “The pattern profile is best fitted with a tetragonal cell (space group $I4/mmm$), although the atomic positions cannot be properly refined due to the presence of disorder in the structure (Figure S1).” How was the Rietveld refinement of the XRD pattern possible without a plausible structural model for $dt\text{-Nb}_{12}\text{WO}_{33}$?

Reply: We used the structure of $M\text{-Nb}_2\text{O}_5$ as a model, which, as mentioned by the reviewer, is closely related to that of $dt\text{-Nb}_{12}\text{WO}_{33}$. However, a reliable refinement could not be achieved and atomic positions could not be refined due to the presence of disorder in the structure. This was not clear in the manuscript. Therefore, we have replaced the above sentence with the one below in the revised manuscript. In addition, Figure S1 of the SI has been modified to include only the Rietveld refinement results for $m\text{-Nb}_{12}\text{WO}_{33}$.

“The pattern is consistent with a tetragonal structure (space group $I4/mmm$) similar to that of $M\text{-Nb}_2\text{O}_5$ ^{24,25}.”

References:

24 Mertin, W., Anderson, S., & Gruehn, R. Über die Kristallstruktur von $M\text{-Nb}_2\text{O}_5$. *J. Solid State Chem.* **1**, 419-424 (1970).

25 Ding, H. et al. Controlled synthesis of pure-phase metastable tetragonal Nb_2O_5 anode material for high-performance lithium batteries. *J. Solid State Chem.* **299**, 122136 (2021).

(5) Page 4-5: The descriptions of Figures S4b and S4e are mixed up.

Reply: We apologize for the mistake. We have corrected it.

“The high resolution TEM (HR-TEM) image of $dt\text{-Nb}_{12}\text{WO}_{33}$ in **Figure S5e** shows lattice fringes with a spacing of 0.28 nm, matching the d-spacing of the (501) planes of $dt\text{-Nb}_{12}\text{WO}_{33}$. The (260), (442) and (2-22) planes in the selected area electron diffraction (SAED) pattern of $dt\text{-Nb}_{12}\text{WO}_{33}$ are typically observed in the diffraction pattern of a tetragonal ($I4/mmm$) structure. The lattice fringes with a spacing of 0.36 nm observed in **Figure S5b** correspond to the (203) planes of the monoclinic structure of $m\text{-Nb}_{12}\text{WO}_{33}$, which is further confirmed through the SAED pattern.”

(6) Page 9: "...Li⁺ diffusion through tetrahedral site cavities..." Is this possible? These sites should be fully occupied by W and Nb.

Reply: In the section mentioned by the reviewer, we discuss the cavity sites available in the structure for Li-ion insertion. These sites differ from those occupied by the metal atoms. Instead, they represent cavities formed between the metal sites in certain regions of the structure. We recognize that our previous description may have been misleading. Therefore, we have rewritten this section and added further discussion on the pathways for Li-ions within the structure, as well as on how the partial disorder in *dt*-Nb₁₂WO₃₃ affects ion diffusion. The revised text is provided below.

"The types of cavity sites for Li-ion insertion present in Wadsley-Roth phases of Nb-based oxides were first categorized by Cava et al.³⁸ The structure of *dt*-Nb₁₂WO₃₃ possesses cavity sites of the types I, II, III, V, and VI, that provide potential pathways for Li-ion diffusion. The position of these cavities in the structure is schematized in **Figure S21**. Koçer et al.¹¹ have performed DFT calculations to determine the energetically favorable pathways for Li⁺-ion motion within niobium tungsten oxide block structures. Li⁺ diffusion was found to be energetically unfavorable across shear planes and through type VI cavities, which are formed by tetrahedrally coordinated tungsten at the junctions of the blocks and octahedral sites in the vertices of the blocks (**Figure 1b and Figure S21**). In contrast, Li-ion diffusion through the sites within the blocks was found to involve lower energy barriers, making these the most likely pathways in the block structures. As suggested by the PDF data, the partial disorder in *dt*-Nb₁₂WO₃₃ shortens the distances between W in tetrahedral sites and Nb in octahedral sites at the corner of the blocks, making it very unlikely for Li-ions to diffuse via type VI cavities. However, disorder also leads to larger distances between the metal sites within the blocks, indicating wider pathways for lithium diffusion inside the blocks of *dt*-Nb₁₂WO₃₃, which contributes to the larger Li⁺ diffusion coefficients and better rate capability of *dt*-Nb₁₂WO₃₃ compared to *m*-Nb₁₂WO₃₃."

(7) Figure 3g: The figure caption is inaccurate as "Nb₁₈W₁₆O₉₃" is not a block structure.

Reply: Thank you for pointing the mistake to us. We have revised the figure and the caption, and have also included in the revised manuscript a comparison with a broader range of other Nb-based

anode materials from the literature, including crystallographic shear structures and tungsten bronze structures.

We have revised Figure 3g and Table S7 of the supplementary information accordingly, and added the additional references, as shown below.

“*dt-Nb₁₂WO₃₃* exhibits superior electrochemical performance compared with the previously reported Nb-based anode materials (Figure 3g and Table S7)^{5,6,13,15,42,43,44,45}”

The related figure and table are shown below:

Figure 3. Li-ion storage properties and diffusion kinetics in *m-Nb₁₂WO₃₃* and *dt-Nb₁₂WO₃₃*. (a) Charge and discharge profiles of *m-Nb₁₂WO₃₃* and *dt-Nb₁₂WO₃₃* at 0.5 C. (b) CV curves of *m-Nb₁₂WO₃₃* and *dt-Nb₁₂WO₃₃* at 0.1 mV s⁻¹. (c) Rate capability of *m-Nb₁₂WO₃₃* and *dt-Nb₁₂WO₃₃*. (d) Linear relationship between the peak current (*i_p*) and the square root of the scan rate (*v*^{1/2}) for peak O1. (e) GITT profiles of *m-Nb₁₂WO₃₃* and *dt-Nb₁₂WO₃₃* as a function of *x* in Li_{*x*}Nb₁₂WO₃₃. (f) Variation of the *D_{Li+}* determined by GITT. (g) Comparison of the rate capability between *dt-Nb₁₂WO₃₃* and other reported Nb-based oxides with block structures. (h) Cycling performance of *dt-Nb₁₂WO₃₃* at 10 C.

Table S7. Comparison of the electrochemical performance of *dt*-Nb₁₂WO₃₃ with that of other previously reported Nb-based materials.

Material	Structure type	Voltage range (V vs. Li ⁺ /Li)	Initial reversible capacity (mAh g ⁻¹)	Rate performance (mAh g ⁻¹)	Current density of rate performance (A g ⁻¹)
Nb ₁₆ W ₅ O ₅₅ ⁹	Shear structure	1.0-3.0	225 (at 34.3 mA g ⁻¹)	~50	10.3
Nb ₁₈ W ₁₆ O ₉₃ ⁹	Tungsten bronze structure	1.0-3.0	205 (at 29.8 mA g ⁻¹)	70	14.9
Nb ₁₄ W ₃ O ₄₄ ¹⁰	Shear structure	1.0-3.0	221.3 (at 89 mA g ⁻¹)	84.4	8.9
Nano-block Nb ₁₄ W ₃ O ₄₄ ¹¹	Shear structure	1.0-3.0	241.1 (at 89 mA g ⁻¹)	109.5	14.2
Nb ₁₂ WO ₃₃ nanowires ¹²	Shear structure	1.0-3.0	228 (at 200 mA g ⁻¹)	145.8	0.7
Nb ₁₈ W ₈ O ₆₉ ¹³	Shear structure	1.0-3.0	~230 (at 32.8 mA g ⁻¹)	~30	9.8
H-Nb ₂ O ₅ ¹⁴	Shear structure	1.0-3.0	269 (at 20 mA g ⁻¹)	94	4
PNb ₉ O ₂₅ ¹⁵	Shear structure	1.0-3.0	~235 (at 25.4 mA g ⁻¹)	30	15.2
VNb ₉ O ₂₅ ¹⁵	Shear structure	1.0-3.0	~180 (at 29.2 mA g ⁻¹)	25	11.7
TiNb ₂₄ O ₆₂ ¹⁶	Shear structure	1.0-3.0	214 (at 20.5 mA g ⁻¹)	~100	3.1
V ₇ Nb ₆ O ₂₉ ¹⁷	Shear structure	1.2-3.3	208 (at 20 mA g ⁻¹)	106	2
Ba _{3.4} Nb ₁₀ O _{28.4} ¹⁸	Tungsten bronze structure	0.8-3.0	167 (at 28.97 mA g ⁻¹)	82	2.9
Nb ₁₂ W ₁₁ O ₆₃ ¹⁹	Tungsten bronze structure	1.3-3	176 (at 22.6 mA g ⁻¹)	100	4.5
CoNb ₁₁ O ₂₉ ²⁰	Shear structure	0.8-3	265 (at 40 mA g ⁻¹)	113	4
Nb ₁₂ O ₂₉ ²¹	Shear structure	1.0-2.5	243 (at 6.6 mA g ⁻¹)	60	0.132
dt -Nb ₁₂ WO ₃₃ (This work)	Shear structure	1-3	257 (at 95.4 mA g ⁻¹)	119.5	15.3

References:

43. Yang, M., Li, S., Huang J. Three-dimensional cross-linked Nb₂O₅ polymorphs derived from cellulose substances: Insights into the mechanisms of Lithium storage. *ACS Appl. Mater. Interfaces* **13**, 39501–39512 (2021).
44. Preefer, M. B. et al. Multielectron redox and insulator-to-metal transition upon lithium insertion in the fast-charging, Wadsley-Roth phase PNB₉O₂₅. *Chem. Mater.* **32**, 4553–4563 (2020).
45. Griffith, K. J., Senyshyn A., Grey, C. P. Structural stability from crystallographic shear in TiO₂–Nb₂O₅ phases: cation ordering and lithiation behavior of TiNb₂₄O₆₂. *Inorg. Chem.* **56**, 4002–4010 (2017).

References in SI:

9. Griffith, K. J., Wiaderek, K. M., Cibir, G., Marbella, L. E. & Grey, C. P. Niobium tungsten oxides for high-rate lithium-ion energy storage. *Nature* **559**, 556-563 (2018).
10. Yang, Y. et al. Achieving Achieving ultrahigh-rate and high-safety Li⁺ storage based on interconnected tunnel structure in micro-size niobium tungsten oxides. *Adv. Mater.* **32**, 1905295 (2020).
11. Guo, C. et al. Nano-sized niobium tungsten oxide anode for advanced fast-charge lithium-ion batteries. *Small* **18**, 2107365 (2022).
12. Yan, L. et al. Electrospun WNB₁₂O₃₃ nanowires: superior lithium storage capability and their working mechanism. *J. Mater. Chem. A* **5**, 8972-8980 (2017).
13. Griffith, K. J., & Grey, C. P. Superionic lithium intercalation through 2 × 2 nm² columns in the crystallographic shear phase Nb₁₈W₈O₆₉. *Chem. Mater.* **32**, 3860-3868 (2020).
14. Yang, M., Li, S., Huang J. Three-dimensional cross-linked Nb₂O₅ polymorphs derived from cellulose substances: Insights into the mechanisms of Lithium storage. *ACS Appl. Mater. Interfaces* **13**, 39501–39512 (2021).
15. Preefer, M. B. et al. Multielectron redox and insulator-to-metal transition upon lithium insertion in the fast-charging, Wadsley-Roth phase PNB₉O₂₅. *Chem. Mater.* **32**, 4553–4563 (2020).
16. Griffith, K. J., Senyshyn A., Grey, C. P. Structural stability from crystallographic shear in TiO₂–Nb₂O₅ phases: cation ordering and lithiation behavior of TiNb₂₄O₆₂. *Inorg. Chem.* **56**, 4002–4010 (2017).

17. Lawrence, E. A. et al. Reversible electrochemical lithium cycling in vanadium(IV)- and niobium(V)-based Wadsley–Roth phase. *Chem. Mater.* **35**, 3470–3483 (2023).
18. Xiong, X. et al. Cation-vacancy ordered superstructure enhanced cycling stability in tungsten bronze anode. *Adv. Energy Mater.* **12**, 2201967 (2022).
19. Ma, X.-H., et al. Influence of cut-off voltage on the lithium storage performance of $\text{Nb}_{12}\text{W}_{11}\text{O}_{63}$ anode, *Electrochim. Acta* **332**, 135380 (2020).
20. Liu, H., Chen, C. Wadsley–Roth phase $\text{CoNb}_{11}\text{O}_{29}$ as a high-performance anode for lithium-ion batteries. *J. Mater. Chem. A*, **12**, 5414-5421 (2024).
21. Li, Y., Sun, C., Goodenough, J. B. Electrochemical Lithium Intercalation in Monoclinic $\text{Nb}_{12}\text{O}_{29}$. *Chem. Mater.* **23**, 2292–2294 (2011).

(8) Figure 4a,b: The reflection at $2\theta \approx 16^\circ$ is not explained.

Reply: The reflections at $2\theta \approx 16.1$ and 16.4° are from the (800) plane of $dt\text{-Nb}_{12}\text{WO}_{33}$ and the lithium foil that was used as counter electrode in the *operando* cell, respectively. We included an explanation for these reflections in the revised manuscript (see below).

“The diffractogram of the pristine $dt\text{-Nb}_{12}\text{WO}_{33}$ (**Figure 4b**) exhibits reflections associated to the planes (101), (440), (431), (800), (701), (770) and (761).”

“The reflection at $2\theta \approx 16.4^\circ$ is attributed to the lithium metal (**Figure S29**), which was used as counter electrode in the *operando* cell.”

“The positions of the (440) and (770) reflections shift to lower angles from the initial state to $\text{Li}_{3.8}\text{Nb}_{12}\text{WO}_{33}$, as a result of the expansion of the a - b plane by 0.8%. The (800) reflection ($2\theta \approx 16.1^\circ$) shifts slightly to lower angles due to the expansion of the a axis.”

Figure 4. Long-range structure evolution of $m\text{-Nb}_{12}\text{WO}_{33}$ and $dt\text{-Nb}_{12}\text{WO}_{33}$ during lithiation and delithiation. Operando XRD patterns of (a) $m\text{-Nb}_{12}\text{WO}_{33}$ and (b) $dt\text{-Nb}_{12}\text{WO}_{33}$ electrodes, collected for the first discharge and charge processes. The corresponding XRD contour plots at selected angle ranges are shown on the left. (c) Unit cell parameters evolution during Li^+ insertion and extraction for the $dt\text{-Nb}_{12}\text{WO}_{33}$ electrode.

Figure S29. XRD pattern of lithium metal.

(9) Supporting Information;

Materials characterization: the types and characteristics of the two different microscopes used for S/TEM should be presented together.

Reply: We have corrected the “Materials characterization” section of the SI to present the two microscopes together, as shown below.

“Transmission electron microscopy (TEM), high-resolution TEM (HRTEM), high-angle annular dark-field scanning transmission electron microscopy (HAADF-STEM) and energy dispersive X-ray (EDX) analysis elemental mapping were conducted on a FEI Talos F200S scanning/transmission electron microscope (S/TEM) at an acceleration voltage of 200 kV. Aberration-corrected STEM (AC-STEM) analysis was performed using a FEI Titan Themis Cubed STEM microscope (FEI Company), equipped with a Cs probe corrector and operated at 300 kV. HAADF images were acquired with collection angles between 66–200 mrad to improve Z-contrast imaging”

References:

1. Cheetham, Allen. Cation Distribution in the Complex Oxide, $W_3Nb_{14}O_{44}$; a Time-of-Flight Neutron Diffraction Study. *J. Chem. Soc., Chem. Commun.* 1983, 1370-1372.

2. Kirkland, Saxton. Cation Segregation in $\text{Nb}_{16}\text{W}_{18}\text{O}_{94}$ Using High Angle Annular Dark Field Scanning Transmission Electron Microscopy and Image Processing. *J. Microsc.* 2002, 206, 1–6.
3. Yang et al. Achieving Ultrahigh-Rate and High-Safety Li^+ Storage Based on Interconnected Tunnel Structure in Micro-Size Niobium Tungsten Oxides. *Adv. Mater.* 2020, 32, 1905295.
4. Mertin, Anderson, Gruehn. Über die Kristallstruktur von $\text{M-Nb}_2\text{O}_5$. *J. Solid State Chem.* 1970, 1, 419-424.
5. Heurung, Gruehn. Neues zur Struktur und Stabilität von $\text{M-Nb}_2\text{O}_5$. *Z. anorg. allg. Chem.* 1982, 491, 101-112.

Reviewer #3 (Remarks to the Author):

The manuscript introduces a novel disordered phase of $\text{Nb}_{12}\text{WO}_{33}$ (*dt*- $\text{Nb}_{12}\text{WO}_{33}$) designed to enhance fast-charging capabilities in lithium-ion batteries. This study effectively highlights the role of structural disorder within crystallographic shear block structures, demonstrating improved Li-ion diffusion kinetics and rate performance. The use of advanced characterization techniques, including operando XRD and XAS, provides a solid foundation for correlating structural features with electrochemical performance. However, there are key concerns and areas for improvement that must be addressed to enhance the scientific rigor and practical relevance of the study.

(1) Discrepancy Between TEM Observations and DFT Model.

The irregular tetrahedral features observed in TEM images underscore the structural disorder inherent in *dt*- $\text{Nb}_{12}\text{WO}_{33}$. However, the computational model employed for DFT calculations appears to assume an idealized structure that does not align with the experimental data. This disconnect raises questions about the accuracy of the computational predictions.

Recommendation: Revise the computational model to reflect the experimentally observed disorder, incorporating variability in tetrahedral sites and distortions. Provide a comparative analysis between TEM-derived parameters and DFT inputs to validate the computational approach.

Reply: We did not perform DFT calculations for *dt*- $\text{Nb}_{12}\text{WO}_{33}$. In the discussion, we reference DFT calculations carried out by other authors for the known monoclinic structure *m*- $\text{Nb}_{12}\text{WO}_{33}$, although we do not use the results reported by those authors to make predictions regarding *dt*- $\text{Nb}_{12}\text{WO}_{33}$. Creating a computational model for *dt*- $\text{Nb}_{12}\text{WO}_{33}$ is more complicated than for *m*- $\text{Nb}_{12}\text{WO}_{33}$ due to the structural complexity of the former introduced by disorder, including variability in block sizes, tetrahedral site positions, and distortions of the tetrahedral sites. Reflecting the experimentally observed disorder would require a very large system and time-consuming calculations, which, in our opinion, would unjustifiably and considerably delay the reporting of our experimental results. We hope the reviewer understands.

(2) Economic Viability of Nb-W-O Compared to Nb-Oxides.

The inclusion of tungsten in $\text{Nb}_{12}\text{WO}_{33}$ introduces higher material costs compared to simpler Nb-based oxides. A detailed discussion of the economic implications and trade-offs between performance enhancements and material cost is essential for evaluating scalability.

Reply: We have added a discussion on the economic viability of the practical application of $dt\text{-Nb}_{12}\text{WO}_{33}$ as an electrode material in LIBs, as presented below.

“To assess the economic viability of $dt\text{-Nb}_{12}\text{WO}_{33}$, particularly the potential cost increase of introducing tungsten into a niobium oxide, we have considered the price of the metals and their estimated availability on the planet. The prices of niobium and tungsten are affected by supply constraints, geopolitical factors, and processing technology. According to a sustainability evaluation carried out by A. H. Tkaczyk *et al.*, the price of tungsten fluctuated after 2005, with peaks in 2006 (37 USD/kg) and 2012 (57 USD/kg), and then decreased to 35 USD/kg in 2017⁴⁹. As for niobium, the price remained stable at around 40-50 USD/kg since 2010, and was 50 USD/kg in 2017. Currently, the prices of niobium and tungsten are approximately 80 USD/kg and 43 USD/kg, respectively. A similar trend is found for the prices of niobium and tungsten precursors with identical characteristics (e.g. ligand and purity) sold by the main chemical reagent suppliers in Germany. Theoretically, the inclusion of W in Nb oxide compounds does not lead to higher material costs, while simultaneously enhancing the electrochemical performance. On the other hand, the tungsten reserves on the planet are smaller than those of niobium, with the tungsten content in the earth crust estimated as 1.25 mg kg⁻¹ compared to 20 mg kg⁻¹ for niobium. However, the atomic percentage of tungsten in $dt\text{-Nb}_{12}\text{WO}_{33}$ is only 8 %, while the performance gains from the introduction of tungsten are significant. For example, $dt\text{-Nb}_{12}\text{WO}_{33}$ delivers a high capacity of 119.5 mAh g⁻¹ at 15.3 A g⁻¹, compared to 94 mAh g⁻¹ at 4 A g⁻¹ for H-Nb₂O₅ and 60 mAh g⁻¹ at 0.132 A g⁻¹ for Nb₁₂O₂₉^{10, 17}. Therefore, considering the current prices of tungsten and niobium, the incorporation of tungsten seems economically advantageous, especially taking into account the potential use of the materials in high-end battery applications such as electric vehicles, where high power density makes it more efficient and user-friendly by reducing charging time.”

Reference:

- 49 Tkaczyk, A. H., Bartl, A., Amato, A., Lapkovskis, V. & Petranikova, M. Sustainability evaluation of essential critical raw materials: cobalt, niobium, tungsten and rare earth elements, *J. Phys. D: Appl. Phys.* **51**, 203001 (2018).

(3) White-Line Edge Analysis in XANES

The XANES spectra, particularly the white-line edge, provide critical information about electronic structure changes resulting from disorder. However, the current discussion is limited and does not fully explain how disorder enhances performance.

Reply: We have revised the manuscript to include more detailed discussions on the white-line edge of the W L_{III}-edge XANES spectra and on how the disorder in the *dt*-Nb₁₂WO₃₃ material enhances performance. The modifications made to the manuscript are shown below.

“The white line in the W L_{III}-edge on the XANES spectra (**Figure 1d**) corresponds to electron transitions from the 2p_{3/2} core level to quasi-bound 5d(W)+2p(O) mixed states²⁸. The position of the edge is influenced by the oxidation state of the metal and suggests that tungsten is in the oxidation state 6+ in both *m*-Nb₁₂WO₃₃ and *dt*-Nb₁₂WO₃₃, as in WO₃. In addition to the oxidation state, the coordination environment, including local disorder, also affects the intensity and shape of the white line, due to its effect on the density of states and distribution of unoccupied d states.”

“In addition, the intensity of the edge is lower for *dt*-Nb₁₂WO₃₃ compared to *m*-Nb₁₂WO₃₃, which is indicative of distortion in the former. This is because less distortion in the coordination environment around the tungsten tends to increase the overlap between W d-orbitals and O p-orbitals, which can rise the intensity of the white line due to increased electron density at the metal. Thus, the results suggest that the WO₄ and WO₆ polyhedra in *dt*-Nb₁₂WO₃₃ are more distorted.”

“A decrease of the pre-edge intensity in the Nb K-edge XANES spectra of both materials occurs during lithiation (**Figure 5c, 5i, S34**). This reflects an increase of the symmetry in the octahedral environment around the Nb atoms, i.e., a gradual decrease of the octahedra distortion, in agreement with experimental results and DFT calculations reported for other niobium tungsten oxide block structures. It is attributed partially to the increase of the energy of the d states during lithiation, as the metal is reduced and its electron configuration deviate from d⁰, mitigating the second-order Jahn-Teller effect. The second derivative of the W L_{III}-edge XANES spectra remains similar to those of the corresponding pristine materials, showing only one peak, suggesting that tungsten remains mainly in tetrahedral coordination in *m*-Nb₁₂WO₃₃ and *dt*-Nb₁₂WO₃₃ during discharge (**Figures S36, S37**). However, there are clear differences in the evolution of the spectra of the materials (**Figure 5e-h**). For *m*-Nb₁₂WO₃₃, the intensity of the edge diminishes during lithiation

and the intensity of the post-edge feature also decreases. This can be attributed to the WO_4 and WO_6 polyhedra getting progressively more distorted as the discharge process advances. As a result of the distortion around the W atoms, a decrease in the overlapping between W d-orbitals and O p-orbitals can occur, decreasing the electron density at the metal center. The changes in the spectra are only partially reversible during the charge process. On the contrary, the intensity of the white line in the spectra of $dt\text{-Nb}_{12}\text{WO}_{33}$ does not change significantly compared to $m\text{-Nb}_{12}\text{WO}_{33}$, suggesting less important changes. Less electron density on the tungsten in $dt\text{-Nb}_{12}\text{WO}_{33}$ due to distortion could contribute to the more accentuated decrease of the oxidation state of tungsten in $dt\text{-Nb}_{12}\text{WO}_{33}$ compared to $m\text{-Nb}_{12}\text{WO}_{33}$ at the initial stages of discharge (**Figure 5k**).

“The types of cavity sites for Li-ion insertion present in Wadsley-Roth phases of Nb-based oxides were first categorized by Cava et al.³⁸ The structure of $dt\text{-Nb}_{12}\text{WO}_{33}$ possesses cavity sites of the types I, II, III, V, and VI, that provide potential pathways for Li-ion diffusion. The position of these cavities in the structure is schematized in **Figure S21**. Koçer et al. have performed DFT calculations to determine the energetically favorable pathways for Li^+ -ion motion within niobium tungsten oxide block structures¹¹. Li^+ diffusion was found to be energetically unfavorable across shear planes and through type VI cavities, which are formed by tetrahedrally coordinated tungsten at the junctions of the blocks and octahedral sites in the vertices of the blocks (**Figure 1b and Figure S21**). In contrast, Li-ion diffusion through the sites within the blocks was found to involve lower energy barriers, making these the most likely pathways in the block structures. As suggested by the PDF data, the disorder in $dt\text{-Nb}_{12}\text{WO}_{33}$ shortens the distances between W in tetrahedral sites and Nb in octahedral sites at the corner of the blocks, making it very unlikely for Li-ions to diffuse via type VI cavities. However, disorder also leads to larger distances between the metal sites within the blocks, indicating wider pathways for lithium diffusion inside the blocks of $dt\text{-Nb}_{12}\text{WO}_{33}$, which contributes to the larger Li^+ diffusion coefficients and better rate capability of $dt\text{-Nb}_{12}\text{WO}_{33}$ compared to $m\text{-Nb}_{12}\text{WO}_{33}$.”

Changes made to the references section:

28. Charton, P., Gengembre, L., & Armand, P. $\text{TeO}_2\text{-WO}_3$ glasses: infrared, XPS and XANES structural characterizations. *J. Solid State Chem.* 168, 175-183 (2002).

(4) Assumptions in GITT-Based Diffusion Coefficient Calculations (Figure 3f)

The reported diffusion coefficients derived from GITT analysis rely on certain assumptions regarding material properties and boundary conditions based on Fick's laws of diffusion. A detailed explanation of these assumptions is crucial for ensuring the reliability of the results.

Reply: According to the reviewer suggestion, we have added a detailed explanation to the SI on the determination of the diffusion coefficients derived from GITT, including assumptions and boundary conditions, as described below.

“Based on Fick's first law³, the electric current (I) can be expressed as:

$$I = \left(-D \frac{\partial c_i}{\partial x} \Big|_{x=0} \right) S z_i q \quad (1)$$

where I is the constant current pulse (I_0), D is the diffusion coefficient of species i (Li^+), x represents the distance coordinate, S is the area of the electrode, z_i is the charge number ($z_i = 1$ for Li), and q is the elementary charge ($q = 1.602 \times 10^{-19}$ C). Equation (1) shows that the current is equal to the number of charge carriers transported at the phase boundary of the electrode and the electrolyte ($x = 0$).

The concentration gradient, $\partial c_{\text{Li}} / \partial x$, is a function of x and time (t). The diffusion coefficient is calculated with Fick's second law:

$$\frac{\partial c_i(x, t)}{\partial t} = D \frac{\partial^2 c_i(x, t)}{\partial x^2} \quad (2)$$

with the initial and boundary conditions:

$$c_i(x, t = 0) = c_0 \quad (0 \leq x \leq L) \quad (3)$$

$$-D \frac{\partial c_i}{\partial x} \Big|_{x=0} = \frac{I_0}{S z_i q} \quad (4)$$

$$\frac{\partial c_i}{\partial x} \Big|_{x=L} = 0 \quad (t \geq 0) \quad (5)$$

Equation (3) is the initial condition at equilibrium. Equations (4) and (5) are boundary conditions at $x = 0$ and $x = L$, respectively. The solution of the differential Equation (2) under conditions (3-5) can be determined as follows for $x = 0$:

$$c_i(x = 0, t) = c_0 + \frac{2I_0 \sqrt{t}}{S z_i q \sqrt{D}} \sum_{n=0}^{\infty} \left(\text{ierfc} \left[\frac{nL}{\sqrt{Dt}} \right] + \text{ierfc} \left[\frac{(n+1)L}{\sqrt{Dt}} \right] \right) \quad (6)$$

where $ierfc(z) = [\pi^{-0.5} \exp(-z^2)] - z[1 - erf(z)]$, and $erf(z)$ is the error function. Considering that $t \ll L^2/D$, Equation (6) can be expressed as:

$$\frac{dc_i(x=0, t)}{d\sqrt{t}} = \frac{2I_0\sqrt{t}}{Sz_iq\sqrt{D}} \left(t \ll \frac{L^2}{D} \right) \quad (7)$$

If the volume change with composition is neglected, and the relationship between the concentration and compositions is:

$$V_M dc_i = N_A d\delta \quad (8)$$

where V_M is the molar volume of the electrode material ($\text{cm}^3 \text{mol}^{-1}$) and N_A is Avogadro's number ($6.02 \times 10^{23} \text{mol}^{-1}$), insertion of equation (8) into equation (7) and expanding by dE , results in the following equation (9):

$$\frac{dE}{d\sqrt{t}} = \frac{2V_M I_0}{SFz_i\sqrt{D}\pi} \frac{dE}{d\delta} \left(t \ll \frac{L^2}{D} \right) \quad (9)$$

The diffusion coefficient can then be obtained from Equation (9) as follows:

$$D = \frac{4}{\pi} \left(\frac{V_M I_0}{SFz_i} \right)^2 \left[\frac{dE}{d\delta} / \frac{dE}{d\sqrt{t}} \right]^2 \left(t \ll \frac{L^2}{D} \right) \quad (10)$$

where F is Faraday's constant (96485 C mol^{-1}). $dE/d\delta$ and $dE/d\sqrt{t}$ represent the change in the steady-state voltage after subtracting the IR drop and transient change in voltage, respectively. When using a sufficiently small current, $dE/d\delta$ could be replaced by the ratio of the finite quantities, $\Delta E_s/\Delta\delta$. Moreover, if E vs. \sqrt{t} exhibits the linear behavior, it can be expressed as $\Delta E_\tau/\sqrt{\tau}$. The stoichiometry changes during the equilibration process in the titration step of Li^+ can be determined as follows:

$$\Delta\delta = \frac{\tau M_B I_0}{Fz_i m_B} \quad (11)$$

where m_B and M_B are the mass of active material and its molecular weight, respectively. Then, equation (10) can be transformed into the following equation (12):

$$D = \frac{4}{\pi\tau} \left(\frac{m_B V_M}{M_B S} \right)^2 \left(\frac{\Delta E_s}{\Delta E_\tau} \right)^2 \left(\tau \ll \frac{L^2}{D} \right) \quad (12)$$

Reference in SI:

3. Weppner, W., & Huggins, R. A. Determination of the kinetic parameters of mixed-conducting electrodes and application to the system Li_3Sb . *J. Electrochem. Soc.*, **124**, 1569 (1977).

Comment 5: Structural Stability and Long-Term Performance

Although the manuscript demonstrates excellent capacity retention over 1000 cycles, it lacks a discussion of potential degradation mechanisms. Long-term cycling may induce fractures or irreversible distortions within the disordered structure.

Reply: We have added a discussion on the degradation mechanisms in $m\text{-Nb}_{12}\text{WO}_{33}$ and $dt\text{-Nb}_{12}\text{WO}_{33}$ after long-term cycling. The discussion included in the revised manuscript and the new figures added to the SI are as follows:

“To evaluate the degradation mechanisms during long-term cycling, SEM, XRD, and XAS analyses of the materials were performed after 500 cycles. SEM images of $m\text{-Nb}_{12}\text{WO}_{33}$ and $dt\text{-Nb}_{12}\text{WO}_{33}$ (Figure S42) show the absence of cracks or holes on the surface of the materials. Comparison between the X-ray diffractograms of the materials before and after cycling does not indicate the irreversible formation of other crystalline phases (Figures S43a, S43b) and suggests that the long-range structure is maintained after cycling. The Nb K-edge XANES spectra of $m\text{-Nb}_{12}\text{WO}_{33}$ and $dt\text{-Nb}_{12}\text{WO}_{33}$ before and after cycling almost overlap (Figure S43c). A similar result is obtained with the W L_{III} -edge XANES spectra (Figure S43d), which suggests reversibility of the redox reactions for both materials. Figure S43e compares the Nb K-edge EXAFS spectra of the initial and cycled $m\text{-Nb}_{12}\text{WO}_{33}$ and $dt\text{-Nb}_{12}\text{WO}_{33}$. In the case of $m\text{-Nb}_{12}\text{WO}_{33}$, there is a shift of the Nb–O peak after cycling from 1.50 to 1.54 Å, accompanied by a decrease of the peak intensity, whereas only a slight decrease in the intensity of the Nb–O peak is observed for $dt\text{-Nb}_{12}\text{WO}_{33}$, without significant peak shift. As for the W L_{III} -edge EXAFS spectra (Figure S43f), a more accentuated decrease in the W–O peak intensity is observed for $m\text{-Nb}_{12}\text{WO}_{33}$ after cycling than for $dt\text{-Nb}_{12}\text{WO}_{33}$. The results reveal that irreversible local distortions occur within the NbO_6 and WO_4 polyhedra in both materials after repeated lithiation and delithiation processes, although the effect is smaller for $dt\text{-Nb}_{12}\text{WO}_{33}$. Therefore, irreversible local distortions appear to be the

primary degradation mechanism during long-term cycling in $m\text{-Nb}_{12}\text{WO}_{33}$ and $dt\text{-Nb}_{12}\text{WO}_{33}$, leading to a decrease in capacity.”

Figure S42. SEM images of the (a, b) $m\text{-Nb}_{12}\text{WO}_{33}$ and (c, d) $dt\text{-Nb}_{12}\text{WO}_{33}$ electrode after 500 cycles at 10 C.

Figure S43. XRD patterns of (a) $m\text{-Nb}_{12}\text{WO}_{33}$ and (b) $dt\text{-Nb}_{12}\text{WO}_{33}$ before and after 500 cycles at 10 C. (c) Nb K-edge and (d) W L_{III}-edge XANES spectra of $m\text{-Nb}_{12}\text{WO}_{33}$ and $dt\text{-Nb}_{12}\text{WO}_{33}$ before and after 500 cycles at 10 C. (e) Nb K-edge and (f) W L_{III}-edge EXAFS spectra of $m\text{-Nb}_{12}\text{WO}_{33}$ and $dt\text{-Nb}_{12}\text{WO}_{33}$ before and after 500 cycles at 10 C.

Comment 6: Comparison with Other Nb-Based Anode Materials

The manuscript compares $dt\text{-Nb}_{12}\text{WO}_{33}$ with its ordered counterpart ($m\text{-Nb}_{12}\text{WO}_{33}$). However, a broader comparison with other Nb-based anode materials from the literature would contextualize the improvements reported in this study.

Reply: Following the suggestion of the reviewer, we have included in the revised manuscript a comparison with a broader range of other Nb-based anode materials from the literature, including crystallographic shear structures and tungsten bronze structures.

We have revised Figure 3g and Table S7 of the supplementary information accordingly, and added the additional references, as shown below.

“*dt-Nb₁₂WO₃₃* exhibits superior electrochemical performance compared with the previously reported Nb-based anode materials (Figure 3g and Table S7)^{5,6,13,15,42,43,44,45}”

Figure 3. Li-ion storage properties and diffusion kinetics in *m-Nb₁₂WO₃₃* and *dt-Nb₁₂WO₃₃*.

(a) Charge and discharge profiles of *m-Nb₁₂WO₃₃* and *dt-Nb₁₂WO₃₃* at 0.5 C. (b) CV curves of *m-Nb₁₂WO₃₃* and *dt-Nb₁₂WO₃₃* at 0.1 mV s⁻¹. (c) Rate capability of *m-Nb₁₂WO₃₃* and *dt-Nb₁₂WO₃₃*. (d) Linear relationship between the peak current (*i_p*) and the square root of the scan rate (*v*^{1/2}) for peak O1. (e) GITT profiles of *m-Nb₁₂WO₃₃* and *dt-Nb₁₂WO₃₃* as a function of *x* in *Li_xNb₁₂WO₃₃*. (f) Variation of the *D_{Li+}* determined by GITT. (g) Comparison of the rate capability between *dt-Nb₁₂WO₃₃* and other reported niobium-based oxides with block structures. (h) Cycling performance of *dt-Nb₁₂WO₃₃* at 10 C.

Table S7. Comparison of the electrochemical performance of *dt*-Nb₁₂WO₃₃ with that of other previously reported Nb-based materials.

Material	Structure type	Voltage range (V vs. Li ⁺ /Li)	Initial reversible capacity (mAh g ⁻¹)	Rate performance (mAh g ⁻¹)	Current density of rate performance (A g ⁻¹)
Nb ₁₆ W ₅ O ₅₅ ⁹	Shear structure	1.0-3.0	225 (at 34.3 mA g ⁻¹)	~50	10.3
Nb ₁₈ W ₁₆ O ₉₃ ⁹	Tungsten bronze structure	1.0-3.0	205 (at 29.8 mA g ⁻¹)	70	14.9
Nb ₁₄ W ₃ O ₄₄ ¹⁰	Shear structure	1.0-3.0	221.3 (at 89 mA g ⁻¹)	84.4	8.9
Nano-block Nb ₁₄ W ₃ O ₄₄ ¹¹	Shear structure	1.0-3.0	241.1 (at 89 mA g ⁻¹)	109.5	14.2
Nb ₁₂ WO ₃₃ nanowires ¹²	Shear structure	1.0-3.0	228 (at 200 mA g ⁻¹)	145.8	0.7
Nb ₁₈ W ₈ O ₆₉ ¹³	Shear structure	1.0-3.0	~230 (at 32.8 mA g ⁻¹)	~30	9.8
H-Nb ₂ O ₅ ¹⁴	Shear structure	1.0-3.0	269 (at 20 mA g ⁻¹)	94	4
PNb ₉ O ₂₅ ¹⁵	Shear structure	1.0-3.0	~235 (at 25.4 mA g ⁻¹)	30	15.2
VNb ₉ O ₂₅ ¹⁵	Shear structure	1.0-3.0	~180 (at 29.2 mA g ⁻¹)	25	11.7
TiNb ₂₄ O ₆₂ ¹⁶	Shear structure	1.0-3.0	214 (at 20.5 mA g ⁻¹)	~100	3.1
V ₇ Nb ₆ O ₂₉ ¹⁷	Shear structure	1.2-3.3	208 (at 20 mA g ⁻¹)	106	2
Ba _{3.4} Nb ₁₀ O _{28.4} ¹⁸	Tungsten bronze structure	0.8-3.0	167 (at 28.97 mA g ⁻¹)	82	2.9
Nb ₁₂ W ₁₁ O ₆₃ ¹⁹	Tungsten bronze structure	1.3-3	176 (at 22.6 mA g ⁻¹)	100	4.5
CoNb ₁₁ O ₂₉ ²⁰	Shear structure	0.8-3	265 (at 40 mA g ⁻¹)	113	4
Nb ₁₂ O ₂₉ ²¹	Shear structure	1.0-2.5	243 (at 6.6 mA g ⁻¹)	60	0.132
dt -Nb ₁₂ WO ₃₃ (This work)	Shear structure	1-3	257 (at 95.4 mA g ⁻¹)	119.5	15.3

References:

43. Yang, M., Li, S., Huang J. Three-dimensional cross-linked Nb₂O₅ polymorphs derived from cellulose substances: Insights into the mechanisms of Lithium storage. *ACS Appl. Mater. Interfaces* **13**, 39501–39512 (2021).
44. Preefer, M. B. et al. Multielectron redox and insulator-to-metal transition upon lithium insertion in the fast-charging, Wadsley-Roth phase PNb₉O₂₅. *Chem. Mater.* **32**, 4553–4563 (2020).
45. Griffith, K. J., Senyshyn A., Grey, C. P. Structural stability from crystallographic shear in TiO₂–Nb₂O₅ phases: cation ordering and lithiation behavior of TiNb₂₄O₆₂. *Inorg. Chem.* **56**, 4002–4010 (2017).

References in SI:

9. Griffith, K. J., Wiaderek, K. M., Cibin, G., Marbella, L. E. & Grey, C. P. Niobium tungsten oxides for high-rate lithium-ion energy storage. *Nature* **559**, 556-563 (2018).
10. Yang, Y. et al. Achieving Achieving ultrahigh-rate and high-safety Li⁺ storage based on interconnected tunnel structure in micro-size niobium tungsten oxides. *Adv. Mater.* **32**, 1905295 (2020).
11. Guo, C. et al. Nano-sized niobium tungsten oxide anode for advanced fast-charge lithium-ion batteries. *Small* **18**, 2107365 (2022).
12. Yan, L. et al. Electrospun WNb₁₂O₃₃ nanowires: superior lithium storage capability and their working mechanism. *J. Mater. Chem. A* **5**, 8972-8980 (2017).
13. Griffith, K. J., & Grey, C. P. Superionic lithium intercalation through 2 × 2 nm² columns in the crystallographic shear phase Nb₁₈W₈O₆₉. *Chem. Mater.* **32**, 3860-3868 (2020).
14. Yang, M., Li, S., Huang J. Three-dimensional cross-linked Nb₂O₅ polymorphs derived from cellulose substances: Insights into the mechanisms of Lithium storage. *ACS Appl. Mater. Interfaces* **13**, 39501–39512 (2021).
15. Preefer, M. B. et al. Multielectron redox and insulator-to-metal transition upon lithium insertion in the fast-charging, Wadsley-Roth phase PNb₉O₂₅. *Chem. Mater.* **32**, 4553–4563 (2020).
16. Griffith, K. J., Senyshyn A., Grey, C. P. Structural stability from crystallographic shear in TiO₂–Nb₂O₅ phases: cation ordering and lithiation behavior of TiNb₂₄O₆₂. *Inorg. Chem.* **56**, 4002–4010 (2017).

17. Lawrence, E. A. et al. Reversible electrochemical lithium cycling in vanadium(IV)- and niobium(V)-based Wadsley–Roth phase. *Chem. Mater.* **35**, 3470–3483 (2023).
18. Xiong, X. et al. Cation-vacancy ordered superstructure enhanced cycling stability in tungsten bronze anode. *Adv. Energy Mater.* **12**, 2201967 (2022).
19. Ma, X.-H., et al. Influence of cut-off voltage on the lithium storage performance of $\text{Nb}_{12}\text{W}_{11}\text{O}_{63}$ anode, *Electrochim. Acta* **332**, 135380 (2020).
20. Liu, H., Chen, C. Wadsley–Roth phase $\text{CoNb}_{11}\text{O}_{29}$ as a high-performance anode for lithium-ion batteries. *J. Mater. Chem. A*, **12**, 5414-5421 (2024).
21. Li, Y., Sun, C., Goodenough, J. B. Electrochemical Lithium Intercalation in Monoclinic $\text{Nb}_{12}\text{O}_{29}$. *Chem. Mater.* **23**, 2292–2294 (2011).²¹

Reviewer #1 (Remarks to the Author):

The manuscript has been greatly improved with the revisions. It is now nearly ready for publication. One issue, not on the technical part, is that the data in the technoeconomics section is dated, in part, and the current prices are inaccurate. Specifically, the sentence "Currently, the prices of niobium and tungsten are approximately 80 USD/kg and 43 USD/kg, respectively." contains values that are inaccurate, and this line does not even have a citation. It is challenging to get good mineral prices, especially for certain elements, but one should compare on an equivalent basis (such as EXW China, Europe, etc) or, if unknown, it would be better not to include faulty data that will mislead others. A more challenging issue that must be addressed is that the comparison in Figure 3g is misleading since a high carbon content (20%) was used in the electrodes in this work. The loading was on the low side of standard, so that could be acceptable, but the carbon content also has a big role and high current data in particular cannot be compared directly with this discrepancy.

Reviewer #2 (Remarks to the Author):

The revision of the manuscript by Liu et al. has been done thoroughly and comprehensively. All reviewer comments have been considered in an adequate manner, and the overall quality of this excellent article has been further improved thereby. In particular, the structural description of the two modifications of Nb₁₂WO₃₃ is now complete: the single crystal structure determination of m-Nb₁₂WO₃₃ and the careful examination of HAADF-STEM images of dt-Nb₁₂WO₃₃ sheds new light onto important structural questions such as the localization of W in the block structures. Moreover, these findings allow a more reliable interpretation of the spectroscopic results obtained for the initial and the lithiated samples. The additional investigation of the influence of crystal size on electrochemical properties is certainly of widespread interest. I recommend the publication of the article after the following minor points have been addressed:

1. Title and text: "partially ordered" is nowhere explained (suggestion: introduce it on page 3 where it is first mentioned in the text).
2. page 1: "a disordered arrangement of the tungsten tetrahedra at the corners of the blocks,"
page 3: "an irregular distribution of the tetrahedral sites within the structure". Can this distribution really be called "disordered" or "irregular"? The tetrahedral sites still occur at their "regular" positions, namely at the block corners. Maybe it is better to combine the two parts of the description, e. g.: "The partially disordered Nb₁₂WO₃₃ phase consists of corner-shared NbO₆ octahedra blocks of varied sizes, including 5x4, 4x4, and 4x3, causing a little ordered arrangement of the tungsten tetrahedra at the corners of the blocks, as well as distortion of the WO₄ tetrahedra."
Also in Fig. 1b, the designation "disordered" and "ordered tetrahedral position" is somewhat misleading. Better, e. g.: "disordered block structure" and "3x4 blocks with tetrahedral sites"
3. page 4: "we have synthesized larger crystals" Larger than what? Better give crystal size (1-3 microns) instead.
4. page 4: "According to the refinement data in Table S1, the tungsten atoms show a preference for the tetrahedral sites at the corners of the blocks, with a 100 % occupancy."
Misleading. In Table S1, the occupancy is given as 0.4438. This means that the site is only partially filled (not 100%) but that there is exclusively W.
From the crystallographic point of view, it is interesting to note that the y value is given as 0.2500(2). Is this measuring error really significant? It turns a high symmetry 2b position (0, 0.25, 0) into a 4g (0,y,0).
The x,y,z values in the header of Table S1 are incorrectly given in Angstrom but these are fractional coordinates without dimensions.
5. page 7 center: "suggest that tungsten is in the oxidation state 6+ in both m-Nb₁₂WO₃₃ and dt-Nb₁₂WO₃₃." Sentence used twice here.
6. page 7: "The magnitude of the ligand field split is higher -> highest for cations in octahedral field,..."
7. page 8: "whereas a weaken -> weak intensity band is observed for dt-Nb₁₂WO₃₃,"
8. page 15: The abbreviations "BM" and "MS" should be introduced properly.
9. Give the direction of observation for Figures S9, S31 and S32.

10. Table 2 ff: curvefit -> curve fit or curve fitting

Reviewer #3 (Remarks to the Author):

During the revision process, the authors significantly improved the overall quality of the manuscript. However, I still have concerns regarding the commercial viability of Nb-oxide anode materials for electric vehicle (EV) applications. The authors state the following: "As for niobium, the price remained stable at around 40–50 USD/kg since 2010, and was 50 USD/kg in 2017. Currently, the prices of niobium and tungsten are approximately 80 USD/kg and 43 USD/kg, respectively."

In comparison, the current price of graphite is around 5–10 USD/kg, while Si/C composites are priced at approximately 60–100 USD/kg. However, Si-based anode materials offer much higher energy densities (200–300 Wh/kg), which provide a compelling advantage for EV markets. In contrast, Nb-oxide still suffers from relatively low energy densities and requires a larger volume of material in battery packs, which may hinder its competitiveness in practical EV applications. It seems usage of Nb-oxides have no beneficial as an anode material.

Replies to the comments of the reviewers

We thank the reviewers for their comments. We have addressed every comment and have answer all questions raised by the reviewers. We hope that the manuscript is now ready for publication. All the modifications made to the manuscript are highlighted in blue.

Reviewer #1

The manuscript has been greatly improved with the revisions. It is now nearly ready for publication.

(1) One issue, not on the technical part, is that the data in the technoeconomics section is dated, in part, and the current prices are inaccurate. Specifically, the sentence "Currently, the prices of niobium and tungsten are approximately 80 USD/kg and 43 USD/kg, respectively." contains values that are inaccurate, and this line does not even have a citation. It is challenging to get good mineral prices, especially for certain elements, but one should compare on an equivalent basis (such as EXW China, Europe, etc) or, if unknown, it would be better not to include faulty data that will mislead others.

Reply: We obtained the current prices of niobium and tungsten from *Shanghai Metals Market (SMM)* (1,2). However, given the challenges of obtaining the prices of these metals and how fast they can become outdated, we have removed the sentence "Currently, the prices of niobium and tungsten are approximately 80 USD/kg and 43 USD/kg, respectively.". This change does not affect the discussion in the manuscript.

(1). <https://www.metal.com/Niobium-Tantalum/201102250606>.

(2). <https://www.metal.com/Tungsten/201102250208>.

(2) A more challenging issue that must be address is that the comparison in Figure 3g is misleading since a high carbon content (20%) was used in the electrodes in this work. The loading was on the low side of standard, so that could be acceptable, but the carbon content also has a big role and high current data in particular cannot be compared directly with this discrepancy.

Reply: We agree with the reviewer that direct comparisons with results from the literature are challenging due to experimental differences; however, they also provide a rough estimate of the performance of a material relative to others. For this reason, reviewers often request such comparisons or to extend them, as was the case with this manuscript. To address the reviewer’s concerns, we have added the mass loading and carbon content of the electrodes to Table S7 (see below), so that all relevant information about the electrodes is provided. The Nb-based materials shown in Figure 3g are also included in Table S7. We have also modified the discussion, as shown below.

“*dt*-Nb₁₂WO₃₃ exhibits superior electrochemical performance compared with the previously reported Nb-based anode materials (Figure 3g and Table S7)^{5,6,13,15,42,43,44,45}, although it should be noted that direct comparisons are challenging due to experimental differences in electrode preparation such as carbon content and mass loading.”

Table S7. Comparison of the electrochemical performance of *dt*-Nb₁₂WO₃₃ with that of other previously reported Nb-based materials.

Material	Structure type	Voltage range (V vs. Li ⁺ /Li)	Initial reversible capacity (mAh g ⁻¹)	Rate performance (mAh g ⁻¹)	Current density of rate performance (A g ⁻¹)	Carbon content (%)	Mass loading (mg cm ⁻²)
Nb ₁₆ W ₅ O ₅₅ ⁹	Shear structure	1.0-3.0	225 (at 34.3 mA g ⁻¹)	~50	10.3	10	2-3
Nb ₁₈ W ₁₆ O ₉₃ ⁹	Tungsten bronze structure	1.0-3.0	205 (at 29.8 mA g ⁻¹)	70	14.9	10	2-3
Nb ₁₄ W ₃ O ₄₄ ¹⁰	Shear structure	1.0-3.0	221.3 (at 89 mA g ⁻¹)	84.4	8.9	20	1.4-1.7
Nano-block Nb ₁₄ W ₃ O ₄₄ ¹¹	Shear structure	1.0-3.0	241.1 (at 89 mA g ⁻¹)	109.5	14.2	10	2
Nb ₁₂ WO ₃₃ nanowires ¹²	Shear structure	1.0-3.0	228 (at 200 mA g ⁻¹)	145.8	0.7	10	1.5

$\text{Nb}_{18}\text{W}_8\text{O}_{69}$ ¹³	Shear structure	1.0-3.0	~230 (at 32.8 mA g ⁻¹)	~30	9.8	10	2.0 ± 0.2
$\text{H-Nb}_2\text{O}_5$ ¹⁴	Shear structure	1.0-3.0	269 (at 20 mA g ⁻¹)	94	4	20	1.0-1.4
$\text{PNb}_9\text{O}_{25}$ ¹⁵	Shear structure	1.0-3.0	~235 (at 25.4 mA g ⁻¹)	30	15.2	15	1.5
$\text{VNb}_9\text{O}_{25}$ ¹⁵	Shear structure	1.0-3.0	~180 (at 29.2 mA g ⁻¹)	25	11.7	15	1.5
$\text{TiNb}_{24}\text{O}_{62}$ ¹⁶	Shear structure	1.0-3.0	214 (at 20.5 mA g ⁻¹)	~100	3.1	10	2-3
$\text{V}_7\text{Nb}_6\text{O}_{29}$ ¹⁷	Shear structure	1.2-3.3	208 (at 20 mA g ⁻¹)	106	2	10	1-1.5
$\text{Ba}_{3.4}\text{Nb}_{10}\text{O}_{28.4}$ ¹⁸	Tungsten bronze structure	0.8-3.0	167 (at 28.97 mA g ⁻¹)	82	2.9	20	1
$\text{Nb}_{12}\text{W}_{11}\text{O}_{63}$ ¹⁹	Tungsten bronze structure	1.3-3	176 (at 22.6 mA g ⁻¹)	100	4.5	10	2
$\text{CoNb}_{11}\text{O}_{29}$ ²⁰	Shear structure	0.8-3	265 (at 40 mA g ⁻¹)	113	4	20	1.3
$\text{Nb}_{12}\text{O}_{29}$ ²¹	Shear structure	1.0-2.5	243 (at 6.6 mA g ⁻¹)	60	0.132	20	10-20
$d\text{t-Nb}_{12}\text{WO}_{33}$ (This work)	Shear structure	1-3	257 (at 95.4 mA g ⁻¹)	119.5	15.3	20	1.5-2

Reviewer #2

The revision of the manuscript by Liu et al. has been done thoroughly and comprehensively. All reviewer comments have been considered in an adequate manner, and the overall quality of this excellent article has been further improved thereby. In particular, the structural description of the two modifications of $\text{Nb}_{12}\text{WO}_{33}$ is now complete: the single crystal structure determination of *m*- $\text{Nb}_{12}\text{WO}_{33}$ and the careful examination of HAADF-STEM images of *dt*- $\text{Nb}_{12}\text{WO}_{33}$ sheds new light onto important structural questions such as the localization of W in the block structures. Moreover, these findings allow a more reliable interpretation of the spectroscopic results obtained for the initial and the lithiated samples. The additional investigation of the influence of crystal size on electrochemical properties is certainly of widespread interest. I recommend the publication of the article after the following minor points have been addressed:

(1) Title and text: “partially ordered” is nowhere explained (suggestion: introduce it on page 3 where it is first mentioned in the text).

Reply: An explanation for the term “partially disordered” is now included in page 3, as shown below.

“By controlling the synthesis conditions, a partially disordered $\text{Nb}_{12}\text{WO}_{33}$ phase (*dt*- $\text{Nb}_{12}\text{WO}_{33}$) can be obtained, whose structure is ordered along the *c* axis whereas the presence of little ordered shear blocks in a variety of sizes along *ab* plane, including 5x4, 4x4, and 4x3, causes a little ordered arrangement of the tungsten tetrahedra at the corners of the blocks, as well as distortion of the WO_4 tetrahedra.”

(2) page 1: “a disordered arrangement of the tungsten tetrahedra at the corners of the blocks,”

page 3: “an irregular distribution of the tetrahedral sites within the structure”. Can this distribution really be called “disordered” or “irregular”? The tetrahedral sites still occur at their “regular” positions, namely at the block corners. Maybe it is better to combine the two parts of the description, e. g.: “The partially disordered $\text{Nb}_{12}\text{WO}_{33}$ phase consists of corner-shared NbO_6 octahedra blocks of varied sizes, including 5x4, 4x4, and 4x3, causing a little ordered arrangement of the tungsten tetrahedra at the corners of the blocks, as well as distortion of the WO_4 tetrahedra.”

Also in Fig. 1b, the designation “disordered” and “ordered tetrahedral position” is somewhat misleading. Better, e. g.: “disordered block structure” and “3x4 blocks with tetrahedral sites”

Reply: We have revised the text and Figure 1b according to the comments of the reviewer, as shown below.

“By controlling the synthesis conditions, a partially disordered $\text{Nb}_{12}\text{WO}_{33}$ phase ($dt\text{-Nb}_{12}\text{WO}_{33}$) can be obtained, whose structure is ordered along the c axis whereas the presence of little ordered shear blocks in a variety of sizes along ab plane, including 5×4 , 4×4 , and 4×3 , causes a little ordered arrangement of the tungsten tetrahedra at the corners of the blocks, as well as distortion of the WO_4 tetrahedra.”

Figure 1. Structural characterization of $m\text{-Nb}_{12}\text{WO}_{33}$ and $dt\text{-Nb}_{12}\text{WO}_{33}$. (a) XRD patterns of $m\text{-Nb}_{12}\text{WO}_{33}$ and $dt\text{-Nb}_{12}\text{WO}_{33}$. (b) Schematic illustration of the structure for $m\text{-Nb}_{12}\text{WO}_{33}$ and $dt\text{-Nb}_{12}\text{WO}_{33}$.

Nb₁₂WO₃₃ (blue and green blocks are offset by 1/2 *b* for *m*-Nb₁₂WO₃₃ and 1/2 *c* for *dt*-Nb₁₂WO₃₃). Nb K-edge (c) and W L_{III}-edge (d) XANES spectra of *m*-Nb₁₂WO₃₃ and *dt*-Nb₁₂WO₃₃. (e) Fitting of the Nb K-edge first and second coordination shells in the FT-EXAFS spectra of *m*-Nb₁₂WO₃₃ and *dt*-Nb₁₂WO₃₃. (f) Fitting of the W L_{III}-edge first coordination shell in the FT-EXAFS spectra of *m*-Nb₁₂WO₃₃ and *dt*-Nb₁₂WO₃₃. (g) PDF pattern of *m*-Nb₁₂WO₃₃ and *dt*-Nb₁₂WO₃₃. (h) Schematic illustration of type I and VI cavities viewed along the *c* axis; the atomic pairs Nb-Nb (blue) and W-Nb (red) in the PDF pattern in g are represented by blue and red lines, respectively, in the schematic structure in h.

(3) page 4: “we have synthesized larger crystals” Larger than what? Better give crystal size (1-3 microns) instead.

Reply: We have revised the mentioned text, as shown below.

“To determine the structure of *m*-Nb₁₂WO₃₃, we synthesized larger crystals (6-19 μm in length) (Figure S2) than those obtained via solvothermal method (1-3 μm) and performed single-crystal XRD measurements.”

(4.1) page 4: “According to the refinement data in Table S1, the tungsten atoms show a preference for the tetrahedral sites at the corners of the blocks, with a 100 % occupancy.”

Misleading. In Table S1, the occupancy is given as 0.4438. This means that the site is only partially filled (not 100%) but that there is exclusively W.

Reply: This was a mistake. We have corrected the text, as shown below.

“According to the refinement data in Table S1, the tungsten atoms show a preference for the tetrahedral sites at the corners of the blocks.”

(4.2) From the crystallographic point of view, it is interesting to note that the y value is given as 0.2500(2). Is this measuring error really significant? It turns a high symmetry 2b position (0, 0.25, 0) into a 4g (0,y,0).

Reply: The crystallographic analysis of the tungsten (W) occupancy in this structure hinges on resolving a discrepancy between symmetry considerations and realistic bonding geometry. The measured y-coordinate of 0.2500(2) is very close to the high-symmetry 2b value but includes a small error margin, raising the possibility that the W atom actually occupies a general 4g position (0, y, 0) with slight positional displacement. This shift from 2b to 4g reduces symmetry but better aligns with observed bond lengths. Forcing W into the 2b position results in unrealistic W–O distances of 1.4927 Å and 2.4262 Å, which deviate significantly from typical W–O bond lengths in oxides (~1.7–2.2 Å). These anomalies indicate geometric strain, undermining the chemical plausibility of the model. In contrast, modeling W in the 4g position with half occupancy (≈ 0.5) produces more reasonable bond distances (W–O 1.7720 Å), consistent with known crystallographic trends. The refined occupancy of 0.4438 aligns with this interpretation when accounting for symmetry-related multiplicity and potential dynamic disorder. The slight deviation in the y-coordinate (0.2500(2)) could reflect minor positional disorder or measurement uncertainty, common in structures with partial occupancy. While the 2b position offers higher symmetry, the anomalous bonding geometry and occupancy data justify prioritizing chemical realism over strict symmetry. This approach is standard in crystallographic refinement, particularly in cases where atomic positions or occupancies are ambiguous. Thus, assigning W to the 4g position with half occupancy resolves both the occupancy contradiction and bond-length inconsistencies, providing a more accurate representation of the structure.

(4.3) The x,y,z values in the header of Table S1 are incorrectly given in Angstrom but these are fractional coordinates without dimensions.

Reply: The units of x,y,z in Table S1 have been removed, as shown below.

Table S1. Crystallographic parameters for *m*-Nb₁₂WO₃₃.

chemical formula

Nb_{1.50}W_{0.12}O_{4.12}

crystal system	monoclinic
space group	C 2/m
a (Å)	22.3002(15)
b (Å)	3.8279(2)
c (Å)	17.7490(12)
α (deg)	90
β (deg)	123.338(2)
V (Å ³)	1265.78(14)
Z	16
density (calculated) (g/cm ³)	4.793
Temp. (K)	101(2)
abs. coeff. (mm ⁻¹)	9.865
$F(0\ 0\ 0)$	1660
Theta range for data collection (deg)	2.186 to 27.521
reflections collected	14015
Independent reflections	1664 [R(int) = 0.0825]
Completeness to theta = 25.242°	99.6%
refinement method	full-matrix least-squares on F ²
data / restraints / parameters	1664 / 0 / 92
goodness-of-fit on F ²	1.038
final R indices [I > 2sigma(I)]	R1 = 0.0383, wR2 = 0.0851
R indices (all data)	R1 = 0.0484, wR2 = 0.0932
largest diff. peak and hole (Å ⁻³)	1.820/-1.590

	x	y	z	occ.	U_{iso}
Nb1	0.41697(4)	0	0.38056(5)	1	0.0145(2)
Nb2	0.46442(4)	0	0.71313(5)	1	0.00531(16)
Nb3	0.29725(4)	0	0.47235(5)	1	0.00553(16)
Nb4	0.24819(4)	0	0.13672(5)	1	0.00485(16)
Nb5	0.12922(4)	0	0.23061(5)	1	0.00460(16)

Nb6	0.63319(4)	0	0.95611(5)	1	0.00455(16)
W1	0	-0.2500(2)	0	0.4438(16)	0.0049(2)
O1	0.6975(3)	0	0.9200(4)	1	0.0047(12)
O2	0.5250(3)	0	0.6749(4)	1	0.0071(12)
O3	0.3508(3)	0	0.4238(4)	1	0.0098(13)
O4	0.1785(3)	0	0.1766(4)	1	0.0068(12)
O5	0.5496(3)	0	0.8452(4)	1	0.0036(11)
O6	0.3777(3)	0	0.5959(4)	1	0.0071(12)
O7	0.2052(3)	0	0.3488(4)	1	0.0099(13)
O8	0.3250(3)	0	0.2493(4)	1	0.0098(13)
O9	0.5	0	0.5	1	0.0126(19)
O10	0.2779(3)	-0.5	0.4805(4)	1	0.0053(12)
O11	0.4515(3)	-0.5	0.7301(4)	1	0.0035(11)
O12	0.0320(3)	0	0.0991(4)	1	0.0067(12)
O13	0.0733(3)	-0.5	0.0168(4)	1	0.0053(12)
O14	0.1039(3)	-0.5	0.2317(4)	1	0.0073(12)
O15	0.2249(3)	-0.5	0.1039(4)	1	0.0065(12)
O16	0.6496(3)	-0.5	0.9922(4)	1	0.0077(12)
O17	0.4105(3)	-0.5	0.3705(4)	1	0.0123(14)

(5) page 7 center: “suggest that tungsten is in the oxidation state 6+ in both *m*-Nb₁₂WO₃₃ and *dt*-Nb₁₂WO₃₃.” Sentence used twice here.

Reply: We apologize for the mistake. We have corrected it.

“The position of the edge is influenced by the oxidation state of the metal and suggests that tungsten is in the oxidation state 6+ in both *m*-Nb₁₂WO₃₃ and *dt*-Nb₁₂WO₃₃, as in WO₃. The edge is caused mainly by 2p → 5d electron transitions.”

(6) page 7: “The magnitude of the ligand field split is higher -> highest for cations in octahedral field,…”

Reply: We have revised the sentence, as shown below.

“The magnitude of the ligand field split is highest for cations in octahedral field, followed by cations in distorted octahedral environment, and the lowest for cations in tetrahedral coordination.”

(7) page 8: “whereas a weaken -> weak intensity band is observed for $dt\text{-Nb}_{12}\text{WO}_{33}$,”

Reply: We have corrected the sentence; as shown below.

“whereas a weak intensity band is observed for $dt\text{-Nb}_{12}\text{WO}_{33}$, suggesting distortion of the WO_4 tetrahedra^{34,35}”

(8) page 15: The abbreviations “BM” and “MS” should be introduced properly.

Reply: We have explained the abbreviations in a more detailed manner, as shown below.

“ $m\text{-Nb}_{12}\text{WO}_{33}$ with a particle size similar to that of $dt\text{-Nb}_{12}\text{WO}_{33}$ was produced by ball milling (BM) of the initial $m\text{-Nb}_{12}\text{WO}_{33}$ material; the ball milled sample is denoted $m\text{-Nb}_{12}\text{WO}_{33}\text{-BM}$.”

“ $dt\text{-Nb}_{12}\text{WO}_{33}$ dense microspheres (MS) with sizes 1.0-1.3 μm were synthesized via solvothermal method and subsequent calcination at 900 °C under air; this sample is denoted $dt\text{-Nb}_{12}\text{WO}_{33}\text{-MS}$.”

(9) Give the direction of observation for Figures S9, S31 and S32.

Reply: The direction of observation is now included in the caption of Figures S9, S31, and S32, as shown below.

“**Figure S9.** AC-STEM images of $dt\text{-Nb}_{12}\text{WO}_{33}$ viewed along the [001] zone axis, showing blocks of different sizes.”

“**Figure S31.** AC-STEM images of lithiated $dt\text{-Nb}_{12}\text{WO}_{33}$ viewed along the [001] zone axis after the second cycle (metals in octahedral sites and tetrahedral sites are indicated by blue and orange spheres, respectively).”

“**Figure S32.** AC-STEM images of lithiated $m\text{-Nb}_{12}\text{WO}_{33}$ viewed along the [1-10] zone axis after the second cycle (metals in octahedral sites and tetrahedral sites are indicated by blue and orange spheres, respectively).”

(10) Table 2 ff: curvefit -> curve fit or curve fitting

Reply: We have replaced all the “curvefit” terms by “curve fitting”.

“**Table S3.** Curve fitting parameters of the W L_{III} -edge EXAFS spectrum of $m\text{-Nb}_{12}\text{WO}_{33}$.”

“**Table S4.** Curve fitting parameters of the W L_{III} -edge EXAFS spectrum of $dt\text{-Nb}_{12}\text{WO}_{33}$.”

“**Table S5.** Curve fitting parameters of the Nb K-edge EXAFS spectrum of $m\text{-Nb}_{12}\text{WO}_{33}$.”

“**Table S6.** Curve fitting parameters of the Nb K-edge EXAFS spectrum of $dt\text{-Nb}_{12}\text{WO}_{33}$.”

Reviewer #3

During the revision process, the authors significantly improved the overall quality of the manuscript. However, I still have concerns regarding the commercial viability of Nb-oxide anode materials for electric vehicle (EV) applications. The authors state the following: "As for niobium, the price remained stable at around 40–50 USD/kg since 2010, and was 50 USD/kg in 2017. Currently, the prices of niobium and tungsten are approximately 80 USD/kg and 43 USD/kg, respectively."

In comparison, the current price of graphite is around 5–10 USD/kg, while Si/C composites are priced at approximately 60–100 USD/kg. However, Si-based anode materials offer much higher energy densities (200–300 Wh/kg), which provide a compelling advantage for EV markets. In contrast, Nb-oxide still suffers from relatively low energy densities and requires a larger volume of material in battery packs, which may hinder its competitiveness in practical EV applications. It seems usage of Nb-oxides have no beneficial as an anode material.

Reply: Although graphite and Si/C have advantages in terms of cost and energy density compared with Nb-based oxides as anode materials for lithium-ion batteries, Nb-based oxides have other advantages that makes them promising in particular for fast-charging applications. The low operating potential and slow lithium-ion intercalation kinetics increase the risk of safety issues due to dendrite growth during fast-charging. Si-based anode materials still exhibit limited cycling life due to high volume expansion (300-400%), and the high silicon content in Si/C composites reduces the rate capability due to sluggish lithium diffusion in silicon (1). Nb-based oxides have moderate working voltage, fast ion transfer kinetics, high safety and minimal volume expansion, which are beneficial properties as anode materials for lithium-ion batteries for fast-charging applications.

Since 2010, the Toshiba Corporation of Japan has filed more than 90 patents for Nb-based oxides, especially for titanium niobium oxides and associated battery systems, which established a strong foundation for commercialization (2). In fact, Toshiba has announced that their next-generation lithium-ion batteries, named *SCiB™Nb*, will be available from Spring 2025. These batteries are based on titanium niobium oxides as anode material (3). Titanium niobium oxides have twice the theoretical volume density of graphite-based electrodes used in lithium-ion batteries. In addition, in 2024, Toshiba in collaboration with the Sojitz Corporation of Japan and CBMM of Brazil (the world's leading producer of niobium) have first unveiled a prototype of an ultra-fast charging

electric bus powered by the next-generation lithium-ion batteries with niobium titanium oxide anodes, which has an ultra-fast charging time of around 10 minutes, and delivers high energy density and long-term stability (4). Therefore, Nb-based oxides not only demonstrate good fast-charging capability at the laboratory scale, but also hold potential for practical electric vehicle applications where high-power density is needed.

(1) *Energy Storage Mater.* **2022**, 46, 482.

(2) *Chem. Mater.* **2021**, 33, 4.

(3) <https://www.global.toshiba/ww/products-solutions/battery/scib/product-next/next/nto/nb.html>

(4) <https://www.global.toshiba/ww/news/corporate/2024/06/news-20240620-01.html>

Figure R1: Section of Figure 2b with line intensity profiles. In 1 and 2, CS marks the positions of the crystallographic shear planes. There is a clear tendency that the metal positions in the block centers appear brighter than the ones in the CS planes (see also Figure 2f). There is no evidence for a preferred occupancy of the tetrahedral sites by W, see line 5. However, at other tetrahedral sites, there is a slightly higher brightness (see also Figure 2e) but never as high as expected for 100% W vs. 100% Nb.